Manuscript prepared for Atmos. Chem. Phys.
with version 2014/09/16 7.15 Copernicus papers of the LaTeX class copernicus.cls.
Date: 20 December 2017

# Modeling the Physical Multi-Phase Interactions of $HNO_3$ Between Snow and Air on the Antarctic Plateau (Dome C) and coast (Halley)

Hoi Ga Chan[1,2], Markus M. Frey[1], and Martin D. King[2]

[1]British Antarctic Survey, Natural Environment Research Council, Cambridge, CB3 0ET,UK
[2]Department of Earth Sciences, Royal Holloway University of London, Egham, Surrey, TW20 0EX, UK

*Correspondence to:* Markus M. Frey
(maey@bas.ac.uk)

**Abstract.** Emissions of nitrogen oxide ($NO_x$ = NO + $NO_2$) from the photolysis of nitrate ($NO_3^-$) in snow affect the oxidising capacity of the lower troposphere especially in remote regions, of high latitudes with little pollution. Current air-snow exchange models are limited by poor understanding of processes and often require unphysical tuning parameters. Here, two multi-phase models were developed from physically-based parameterisations to describe the interaction of nitrate between the surface layer of the snowpack and the overlying atmosphere. The first model is similar to previous approaches and assumes that below a threshold temperature, $T_o$, the air-snow grain interface is pure ice and above $T_o$, a disordered interface (DI) emerges covering the entire grain surface. The second model assumes that air-ice interactions dominate over all temperatures below melting of ice and that any liquid present above the eutectic temperature is concentrated in micropockets. The models are used to predict the nitrate in surface snow constrained by year-round observations of mixing ratios of nitric acid in air at a cold site on the Antarctic Plateau (Dome C, $75°06′S, 123°33′E$, 3233 m a.s.l.) and at a relatively warm site on the Antarctic coast (Halley, $75°35′S, 26°39′E$, 35 m a.s.l). The first model agrees reasonably well with observations at Dome C ($C_v(RMSE)$ = 1.34), but performs poorly at Halley ($C_v(RMSE)$ = 89.28) while the second model reproduces with good agreement observations at both sites ($C_v(RMSE)$ = 0.84 at both sites). It is therefore suggested that in winter air-snow interactions of nitrate are determined by non-equilibrium surface adsorption and co-condensation on ice coupled with solid-state diffusion inside the grain, similar to Bock et al. (2016). In summer, however, the air-snow exchange of nitrate is mainly driven by solvation into liquid micropockets following Henry's law with contributions to total surface snow $NO_3^-$ concentrations of 75% and 80% at Dome C and Halley respectively. It is also found that the liquid volume of the snow grain and air-micropocket partitioning of $HNO_3$ are sensitive to both the total solute concentration of

mineral ions within the snow and pH of the snow. The second model provides an alternative method to predict nitrate concentration in the surface snow layer which is applicable over the entire range of environmental conditions typical for Antarctica and forms a basis for a future full 1D snowpack model as well as parameterisations in regional or global atmospheric chemistry models.

## 1 Introduction

Emissions of nitrogen oxides, $NO_x = NO + NO_2$, from snow to the overlying air as a result of photolysis of the nitrate anion, $NO_3^-$, within snow have been observed in polar (Jones et al., 2001; Beine et al., 2002) and midlatitude regions (Honrath et al., 2000). They were found to have a significant impact on the oxidising capacity of the atmospheric boundary layer, especially in remote areas, such as the polar regions, where anthropogenic pollution is small (Grannas et al., 2007). The cycling of NO and $NO_2$ in the troposphere alters the concentration of tropospheric ozone, $O_3$, partitioning of hydroxy radicals, $HO_x$, and organic peroxy radicals, $RO_x$. Tropospheric ozone is a pollutant and a greenhouse gas, and changes in the concentration can impact the regional energy balance and therefore climate (Fowler et al., 2008). Conversely, $HO_x$ radicals are responsible for removal of many atmospheric pollutants (e.g. Gligorovski et al., 2015), such as the greenhouse gas methane, and $RO_x$ radicals play an important role in the oxidation of volatile organic compounds (VOCs). Furthermore, $NO_x$ emission from $NO_3^-$ in snow imply post-depositional loss of $NO_3^-$, which complicates the interpretation of $NO_3^-$ measured in polar ice cores (Wolff et al., 2008; France et al., 2011).

The exchange of nitric acid, $HNO_3$, between the atmosphere or snow interstitial air and snow grains is complex, and is controlled by chemical and physical processes. The relative contribution of photochemical and physical processes has been a matter of debate (Röthlisberger et al., 2000). Isotopic studies have shown that photolysis of $NO_3^-$ is the dominating loss process of $NO_3^-$ in snow (Frey et al., 2009; Erbland et al., 2013). Based on a typical photolysis rate coefficient of nitrate, $J_{NO_3^-}$ $\approx 1 \times 10^{-7}$ s$^{-1}$ (at the surface in Dome C at a solar zenith angle of 52°, France et al., 2011), the characteristic time for nitrate photolysis is $\sim 10^7$ s. Thus, the characteristic time of nitrate photolysis is much larger compared to other physical processes near the snowpack surface, such as grain surface adsorption and solid-state diffusion (Table 1). At the top few mm of snowpack, hereafter called the skin layer and the focus region of snowpack in this paper, the physical uptake of nitrate is much quicker than the photochemical loss due to the availability of nitric acid at the snowpack surface. Therefore, it is assumed that the photochemical processes are negligible, and only physical processes are considered. The skin layer is defined as the top 4 mm of the snowpack, which is the depth of which the surface snow nitrate samples were collected at Dome C (Sect. 4.1).

The snow grain and the air around it form together a complex multiphase interface (Bartels-Rausch et al., 2014). Gaseous $HNO_3$ can be taken up by different reservoirs in snow, for example the molecule can 1) adsorb on the ice surface; 2) diffuse into the ice crystal and form solid solution;

3) co-condense to the growing ice or 4) dissolve into the liquid solution located in grain boundaries, grooves at triple junctions or quadruple points.

Air-snow models have been developed to predict the exchange of trace gases between the snow-pack and the overlying atmosphere and the greatest challenge faced currently is the model description of the air-snow grain interface. One group of models assume a disordered interface, DI, at the snow grain surface with liquid-like properties (e.g. Boxe and Saiz-Lopez, 2008; Thomas et al., 2011; Toyota et al., 2014; Murray et al., 2015). The DI is defined as a thin layer on the surface of the snow

grain and is assumed to have the following characteristics; 1) DI reaction and partition rate constants are similar to those in the aqueous phase, e.g. Henry's Law coefficients are used to describe the partitioning between air and the DI; 2) DI thickness of pure ice ranges from <1 to ∼100 nm based on observations (Bartels-Rausch et al., 2014) but is often set to an arbitrary value, e.g. 10 nm (Thomas et al., 2011; Murray et al., 2015); and 3) all (Toyota et al., 2014) or a fraction (Thomas et al., 2011;

Murray et al., 2015) of the total solutes are located in the DI.

      Another group of models assumes the interface between snow grain and surrounding air to be ice (e.g. Hutterli et al., 2003; Bock et al., 2016). The distribution of hydrogen peroxide, $H_2O_2$, and formaldehyde, HCHO, within the snowpack has been estimated using a physical air-snow and firn transfer model which included temperature driven 'Air-Ice' uptake and release (Hutterli et al., 2003;

McConnell et al., 1998). The bulk concentration of $H_2O_2$ is determined by solid-state diffusion of $H_2O_2$ in ice while the bulk concentration of HCHO is determined by linear isotherm adsorption of HCHO on ice. A physical exchange model has been developed by Bock et al. (2016) to describe the concentration of $NO_3^-$ in the skin layer at Dome C, East Antarctic Plateau. Bock et al. (2016) proposed that the skin layer snow nitrate concentration at Dome C is determined by thermodynamic

equilibrium ice solubility on the grain surface followed by solid-state diffusion during winter. During summer the large increase in $NO_3^-$ concentration in the skin layer snow is mainly attributed to co-condensation of $HNO_3$ and $H_2O$. However, Bock et al. (2016) model implies no loss of $NO_3^-$ due to sublimation, a process that has been suggested to be important in surface snow dynamics (Röthlisberger et al., 2000).

Both types of models require tuning parameters used to fit the model output to a chosen set of observations. Some of these parameters do have a physical meaning yet the tuned values may not, for example the fraction of solute in the DI (Thomas et al., 2011) or the ion partitioning coefficients (Hutterli and Röthlisberger, 1999). Whereas some may not have a strict physical meaning, for example the co-condensation related parameters were adjusted in Bock et al. (2016) model, one

of their configurations (configuration 2-BC2), total snow nitrate concentration contributed by co-condensation, which is the simultaneous condensation of water vapour and trace gases at the air-ice interface, has an empirical relationship with the partial pressure of nitric acid and water vapour while in another configuration (configuration 2-BC3) they varied the complementary error function when calculating the contribution from co-condensation to match the modelled results to the observations.

Any 'tuning' of a model to a specific set of observations may affect the confidence in model runs under different conditions or scenarios.

The aim of this paper is to develop a physical exchange model based on physical parameterisations and experimental data to describe the exchange of nitrate between the atmosphere and the skin layer of snow and minimising the number of tuning parameters. It is a first step towards a full snowpack

model that would include deeper snow and other processes, such as wind pumping, molecular diffusion, and photochemistry. Two temperature dependent, multi-phase models (Model 1 and Model 2), are developed to evaluate two different concepts to describe the interaction of nitrate between air and snow.

Model 1 is based on the hypothesis of the existence of a DI covering the entire snow grain above

105 a threshold temperature, $T_o$ (Sect. 3.1). Below $T_o$, the interface between snow grain and air is assumed to be 'Air-Ice', and the concentration of $NO_3^-$ at the grain boundary is determined by non-equilibrium surface adsorption and co-condensation coupled with solid-state diffusion into the grain. Above $T_o$, the interface is assumed to be 'Air-DI' of which the $NO_3^-$ concentration is defined by non-equilibrium solvation into the DI based on Henry's Law coefficient. This is similar to the approach

taken by other models (e.g. Thomas et al., 2011; Toyota et al., 2014).

Model 2 is based on the hypothesis of Cho et al. (2002), that liquid co-exists with ice above eutectic temperature, $T_e$. The liquid forms micropockets and is assumed to be located in grooves at grain boundaries or triple junctions due to the limited wettability of ice (Domine et al., 2013). Therefore, at all temperatures below melting the major interface between air and snow grain is

115 assumed to be pure ice and the concentration of $NO_3^-$ in ice is defined by non-equilibrium surface adsorption and co-condensation followed by solid-sate diffusion within the grain. Above $T_e$, the partitioning of $HNO_3$ to the liquid micropockets is described by Henry's Law (Sect. 3.2).

The models are validated with available observations from two sites in Antarctica that have very different atmospheric composition, temperatures and humidities: Dome C on the East Antarctic

Plateau and Halley in coastal Antarctica.

## 2   Current Understanding of Physical Air-Snow Processes

Below we briefly review the current understanding of physical air-snow processes, which are relevant to nitrate. A more comprehensive discussion can be found in a recent review paper (Bartels-Rausch et al., 2014).

### 2.1   Surface Adsorption at the Air-Ice Interface

The probability of a gas molecule being adsorbed on a clean ice surface can be described by the dimensionless surface accommodation coefficient, $\alpha$ (Crowley et al., 2010). The adsorbed molecule can then be desorbed thermally or it can be dissociated and diffuse into the bulk and form a solid

solution (Abbatt, 1997; Huthwelker et al., 2004; Cox et al., 2005). At a low partial pressure of $HNO_3$,
the adsorption of $HNO_3$ on an ice surface can be described by the single-site Langmuir adsorption
(Ullerstam et al., 2005b):

$$HNO_{3,(g)} + S \underset{k_{des}}{\overset{k_{ads}}{\rightleftharpoons}} HNO_{3,(ads)} \tag{R1}$$

where $HNO_{3,(g)}$ and $HNO_{3,(ads)}$ are the gas-phase and surface adsorbed nitric acid, and $S$ is the
surface site for adsorption. The concentration of surface sites, [S], i.e. number of site available per
unit volume of air, is defined as follow:

$$[S] = (1 - \theta) N_{max} \frac{A_{ice}}{V_{air}} \tag{1}$$

Here, $\theta$ is the fraction of surface sites being occupied, $N_{max}$ is the maximum number of surface
sites with a unit of $molecule\, m_{ice}^{-2}$, $A_{ice}$ is the surface area of ice per unit volume of snowpack with
a unit of $m_{ice}^2\, m_{snowpack}^{-3}$, and $V_{air}$ is the volume of air per unit volume of snowpack with a unit
of $m_{air}^3\, m_{snowpack}^{-3}$. Note that [S] has units of $molecule\, m^{-3}$. The adsorption coefficient, $k_{ads}$ ,and
desorption coefficient, $k_{des}$, in R1 are defined as

$$k_{ads} = \frac{\alpha \overline{v}}{4} \frac{1}{N_{max}} \tag{2}$$

$$k_{des} = \frac{k_{ads}}{K_{eq}} \tag{3}$$

Note that $k_{ads}$ has a unit of $m^3\, molecule^{-1}\, s^{-1}$ while the unit of $k_{des}$ is $s^{-1}$, $\overline{v}$ is the average gas-
phase molecular speed and $K_{eq}$ is the equilibrium constant for Langmuir adsorption on ice with
a unit of $m^3\, molecule^{-1}$. The value of $K_{eq}$ for $HNO_3$ is inversely correlated with temperature
because the scavenging efficiency of $HNO_3$ due to adsorption increases as temperature decreases.
The parameterisations and values for the above variables used in this study are listed in Table A1. The
value of the accommodation coefficient, $\alpha$, is the same as the experimental initial uptake coefficient,
$\gamma_0$, if the time resolution of the laboratory experiments is high enough (Crowley et al., 2010). Fig. A1
shows the experimental initial uptake coefficients, $\gamma_0$, by various studies as a function of temperature.
A comparison of different parameterisations of $K_{eq}$ is shown in Fig. A2.

## 2.2 Solid-State Diffusion

Due to its solubility and diffusivity, $HNO_3$ can form a solid solution in ice. The solid-state diffu-
sion in natural snow was found to be an important process for understanding the partitioning of
highly soluble gases, including $HNO_3$, between the atmosphere and snow (Bartels-Rausch et al.,
2014). Thibert et al. (1998) derived a solid-state diffusion coefficient, $k_{diff}$, and a thermodynamic
solubility of $HNO_3$ in ice from sets of $HNO_3$ concentration diffusion profiles obtained by exposing
single ice crystal to diluted $HNO_3$ at different temperatures for a period of days to weeks. However,
Thibert et al. (1998) did not present the kinetics of $HNO_3$ uptake on ice and hence a characteristic
time for equilibrium between air and ice could not be established. A diffusion-like behaviour has

been observed from flow-tube studies for trace gas uptake onto ice (e.g. Abbatt, 1997; Huthwelker et al., 2004; Cox et al., 2005) suggesting the solid-state diffusion of nitrate molecules can occur concurrently with surface adsorption (R1), such that

$$\text{HNO}_{3,\,(\text{ads})} \overset{k_{\text{diff}}}{\rightleftarrows} \text{HNO}_{3,\,(\text{ice})} \tag{R2}$$

where $\text{HNO}_{3,\,(\text{ice})}$ is the nitric acid incorporated into the ice matrix.

### 2.3 Coexistence of Liquid Solution with Ice

Liquid aqueous solution coexists with ice in the presence of soluble impurities, such as sea salt and acids. The liquid exist down to the eutectic temperature defined by the composition and solubility of the impurities in the ice. Cho et al. (2002) parameterised the liquid water fraction, $\phi_{\text{H}_2\text{O}}(T)$, as a function of total ionic concentration of impurities, $\text{Ion}_{\text{tot}}$, and temperature as follows:

$$\phi_{\text{H}_2\text{O}}(T) = \frac{\overline{m}_{\text{H}_2\text{O}} R T_f}{1000 \Delta H_f^0} \left( \frac{T}{T_f - T} \right) \Phi_{\text{bulk}}^{\text{aq}} \left[ \text{Ion}_{\text{tot}\,(\text{bulk})} \right] \tag{4}$$

where $\phi_{\text{H}_2\text{O}}(T)$ has a units of $\text{m}_{\text{liquid}}^3\,\text{m}_{\text{liquid+solid}}^{-3}$, $\overline{m}_{\text{H}_2\text{O}}$ is the molecular weight of water, $R$ is the ideal gas constant, $T_f$ is the freezing temperature of pure water in K, $\Delta H_f^0$ is the enthalpy of fusion in $\text{J}\,\text{mol}^{-1}$, $\Phi_{\text{bulk}}^{\text{aq}}$ is the fraction of the total solute in the aqueous phase and $[\text{Ion}_{\text{tot, bulk}}]$ is the total ionic concentration in the melted sample. There are different hypothesises regarding the location of the liquid solution. Most studies assume the liquid solution forms a thin layer covering the whole grain surface (e.g. Kuo et al., 2011) while Domine et al. (2013) suggested the liquid is located in grooves at grain boundaries and triple junctions. The arguments of the latter study were 1) the ionic concentration is so low in natural snow that only a small amount of liquid can be formed; and 2) the wettability of ice by liquid water is imperfect, preventing the liquid drop from spreading out across the entire solid surface. The volume of liquid is small relative to the ice grain and if spread uniformly across the ice grain the thickness would be less than the diameter of the $\text{H}_2\text{O}$ molecule which is unrealistic.

The partitioning of atmospheric acidic gases between air and the liquid fraction of snow can be described by Henry's law using the effective dimensionless Henry's law coefficient, $k_{\text{H}}^{\text{eff}}$, (Sander, 1999)

$$k_{\text{H}}^{\text{eff}} = k_{\text{H}}^{\text{cc}} \frac{\text{K}_{\text{a}}}{[\text{H}_{(\text{aq})}^+]} \tag{5}$$

where $k_{\text{H}}^{\text{cc}}$ is the dimensionless temperature dependent Henry's Law coefficient (App. A), $\text{K}_{\text{a}}$ is the acid dissociation constant and $[\text{H}_{(\text{aq})}^+]$ is the concentration of hydrogen ions. Fig. A3 shows the temperature and pH dependence of $k_{\text{H}}^{\text{eff}}$. At a given temperature, $k_{\text{H}}^{\text{eff}}$ increases by an order of magnitude between pH 5 and 6.5 (Fig. A3 A), the typical range of pH in natural snow (Udisti et al., 2004). While at a given pH, $k_{\text{H}}^{\text{eff}}$ decreases by 2 orders of magnitude between -40°$C$ and 0°$C$ (Fig. A3 B). Note that the range of pH measured by Udisti et al. (2004) is the pH of the melted sample,

which might be different from the pH of the liquid fraction of the snow grain not observable by current measurement techniques.

## 3  Modelling Approach

The aim of this paper is to focus on the physical exchange mechanisms of $HNO_3$ between air and snow to predict the concentration of nitrate in the skin layer of the snowpack, as a first step towards a
full snowpack model. The two models are constrained by the observed atmospheric concentration of $HNO_3$, air temperature, skin layer temperature, atmospheric pressure and humidity. The loss or gain in the atmospheric $HNO_3$ due to the mass exchange between air and snow are included implicitly by constraining the models with the observed atmospheric concentration of $HNO_3$. The following assumptions were made in both Model 1 & 2: 1) the concentration of $HNO_3$ in snow interstitial
air is the same as in the overlying atmosphere justified by a short characteristic time scale for gas-phase diffusion of $\sim 1$ s (Table 1); 2) the physical properties of the skin layer are homogeneous and include density and specific surface area (SSA); and 3) the snow grain is assumed to be a radially symmetrical sphere with an effective radius, $R_{eff}$, which is estimated from the SSA as the follows:

$$R_{eff} = \frac{3}{\rho_{ice} SSA} \tag{6}$$

where $\rho_{ice}$ is the density of ice. Snow metamorphism and resulting changes in snow grain size are not modeled explicitly, but are approximated instead by prescribing temporal changes in SSA. Here an annual cycle of SSA is included based on observations at Dome C (Picard et al., 2016), ranging from 25 $m^2 kg^{-1}$ in summer to 90 $m^2 kg^{-1}$ in the winter (details in Sect. 4.3 and Fig. A4a), and yielding a $R_{eff}$ of $\sim 130$ μm in summer, which gradually reduces to $\sim 30$ μm in winter (Fig. A4b).
Modeled co-condensation (Eq. 9 & 10) does not change model snow grain size, since the involved ice volumes are relatively small compared to the volume of the snow grain. The model set up implies also that the snow grain size remains constant during each model time step of $\Delta t = 10$ min.

For the calculation of solid-state diffusion the snow grain is divided into $N$ concentric shells of equal thickness. To optimise model performance and computational cost, the number of concentric
shells is fixed to $N = 85$, yielding a model shell thickness $\Delta r$ of $\sim 1.5$ μm in summer and $\sim 0.5$ μm in winter due to seasonal change in grain size. $\Delta r$ remains at all times smaller than the minimum typical length-scale, $<x>$, a molecule diffuses over a finite time, $\Delta t$, and described by the root-mean square displacement, $<x> = \sqrt{6 \Delta t \, k_{diff}}$. Minimum typical length-scales occur in winter when air temperatures are lowest, and for a modeling time step, $\Delta t = 10$ min, they range between 1.5 μm at
Dome C and 5.5 μm at Halley.

### 3.1  Model 1 - Surface Adsorption/Solvation & Solid Diffusion

In Model 1, the uptake of $HNO_3$ is treated as a two-step process consisting of interfacial mass transport across the air-snow grain boundary and subsequent diffusion into the bulk, a similar approach as

taken by Bock et al. (2016). Below a threshold temperature, $T_o$, the snow grain boundary is assumed to be 'Air-Ice' and the concentration of the outermost model shell is determined by the combination of adsorption and co-condensation on ice (details in Sect. 3.1.1 & Fig. 1a). Above $T_o$, the air-snow grain boundary is assumed to be 'Air-DI', and the concentration of the outermost model shell is determined by solvation governed by Henry's law into the disordered interface, DI, (Details in Sect. 3.1.2 & Fig. 1b).

The threshold temperature, $T_o$, is a value based on lab experiments. The temperature at which a disordered interface is detected on pure ice varies between 238 and 270 K depending on the measurement technique (Domine et al., 2013 and references therein). Here, $T_o$, is set to 238 K, the lower end of the range. Model uncertainties due to the uncertainties in $T_o$ are evaluated in a sensitivity study further below (Sect. 6.5).

The physical properties of the DI are still poorly known, and currently there are no physical parameterisations available to estimate DI thickness, partitioning coefficients or diffusivities. Hence, for the DI in Model 1 the following four assumptions are made: 1) the partitioning between air and the DI follows Henry's law, similar to previous models (e.g.Thomas et al., 2011 & Toyota et al., 2014); 2) the model geometry described above implies that the DI, i.e. the outermost model shell of the snow grain, follows the seasonal cycle of snow grain specific surface area and has a thickness of 1.5 μm in summer decreasing to 0.5 μm in winter. A seasonal cycle is qualitatively consistent with laboratory measurements, which show that DI thickness increases with temperature (Bartels-Rausch et al., 2014). But the absolute model values are larger than previous lab measurements on pure ice, which range from the thickness of a monolayer of water (0.3 nm) to $\sim$100 nm, depending on the measurement technique (e.g. Bartels-Rausch et al., 2014), or values adopted in previous model studies (range 10-30 nm) (e.g. Thomas et al., 2011, Toyota et al., 2014, Murray et al., 2015). However, DI thickness is also sensitive to the type and concentration of impurities, and generally increases with ion concentration (e.g. Dash et al., 2006; Bartels-Rausch et al., 2014); 3) the DI is interacting with the bulk ice, i.e. solvated nitrate ions diffuse into the interior of the snow grain and the mass transport is determined by the solid-state diffusion coefficient of ice, $k_{\mathrm{diff}}$ and the concentration gradient across the snow grain; and 4) the solid-state concentration of nitrate in the bulk is limited by the thermodynamic equilibrium solubility of ice (e.g. by Thibert et al., 1998 as shown in Eq. 19), except the outermost model shell of the snow grain.

### 3.1.1 $T \leq 238$ K: Non-Equilibrium Surface Adsorption & Co-condensation

At a temperature below $T_o = 238$ K the interface between air and snow grain is assumed to be pure ice. The concentration of nitrate at the grain boundary, $[\mathrm{HNO_{3\,(surf)}}]$, is determined by a combination of non-equilibrium kinetic adsorption and co-condensation:

$$[\mathrm{HNO_{3\,(surf)}}] = [\mathrm{HNO_{3\,(ads)}}] + [\mathrm{HNO_{3\,(cc)}}] \qquad \text{if} \quad T \leq 238\mathrm{K} \qquad (7)$$

where $[\mathrm{HNO}_{3\,(\mathrm{ads})}]$ is the concentration contributed by the sum of surface adsorption and desorption and $[\mathrm{HNO}_{3\,(\mathrm{cc})}]$ is the concentration contributed by co-condensation or co-sublimation. This configuration but without the contribution by co-condensation is referred to as 'Model 1 - BCice', where 'BC' stands for boundary condition. The net rate of adsorption can be described as $\frac{d[\mathrm{HNO}_{3\,(\mathrm{ads})}]}{dt} = k_{\mathrm{ads}}[\mathrm{HNO}_{3\,(\mathrm{g})}]\,[\mathrm{S}] - k_{\mathrm{des}}[\mathrm{HNO}_{3\,(\mathrm{ads})}]$. Substituting $k_{\mathrm{des}}$ with Eq. (3), the net adsorption rate is expressed as

$$\frac{d[\mathrm{HNO}_{3\,(\mathrm{ads})}]}{dt} = k_{\mathrm{ads}}\left([\mathrm{HNO}_{3\,(\mathrm{g})}]\,[\mathrm{S}] - \frac{[\mathrm{HNO}_{3\,(\mathrm{ads})}]}{K_{\mathrm{eq}}}\right) \tag{8}$$

Ullerstam et al. (2005b) have shown that for partial pressures of $\mathrm{HNO}_3$ lower than $10^{-5}$ Pa the ice surface is not entirely covered with $\mathrm{HNO}_3$, and therefore, undersaturated. The annual average atmospheric partial pressure of $\mathrm{HNO}_3$ recorded at Dome C is $\sim 10^{-6}$ Pa (Traversi et al., 2014) and is $\sim 10^{-7}$ Pa at Halley (Jones et al., 2008), hence, the ice surface is unlikely to be saturated with $\mathrm{HNO}_3$. A non-equilibrium kinetic approach is taken instead of an equilibrium adsorption as natural snowpacks are constantly undergoing sublimation and condensation of $\mathrm{H}_2\mathrm{O}$, especially in the skin layer, due to temperature gradients present over a range of timescales from a fraction of seconds to days and seasons (Bartels-Rausch et al., 2014). Pinzer et al. (2012) observed that up to 60% of the total ice mass was redistributed under a constant temperature gradient of 50 K m$^{-1}$ over a 12 hour period. Field observations (Frey et al., 2013) and the results from a heat transfer model (Hutterli et al., 2003) at Dome C in summer show temperature gradients of 71 K m$^{-1}$ across the top 2 cm and 130 K m$^{-1}$ across the top 4 mm of the snowpack, respectively. At Halley, the modelled summer temperature gradient in the top cm of snow is about 41 K m$^{-1}$. Therefore, the dynamic $\mathrm{H}_2\mathrm{O}$ exchange and redistribution at the snow grain surface prevent the equilibrium of adsorption from being reached and require a kinetic approach.

The temperature gradient and relative humidity gradient between the surface of the snowpack and the skin layer create a gradient in water vapour pressure, which drives condensation or sublimation of ice, depending on the sign of the gradient. Uptake of $\mathrm{HNO}_3$ molecules to growing ice is known as co-condensation. The surface concentration of $\mathrm{NO}_3^-$ contributed by co-condensation or co-sublimation, $[\mathrm{HNO}_{3\,(\mathrm{cc})}]$, is given by

$$[\mathrm{HNO}_{3(\mathrm{cc})}] = X_{\mathrm{HNO}_3}\,\frac{\rho_{ice}\,N_A}{\overline{m}_{\mathrm{H}_2\mathrm{O}}}\,\frac{\Delta t}{\mathrm{V}_{\mathrm{grain}}}\,\frac{d\mathrm{V}}{dt} \tag{9}$$

where $X_{\mathrm{HNO}_3}$ is the mole fraction of $\mathrm{HNO}_3$ condensed along with water vapour ($X_{\mathrm{HNO}_3} = 10^{-3.2}\,P_{\mathrm{HNO}_3}^{0.56}$, Ullerstam and Abbatt, 2005a), $\rho_{ice}$ is the density of ice (in kg m$^{-3}$), and $N_A$ is Avogadro's constant ($6.022 \times 10^{23}$ molecule mol$^{-1}$). The rate of volume change of snow grain, $\frac{d\mathrm{V}}{dt}$, is specified by the growth law described by Flanner and Zender (2006)

$$\frac{d\mathrm{V}}{dt} = \frac{4\pi\,\mathrm{R}_{\mathrm{eff}}^2}{\rho_{ice}}\,D_v\left(\frac{d\rho_v}{dx}\right)_{x=r} \tag{10}$$

where $\mathrm{D}_v$ is the diffusivity of water vapour in air and $\frac{d\rho_v}{dx}$ is the local water vapour density gradient, i.e. between air away from the snow grain and the air near the grain surface. However, to the author's

knowledge there are no observations reported and the calculation of water vapour density at these microscopic scales is computational costly as it would require 3-D modelling of the metamorphism of the snow grain. For simplicity, the macroscopic (few mm) water vapour gradient across the skin layer was used to estimate the rate of volume change of snow grain due to condensation or sublimation, i.e. $\left(\frac{d\rho_v}{dx}\right)_{x=r}$ in Eq. 10 is replaced by $\left(\frac{d\rho_v}{dz}\right)_{z=4\text{mm}}$. The water vapour density, $\rho_v$, can be calculated as follows:

$$\rho_v = \frac{P_{sat}\,\text{RH}}{100\,R_v\,T} \tag{11}$$

where $P_{sat}$ is the saturated vapour pressure (Pa), RH is the relative humidity (%), $R_v$ is the gas constant ($\text{J kg}^{-1}\,\text{K}^{-1}$) and $T$ is temperature (K). There are no measurements of fine resolution of vertical snow profile of RH and temperature available, therefore, RH within the snowpack was assumed to be 100% and the temperature of the skin layer is estimated using a heat transfer temperature model based on the heat diffusion equation (Hutterli et al., 2003):

$$\frac{\partial T}{\partial t} = \frac{\partial}{\partial z} k_w(z) \frac{\partial T}{\partial z} \tag{12}$$

where $T$ is the temperature, $t$ is time, $k_w$ is the thermal conductivity (App. A, Table A1) of snowpack and $z$ is the depth.

### 3.1.2  $T > 238$ K: Non-Equilibrium Solvation

At temperatures above $T_o = 238$ K the interface between air and the entire surface of the snow grain is assumed to be a DI.

$$[\text{HNO}_{3\,(\text{surf})}] = [\text{HNO}_{3\,(\text{DI})}] \qquad \text{if} \quad T > 238\text{K} \tag{13}$$

The DI is also assumed to be out of equilibrium with the surrounding air as the exchange of water molecules at the surface of the snow grain is expected to be rapid that the surface is redistributed before equilibrium is reached (Details in Sect. 3.1.1). The concentration of the DI is then defined by the following equation:

$$\frac{d[\text{HNO}_{3\,(\text{DI})}]}{dt} = k_{\text{mt}} \left( [\text{HNO}_{3\,(\text{g})}] - \frac{[\text{HNO}_{3\,(\text{DI})}]}{k_{\text{H}}^{\text{eff}}} \right) \tag{14}$$

The mass-transfer coefficient, $k_{\text{mt}}$, is defined as $k_{\text{mt}} = \left( \frac{R_{\text{eff}}^2}{3 D_g} + \frac{4 R_{\text{eff}}}{3 \bar{v} \alpha} \right)^{-1}$, where $D_g$ is the gas-phase diffusivity (Sander, 1999). Note that in this model the concentration of the DI is used as the outermost boundary condition for solid-state diffusion within the grain (See Sect. 3.1.3) and the transfer of $\text{NO}_3^-$ into the bulk is limited by the concentration gradient across the snow grain, the maximum solubility and diffusivity of ice.

### 3.1.3  Solid-State Diffusion

The concentration gradient between the grain boundary and its centre drives solid state diffusion of nitrate within the bulk ice. The $\text{NO}_3^-$ concentration profile within the snow grain can be found by

solving the following partial differential equation

$$\frac{\partial [\text{NO}_3^-](n)}{\partial t} = k_{\text{diff}} \left( \frac{2}{n} \frac{\partial [\text{NO}_3^-](n)}{\partial n} + \frac{\partial^2 [\text{NO}_3^-](n)}{\partial n^2} \right) \tag{15}$$

where $[\text{NO}_3^-](n)$ is the nitrate concentration in the $n^{th}$ concentric model shell, with $n = 0, 1, 2, \ldots, N$ and $k_{\text{diff}}$ is the solid-state diffusion coefficient, which is assumed to be homogeneous across the snow grain. By substituting $U(n\Delta r) = \frac{n\Delta r}{R_{\text{eff}}} [\text{NO}_3^-](n)$, Eq. 15 can be re-written as

$$\frac{\partial U(n\Delta r)}{\partial t} = k_{\text{diff}} \left( \frac{\partial^2 U(n\Delta r)}{\partial n^2} \right) \tag{16}$$

where $U(n\Delta r)$ is the concentration at distance $n\Delta r$ from the centre of the snow grain, with $N\Delta r = R_{\text{eff}}$. The nitrate concentration at the centre is set to $U(0) = 0$ and at the grain boundary $U(N\Delta r) = [\text{HNO}_{3\,(\text{surf})}]$, which is defined by surface adsorption and co-condensation at temperatures below $T_o$ (Eq. 7) or by solvation into the DI at temperature above $T_o$ (Eq. 13).

The diffusion equation is solved with the Crank-Nicolson scheme (Press et al., 1996) and the bulk concentration of $\text{NO}_3^-$ in the ice grain, $[\text{NO}_{3\,(\text{bulk})}^-]$, is the sum of the number of $\text{NO}_3^-$ molecules in each shell divided by the volume of the whole grain, expressed as

$$[\text{NO}_{3\,(\text{bulk})}^-] = \frac{\sum [\text{NO}_3^-](n)\,V(n)}{\sum V(n)} = \frac{\sum [\text{NO}_3^-](n)\,V(n)}{V_{\text{grain}}} \tag{17}$$

where $V(n)$ is the volume of the $n^{\text{th}}$ layer of the concentric shell, $\sum V(n)$ is the total volume of the grain, $V_{\text{grain}}$, and the concentration of nitrate in the $n^{\text{th}}$ layer can be determined by re-substituting $U$ that $[\text{NO}_3^-](n) = \frac{R_{\text{eff}}}{n\Delta r} U(n\Delta r)$.

### 3.2 Model 2 - Non-Equilibrium Kinetic Adsorption & Solid Diffusion and Equilibrium Air - Liquid Micropocket

Model 2 is based on the hypothesis that the major air-snow grain interface is pure ice at all temperatures below melting temperature, $T_m$, and that liquid coexists with ice when the temperature is above the eutectic temperature, $T_e$ (Fig. 2). The liquid solution is assumed to be located in grooves at grain boundaries or triple junctions between grains and in the form of micropockets. This assumption implies that the grain surface area being covered by liquid solution is negligible. The bulk concentration of $\text{NO}_3^-$ in Model 2 is defined as follows:

$$[\text{NO}_{3\,(\text{bulk})}^-] = \begin{cases} \frac{\sum [\text{NO}_3^-](n)\,V(n)}{V_{\text{grain}}} & \text{if} \quad T < T_e. \\ \frac{\sum [\text{NO}_3^-](n)\,V(n)}{V_{\text{grain}}} + \phi_{\text{H}_2\text{O}}\,k_{\text{H}}^{\text{eff}}\,[\text{HNO}_{3\,(\text{g})}] & \text{if} \quad T_e \leq T < T_m. \end{cases} \tag{18}$$

The term '$\frac{\sum [\text{NO}_3^-](n)\,V(n)}{V_{\text{grain}}}$' in Eq. 18 is representing the nitrate concentration in the ice-phase and is applied to all temperatures below the melting temperature, $T_m$. At $T < T_m$, $\text{HNO}_3$ can be adsorbed/desorbed and co-condensed/co-sublimated from the ice surface as was the case in Model 1

when $T < T_o$ (Sect. 3.1.1). The adsorbed and co-condensed molecules on the ice surface then diffuse into or out of the bulk ice depending on the concentration gradient of nitrate as was the case in Model 1 (Sect. 3.1.3). The nitrate in the snow grain contributed by these processes is referred to as the ice-phase nitrate.

The term '$\phi_{H_2O} \, k_H^{\text{eff}} \, [\text{HNO}_{3\,(g)}]$' in Eq. 18 is representing the nitrate concentration in the liquid-phase when $T \geq T_e$. At $T \geq T_e$, liquid co-exists with ice, and the bulk mass of $\text{NO}_3^-$ is contributed by $\text{NO}_3^-$ located both within the ice and in the liquid micropocket. The volume of liquid can be calculated from the liquid water fraction, $\phi_{H_2O}$ (Eq. 4). The liquid in the micropocket is assumed to be ideal and the partitioning between air and liquid micropocket is described by Henry's Law (Eq. 5). This implies instantaneous equilibrium between air and liquid micropocket, and is justified because; 1) the volume of the liquid solution is small which up to $10^{-7} - 10^{-6}\%$ of the total volume of the ice grain (as discussed below); 2) $\text{HNO}_3$ is strongly soluble in solution; 3) the characteristic time of the interfacial mass transport across a liquid surface of a droplet with 70 μm diameter is only $\sim 10^{-7}$ s (Table 1); and 4) the diffusivity of $\text{HNO}_3$ is faster in liquid-phase ($9.78 \times 10^{-10}$ m$^2$ s$^{-1}$ at 0°C, Yuan-Hui and Gregory, 1974 ) than in ice ($3.8 \times 10^{-14}$ m$^2$ s$^{-1}$ at 0°C). The characteristic time of liquid-phase diffusion within a 70 μm diameter water droplet is $\sim 1$ s (Table 1).

Both the values of pH and $\Phi_{\text{bulk}}^{\text{aq}}$ (in Eq. 4) are updated at each model time step with values from the previous time step. At Dome C, the major anion in melted snow is $\text{NO}_3^-$ (e.g. Udisti et al., 2004). Therefore, it is assumed that nitrate and hydrogen ions are the only ions present in the skin layer snow, i.e. $[\text{Ion}_{\text{tot (bulk)}}] = 2 \times [\text{NO}_3^-]$ in Eq. 4, and the eutectic temperature of a $\text{H}_2\text{O-HNO}_3$ system of 230.64 K (Beyer et al., 2002) is chosen as the threshold temperature for the existence of micropockets. In contrast, at Halley snowpack ion chemistry is dominated by NaCl (Wolff et al., 2008), contributing $\sim 70\%$ to the total ion concentration in the 2004-05 Halley data set, due to the proximity of sea ice and open ocean. Surface snow at Halley also contains a significant amount of sulphate ion, $\text{SO}_4^-$, from sea salt sulphate and sulphuric acid, together contributing $\sim 20\%$ of the total ion concentration. However, for simplicity, the only anions included in the calculation of $\phi_{H_2O}$ at Halley are $\text{NO}_3^-$ and $\text{Cl}^-$, such that $[\text{Ion}_{\text{tot (bulk)}}] = 2 \times ( \, [\text{Cl}^-] + [\text{NO}_3^-])$ in Eq. 4 and the value of $T_e$ used is that for a $\text{H}_2\text{O-NaCl}$ system of 251.95 K (Akinfiev et al., 2001).

### 3.3 Model BC1 by Bock et al. (2016)

Previously Bock et al. (2016) developed a model for air-ice exchange of nitrate in surface snow assuming only air-ice interaction and equilibrium with the surrounding air. They defined the concentration of nitrate in the outermost model shell of the snow grain in their Configuration 2 - BC1 by the thermodynamic equilibrium solubility parameterisation by Thibert et al. (1998):

$$[\text{NO}_3^-](n = N) = 2.37 \times 10^{-12} \exp\left(\frac{3532.2}{T}\right) P_{\text{HNO}_3}^{1/2.3} \frac{\rho_{ice} N_A}{\bar{m}_{H_2O}} \tag{19}$$

where $N$ is the number of concentric shells in the snow grain, $T$ is the snow temperature (K), $P_{\mathrm{HNO_3}}$ is the partial pressure of $\mathrm{HNO_3}$ (Pa) and $\bar{m}_{\mathrm{H_2O}}$ is the molar mass of $\mathrm{H_2O}$. They concluded that the concentration of nitrate in surface snow at Dome C during winter is mainly governed by thermodynamic equilibrium solubility coupled to solid-state diffusion. (Bock et al., 2016)

The configuration after Bock et al. (2016) (referred to as 'Bock - BC1' from hereon) is compared with the non-equilibrium adsorption coupled to solid-state diffusion model presented in this paper ('Model 1 - BCice', Sect. 3.1.1). Note that co-condensation was excluded in these model runs to allow a direct comparison between the two different approaches. The two configurations are analysed and discussed in Sect. 6.1 based on data collection during winter at Dome C and Halley.

## 4 Model Validation

Model calculations are constrained and validated with existing observations of atmospheric $\mathrm{NO_3^-}$, skin layer snow $\mathrm{NO_3^-}$ concentration, and meteorology at Dome C and Halley, which are summarised below.

### 4.1 Observation at Dome C

Dome C is characterised by the following: 1) air temperatures are below the freezing point year round, and no snow melt occurs, with an annual mean of $-52^\circ$C, maximum of $-17^\circ$C in summer (mid November until the end of January) and minimum temperature of $-80^\circ$C in winter (April to mid September) as shown in Fig. 3a (Erbland et al., 2013). The diurnal temperature variation is $\sim$10 K in summer, spring (mid September until mid November) and autumn (February to March). 2) the air-snow chemistry of reactive nitrogen is relatively simple due to the remoteness of the site. In particular, concentrations of sea salt and other particles that may scavenge atmospheric $\mathrm{HNO_3}$ are low on the East Antarctica Plateau (Legrand et al., 2016). Hence, the main atmospheric nitrate is gaseous $\mathrm{HNO_3}$ that dissolves in or adsorbs onto snow grains (Traversi et al., 2014). 3) Furthermore, a low snow accumulation rate of 27 $\mathrm{kg\,m^{-2}\,yr^{-1}}$ (Röthlisberger et al., 2000) leads to significant post-depositional processing of nitrate driven by photolysis before the surface snow is buried by new snowfall (e.g. Röthlisberger et al., 2000; Frey et al., 2009).

Observations of skin layer snow nitrate concentration, atmospheric nitrate concentration, temperature, and pressure were carried out previously at Dome C during January 2009 to 2010 (Erbland et al., 2013) and are shown in Fig. 3. The snow samples were collected from the 'skin layer' snow, the top $4 \pm 2$ mm of the snowpack, approximately every 3 days (Erbland et al., 2013). The skin layer was assumed to be spatially heterogeneous with an uncertainty in thickness of about 20% due to the softness of the uppermost layer and sampling by different people. The nitrate concentration in the melted sample was measured by ion chromatography (Erbland et al., 2013).

The concentration of atmospheric nitrate, i.e. the sum of atmospheric particulate nitrate ($p-NO_3^-$) and the concentration of gaseous nitric acid ($HNO_3$), was collected on glass fibre filters with a high volume air sampler (HVAS) as described in Morin et al. (2008). Erbland et al. (2013) stated that the concentration of atmospheric nitrate shows good agreement with $HNO_3$ gas-phase concentration measured by denuder tubes at Dome C over the same time period, therefore we equate the observed atmospheric nitrate with gaseous $HNO_3$. The filter was positioned approximately 1 m above the snow surface and changed weekly. The atmospheric boundary layer is assumed to be well mixed so that the atmospheric nitrate at the snowpack surface would be the same at 1 m. The characteristic transport time of $HNO_3$ from the snowpack surface to the skin layer (4 mm) is on the order of 1 s, which is much shorter than the temporal resolution of the model (10 min, Table 1). Therefore, the concentration of gaseous $HNO_3$ in the open pore space of the skin layer was assumed to be the same as in the air above the snow. The concentration of gaseous $HNO_3$ was more than 2 orders of magnitude higher in the summer than in autumn/ early winter (Fig. 3b).

Continuous meteorological observation and snow science are carried out at Dome C under the 'Routine Meteorological Observations' of the Concordia Project by the Italian National Antarctic Research Programme, PNRA, and the French Polar Institute, IPEV (http://www.climantartide.it). Temperature and humidity were measured at 10 s resolution. Both the temperature and relative humidity were measured at 1.6 m above the snow surface with a platinum resistance thermometer (VAISALA PT100 DTS12) with a precision of $\pm$ 0.13 °C at $-15$ °C, and the humidity sensor (HU-MICAP, VAISALA) had a precision of $\pm$ 2 %. Based on the assumption of a well mixed boundary layer, the RH above the snowpack surface was assumed to be the same as that at 1.6 m. Atmospheric nitrate concentrations and meteorological data used as model input have been linearly interpolated to 10 minute resolution.

## 4.2  Observation at Halley

Halley is at a similar latitude as Dome C but in coastal Antarctica at sea level and with very different geographic features. Halley is on the Brunt Ice Shelf and is close to the Weddell Sea in three directions. Hence the temperature, relative humidity, and concentration of atmospheric aerosol are much larger at Halley than Dome C. The average surface temperature in summer is around $-10$ °C and below $-20$ °C in the winter. Occasionally, the temperature can rise above $0$ °C (surface melt is possible) or drop to $-55$ °C (See Fig. 4a). The annual mean snow accumulation rate at Halley is 480 $kg\,m^{-2}\,yr^{-1}$ (Wolff et al., 2008), about one order of magnitude larger than at Dome C and therefore limiting post-depositional processes relative to Dome C.

Meteorological and chemical data were collected at Halley under the CHABLIS (Chemistry of the Antarctic Boundary Layer and the Interface with Snow) campaign at the Clean Air Sector Laboratory (CASLab), (details in Jones et al., 2008, 2011). The site description and data given in details elsewhere (Jones et al., 2008), below is a brief description. Measurement of atmospheric concentration

of $HNO_3$ were carried out at weekly resolution using annular denuders (URG corporation) mounted 7-8 m above the snow surface with a collection efficiency of 91% (Fig. 4 B). The atmospheric boundary layer is assumed to be well-mixed so that the nitric acid concentration at the snowpack surface would be the same as at 7-8 m. Surface snow (the top 10 to 25 mm) was collected on a daily basis and the samples were analysed using ion chromatography (Fig. 4 B). Bulk concentrations of the major anions and cations were measured, including $Cl^-$, $SO_4^{2-}$ and $NO_3^-$ (Wolff et al., 2008). The concentrations were interpolated to the 10 minutes model resolution.

Other meteorological data included 10 minute averages of air temperature by Aspirated PRT, RH by Humidity probe (Vaisala Corp) and wind speed and direction by Propeller vane. All sensors were at 1 m above the snow surface. All values were linearly interpolated to the model time step of 10 min.

### 4.3 Other Model Inputs

There are no available pH measurements of the snowpack, therefore, the pH of the DI in Model 1 and the initial pH in Model 2 is assumed to be 5.6 (Udisti et al., 2004, based on the pH of the completely melted samples) at both Dome C and Halley. There are no measurements of SSA recorded during 2009-2010 for skin layer snow. The SSA and effective grain radius in this study are estimated based on observations at Dome C from 2012 to 2015 by Picard et al. (2016) as well as the annual temperature variation, as shown in Fig. A4. To the author's knowledge there are no observations of SSA are available for Halley. Therefore the observations of SSA from Dome C were adjusted taking into account the shorter cold period, which tends to have a larger SSA (Fig. A4, dashed line).

### 4.4 Statistical Analysis

Three-day running means are calculated from all model outputs to better match the time resolution of the snow observations. The performance of the models is assessed by the coefficient of variation of RMSE, $C_v(RMSE)$, as a goodness of fit. The $C_v(RMSE)$ is defined as

$$C_v(RMSE) = \frac{\sqrt{\sum_{t=1}^{n}(obs(t) - model(t))^2 / n}}{\overline{obs}} \tag{20}$$

where $obs(t)$ and $model(t)$ are the observed value and modelled value at time $t$ respectively, $n$ is the number of observations, and $\overline{obs}$ is the observation mean.

## 5 Results

### 5.1 Dome C

The predicted concentration of nitrate in skin layer snow for Model 1 and Model 2 in Dome C (Fig. 5 and Table 2) are discussed by season - Winter to Spring (April - Mid November) and Summer (Mid November - January).

### 5.1.1 Winter to Spring

The average temperature ($\pm 1\sigma$) at Dome C between late autumn to late spring in 2009 is 213.7 ($\pm 7.9$) K (Fig. 3 A), which is below the threshold temperature, $T_o$, for detection of DI (set at 238 K, purple shaded area in Fig. 5 A) for Model 1 and below the eutectic temperature, $T_e$, for a $H_2O$-$HNO_3$ mixture (230 K, yellow shaded area in Fig. 5 B) for Model 2. Therefore, in winter, the skin layer concentration of nitrate is well described by non-equilibrium kinetic surface adsorption and co-condensation coupled to solid-state diffusion within the snow grain in both models. The models combine both processes and agree very well with the observations of nitrate (Fig. 5 A & B) with a $C_v$(RMSE) = 0.73 (Table 2). Both models captured the small peak from mid April to early May and another peak from mid to end of August then a steady increase from middle September till the beginning of November, except for the peak in late February.

The results from 'Bock-BC1' and 'Model 1 - BCice' are shown in Fig. 6a. Both the configurations resulted in a very similar trend and variation until mid Sept. Despite the 'Model 1 - BCice' approach yielding a larger $C_v$(RMSE) = 0.65 compared to the "Bock-BC1' approach $C_v$(RMSE) = 0.52, (Table. 2), the 'Model 1 - BCice' approach captures the temporal pattern from mid September till early November but not in the 'Bock-BC1' approach.

### 5.1.2 Summer

The average temperature ($\pm 1\sigma$) from late spring to early autumn is 240.0 ($\pm 5.0$) K (Fig. 3a) and the dominant process determining the snow nitrate concentration are solvation into the DI coupled with solid state diffusion in Model 1 and partitioning of nitrate to the liquid micropockets in Model 2.

Model 1 captures some trends observed in early spring and during the summer period, including the decrease in concentration of nitrate from the beginning of February, the rise between mid and late November, and the sharp increase in mid December (Fig. 5a). It also reproduced the steep decrease in concentration at the beginning of 2010 (Fig. 5a) . However, Model 1 (with $T_o$ = 238 K) did not capture the peak in early February and overestimated the concentration of nitrate by a factor of 1.5-5 in December (Fig. 5 A).

The results from Model 2 agreed reasonably well with the observation in these few months with $C_v$(RMSE) of 0.67. With the contribution from the partitioning of $HNO_3$ in the micropockets, the features in early February and the peaks between November and mid December were captured (Fig. 5 B). The model underestimates the the nitrate concentration from mid December until January 2010 by a factor of 3. During the summer period, the partitioning into the micropockets contributed $\sim 75\%$ to the total $NO_3^-$ concentration.

## 5.2  Halley

Model results for Model 1 and Model 2 in Halley (Fig. 7 and Table. 3) are presented by the season - Late Autumn to Winter (April - Mid September) and Spring to Early Autumn (Mid September - February).

### 5.2.1  Late Autumn to Winter

The mean temperature ($\pm 1\sigma$) during this period at Halley is 244.72($\pm 7.7$) K (Fig. 4a). During this period, the temperature was mostly above the threshold temperature ($T_o = 238$ K, purple shaded area in Fig. 7 A) used in Model 1 but below the eutectic temperature for a $H_2O$-NaCl mixture (252 K, yellow shaded area in Fig. 7 B ) used in at Halley in Model 2. Therefore, the main process controlling the concentration of $NO_3^-$ in Model 1 is solvation into the DI whereas in Model 2 the main controlling processes are the combination of non-equilibrium adsorption and co-condensation coupled with solid-state diffusion. Performance of Model 1 was poor ($C_v(\text{RMSE}) = 27.78$), overestimating the concentration of $NO_3^-$ by two orders of magnitude (Fig. 7 A). However, some of the trends were reproduced during this cold period such as the two small peaks in mid April and early May, and the rise in mid September (Fig. 7 A).

The modelled results from Model 2 ($C_v(\text{RMSE}) = 1.08$) were a much closer match to the observations compared to Model 1. It captured the first peak in mid April and the small peak in beginning of September (Fig. 7 B). However, it did not reproduce the peak in mid August and underestimated the $NO_3^-$ concentration for the majority of the time.

The results from 'Bock-BC1' and 'Model 1 - BCice' are shown in Fig. 6b. Similar to the Dome C site, the modelled results from both approaches are very similar in value and temporal variations and both the configurations failed to reproduce the peak in mid August.

### 5.2.2  Spring to Early Autumn

Similar to the winter months, Model 1 overestimated the bulk $NO_3^-$ concentration at Halley by an order of magnitude and failed to capture any of the variability (Fig. 7 A) with $C_v(\text{RMSE}) = 89.28$. Model 2, however, reproduced some features during the warmer months, such as the peak in late September followed by a steady rise in October, the spikes in mid December, beginning of and mid January and also the peak and trough in late January (Fig. 7 B). The partitioning to the micropockets contributed $\sim 80\%$ of the total $NO_3^-$ concentration during this period. Results from Model 2 are within the same order of magnitude compared to the observations ($C_v(\text{RMSE}) = 0.65$, Table. 3).

## 6  Discussion

The results from both Model 1 and 2 show that the bulk $NO_3^-$ concentration in surface snow can be reasonably well described by non-equilibrium adsorption and co-condensation coupled with solid-

560 state diffusion during autumn to spring at Dome C and in winter at Halley, i.e. when it is cold and the solar irradiance is small. In the summer months, the combination of warmer temperatures and a larger range of diurnal temperature causes the 'Air-Ice' only processes to no longer provide an accurate prediction. The concentration of $NO_3^-$ in the surface snow, during the warmer months, is mainly determined by solvation into DI in Model 1 or partitioning into micropockets in Model 2.

Overall, the results from Model 1 match reasonably well with the year-round observations at Dome C ($C_v$(RMSE) = 1.34). However, for Halley, Model 1 overestimated the concentration by two order of magnitude ($C_v$(RMSE) = 89.28). On the other hand, results from Model 2 agree well for both study sites year-round ($C_v$(RMSE) = 0.84 for both Dome C and Halley). The mismatch between the models and observations can be separated into 2 categories - data limitations and model 570 configurations, and will be discussed below.

The temporal resolution of the concentration of atmospheric nitrate at both study sites was roughly 5 to 10 days, therefore, any substantial changes in the atmospheric input within a short time scale might be missed and consequently the relative changes in concentration of nitrate in snow might not be observed. Secondly, the vertical snow pit profile of $NO_3^-$ at Dome C (and sites with a low 575 accumulation rate) tended to have a maximum concentration of $NO_3^-$ at the surface of the snowpack (Röthlisberger et al., 2000), especially during the summer period, and the concentration of $NO_3^-$ decreases sharply with the depth in the snowpack. The skin layer is the most responsive layer of snow to the changes in the concentration of $HNO_3$ in the atmosphere above. The snow samples from Dome C were collected carefully from the top $4\pm2$ mm while the snow samples from Halley 580 were collected from the top 25 mm. It is possible that the snow $NO_3^-$ concentrations measured at Halley may be 'diluted' from deeper snow, with a smaller nitrate concentration than the surface layer, leading to a positive model bias.

Thirdly, atmospheric nitrate can be found in the particulate forms of $NO_3^-$, i.e. associated with $Na^+$, $Ca^{2+}$ or $Mg^{2+}$ (Beine et al., 2003). An increase in sea salt aerosol concentration can shift 585 gaseous $HNO_3$ to particle-phase (i.e. $NaNO_3$, Dasgupta et al., 2007), and therefore, decreases the ratio of gaseous $HNO_3$ and the total atmospheric nitrate. At Dome C, the atmospheric sea salt aerosol concentration in late winter or early spring can be up to a factor of 4 larger than the annual mean ($\sim$ 5 ng m$^{-3}$, Legrand et al., 2016) due to the large sea ice extend (Jourdain et al., 2008). Therefore, using the total measured atmospheric nitrate as gaseous $HNO_3$ for constraining the models might 590 lead to an overestimate of [$NO_3^-$] in snow at Dome C, especially in early summer. At the coastal site of Halley, there is a strong influence from sea salt aerosol with corresponding larger concentration of nitrate containing aerosol, especially in spring time that the monthly mean p$-NO_3^-$ mixing ratio is $\sim$ 4.6 pptv (Rankin et al., 2003; Jones et al., 2011). Therefore, neglecting the dry deposition of nitrate aerosols might underestimate the concentration of nitrate in the surface snow in spring time. 595 The concentration of p$-NO_3^-$ (data not show here, see Jones et al., 2008 for more information) is typically 2.6 and 3.0 times higher than the concentration of nitric acid in winter and summer,

respectively, but was up to 8.3 times higher in spring during 2004-2005 at Halley. This might explain the underestimation of concentration of nitrate in surface snow in winter and spring at Halley.

Lastly, no detailed information is available on timing and amount of snowfall events for the time periods in question at both study sites. Single snowfall events can increase the nitrate concentration in surface snow by up to a factor of 4 above the background (Wolff et al., 2008). The contribution of snow nitrate from fresh precipitation may be less important at low accumulation sites, such as Dome C compared to sites with large snow accumulation like Halley. Wolff et al. (2008) reports that the large concentration of $NO_3^-$ recorded from mid until end of August was due to new snowfall, which explains why both models failed to reproduce the peak. In the following sections, various processes included in Model 1 and 2 will be discussed.

### 6.1 Kinetic 'Model 1-BCice' Approach vs Equilibrium 'Bock-BC1' Approach

The 'Model 1-BCice' approach defines the snow grain boundary concentration of $NO_3^-$ by non-equilibrium, kinetic surface adsorption while the 'Bock-BC1' approach after Bock et al. (2016) defines the concentration of the outermost shell of the snow grain by thermodynamic equilibrium ice solubility. Both approaches describe the interaction between air and ice, therefore, only results from the winter period are compared. For both sites, the 'Model 1-BCice' and 'Bock-BC1' approach resulted in very similar trends except the peak in late October at Dome C (Fig. 6, Table 2 & 3), of which the 'Model 1-BCice' approach managed to reproduce but not the 'Bock-BC1' approach.

The peak of snow nitrate in late October at Dome C corresponds to an increase in atmospheric $HNO_3$ (Fig. 3 B). The grain surface concentration of the 'Bock-BC1' approach is a function of the partial pressure of $HNO_3$ with an exponent of 1/2.3 (Eq. 19), while the concentration of the grain boundary defined by the 'Model 1- BCice' approach is linearly related to the concentration of atmospheric nitrate (Eq. 8). Therefore, the 'Model 1- BCice' approach is more responsive to any changes in the atmospheric nitrate concentration compared to the 'Bock-BC1' approach. Other advantages of the former approach are, 1) dynamic characteristics of the grain surface due to changing temperature gradients are taken into consideration; 2) applicability even for sites with high accumulation rates where the skin layer is buried by subsequent snowfall before reaching equilibrium.

At Halley, in winter, the concentrations of $NO_3^-$ are underestimated by both approaches (Fig. 6 and Table 3). There are 2 possible explanations. First, the SSA values used may be underestimated and lead to an underestimation of adsorption or dissolution in the outermost shell of the snow grain, further field observations are required to verify this. Secondly, due to higher temperatures at Halley compared to Dome C, other processes might be involved in controlling the snow surface concentration of $NO_3^-$, such as snowfall (not included in the models) or partitioning into liquid micropockets in Model 2 (discussed in Sect. 6.4).

### 6.2 Co-Condensation - 'Air-Ice' Interaction

The process of co-condensation/sublimation is considered as part of the 'Air-Ice' interaction in both Models 1 and 2. It is driven by the difference in water vapour density across the skin layer snow and the overlying atmosphere. The water vapour density gradient depends exponentially on the temperature gradient. At Dome C the temperature is extremely low and relatively dry, especially in winter, and therefore it is not surprising that only 2% of the grain surface concentration of $NO_3^-$ is from co-condensation during winter and spring (Fig. 6 A, difference between the light and dark blue line). In contrast, at Halley, where winter is warmer and it is relatively humid, $\sim$21% of the grain surface concentration is contributed by co-condensation during winter (Fig. 6 B, difference between the light and dark blue line). As shown in Table 3, the $C_v(RMSE)$ decreased slightly in winter after including co-condensation as part of the 'Air-Ice' interaction. In the summer, the dominant process in Model 1 is solvation into the DI (See Sect. 6.3) while in Model 2 the dominant process is partitioning into the micropockets (See Sect. 6.4), hence the contribution from co-condensation to the skin nitrate concentration is insignificant.

There are a few possible sources of uncertainties in the calculation of co-condensation/sublimation processes. For example, the macro-scale gradients of water vapour pressure (across a few mm) were used instead of micro-scale gradients (across a few μm) and there were no precise measurements of skin layer snow density. Uncertainty in the density would lead to uncertainty in the modelled skin layer snow temperature (Eq. 12). Despite the potential errors in the calculation of co-condensation, the large $NO_3^-$ concentrations in the skin layer in the summer are unlikely to be driven by co-condensation. An unrealistically large average rate of volume change, $\frac{dV}{dt}$, of 130 and 118 $\mu m^{-3} s^{-1}$, equivalent to an average grain volume increases of 170% and 135% per day, would be required for Dome C and Halley respectively if the large concentration of $NO_3^-$ in summer was contributed by co-condensation (Eq. 9 & 10). Assuming the RH in the open pore space of the skin layer snow to be 100% and RH of the overlying atmosphere to be the same as measured at 1 m above snowpack, a macro-temperature gradient as high as $2.7 \times 10^3$ $K\,m^{-1}$ would be required across the top 4 mm of the snowpack to match the large concentration of bulk $NO_3^-$ in the summer at Dome C and in an average temperature gradient of 500 $K\,m^{-1}$ would be required across the top 10 mm of the snowpack at Halley. Therefore, the required temperature gradients are 1- 2 orders of magnitude larger than indicated by observations or modelled result (Frey et al., 2013, and as listed in Sect. 3.1.1).

### 6.3 Disordered Interface - Model 1 (T > $T_o$ = 238 K)

In Model 1, the air-snow grain interface is described as 'Air-DI' at $T>$ 238 K. Therefore, at Dome C, the 'Air-DI' regime applies only during summer months due to the extremely cold temperatures in winter, whereas, at Halley most of the time the interface is considered as 'Air-DI'. Model

1 simulations suggest that an 'Air-DI' interfaceat $T > 238$ K leads to an overestimation of nitrate concentration in early December at Dome C and all year round at Halley. The poor performance of Model 1 at Halley and at Dome C in summer is attributed to uncertainties in physical and chemical properties of the DI.

Here, $T_o$ has been set to the lower end of the temperature range, where the onset of a DI is observed in the lab on a pure ice surface (Bartels-Rausch et al., 2014). The exact DI onset temperature is uncertain as reported values vary with different experimental setups, probing and sample preparation techniques (Bartels-Rausch et al., 2014). Furthermore, for a mixture of $H_2O$ and impurities it has been observed that already at 100 K below the melting point a small fraction of water molecules

begins to leave the outermost crystalline layer of the ice with the number of mobile molecules increasing with temperature (Conde et al., 2008). When the temperature is within 10 K below the melting point, molecules might even begin to leave the deeper crystalline layer. Therefore, the chosen threshold temperature, $T_o$, might be substantially different from what would be found in natural snow or it might not be representative to be used as the threshold all year-round. The Model 1 sen-

sitivities to $T_o$ are evaluated below (Sect. 6.5), and suggest that goodness of fit improves slightly at Dome C with a 2 K increase, but shows no significant improvement at Halley (Table 4).

    The onset and thickness of the DI not only depend on temperature, but also the speciation and concentration of impurities present within the snow grain (McNeill et al., 2012; Dash et al., 2006). Different impurities have different impacts on the hydrogen bonding network at the ice surface and

hence have a different impact on the thickness of the DI, leading in general to a thickening compared to pure ice (Bartels-Rausch et al., 2014). However no accepted model parameterisation is available. In this model imposing a seasonal cycle of SSA and therefore grain size causes the thickness of the outermost model shell to vary between 1.5 µm in summer and 0.5 µm in winter (Sect. 3.1), relatively large values and potentially contributing to the positive bias in Model 1. This is explained as follows:

the bulk concentration of $NO_3^-$ is calculated as the sum of number of molecules in each model shell divided by the total volume of the snow grain (Eq. 17). At $T > T_o$ the outermost model shell is equivalent to a DI and its concentration is determined by Henry's Law (Eq. 13), which is independent of grain size and thus model shell thickness $\Delta r$. However, the absolute number of molecules in each model shell including the DI, increases with $\Delta r$ yielding a larger bulk concentration in summer.

Choosing a thinner outermost model shell may reduce the Model 1 bias at Halley.

    In summary, a combination of potential factors contribute to why Model 1 performs reasonably well at Dome C, but not at Halley: 1) at Dome C the chemical composition of surface snow is relatively simple, dominated by the nitrate anion, which would induce only insignificant changes to the hydrogen bonding network at the DI surface compared to a more complicated snow composition

(Bartels-Rausch et al., 2014) and suggesting that the surface properties of snow at Dome C are likely to be comparable to pure ice; 2) at Halley temperatures occasionally rise above 0 °C potentially causing melting and significant changes in snow grain morphology at the surface especially; 3) as

temperature increases the DI may become more distinct from ice and more isolated from the bulk and may have less or even no interaction with the bulk. This is supported by previous laboratory experiments showing that physical properties, such as extinction coefficient and refractive index, of the ice surface gradually change from the measured value of ice to the measured value for water and the layer of disordered water molecules grows increasingly thicker as temperature approaches the melting point (Huthwelker et al., 2006).

### 6.4 Micro-Liquid Pocket - Model 2 ($T \geq T_e$)

Model 2, which includes non-equilibrium surface adsorption and co-condensation coupled with solid diffusion within the grain and partitioning in liquid micropockets, successfully reproduces the concentration of $NO_3^-$ of the surface snow without any tuning parameters for both Dome C and Halley all year round. This is a crucial outcome as it indicates that Model 2 can be used for predicting the air-snow exchange of nitrate at the surface for a wide range of meteorological and depositional conditions that are typical for the entire Antarctic ice sheet.

The liquid water fraction is a function of the total ionic concentration (Eq. 4). Hence, neglecting the existence of other ions may lead to underestimation of the micropocket volume. The additional liquid would increase the dissolution capacity of $HNO_3$ and hence increase the estimated $NO_3^-$ concentration. As shown in Fig. 7 B, the estimated bulk $NO_3^-$ concentration followed a similar trend as the 'other ions concentration', which is the observed $Cl^-$ concentration. Despite $NO_3^-$ being the major anion in the surface snow in Dome C, other anions, such as $Cl^-$ and $SO_4^{2-}$, were also detected from the same samples (Udisti et al., 2004). Jones et al. (2008) also measured $SO_4^{2-}$ along with $Cl^-$ and $NO_3^-$ from the surface snow samples from Halley. The mismatch between modelled and observed nitrate concentration in the summer can be explained by assuming nitrate to be the only impurity at Dome C, or nitrate and sea salt as the only impurities at Halley. Nevertheless, the underestimation of the $NO_3^-$ concentration due to underestimating the liquid-water content may be compensated or even overwhelmed if atmospheric deposition of other acids such as HCl or $H_2SO_4$ increases, which lowers the pH and reduces the solubility of $HNO_3$ in the micropocket.

Note that the micropockets only exist above the eutectic temperature. For simplification, the eutectic temperature was based on a system containing $H_2O$ and the most abundant solute within surface snow. However, in reality, the presence of other impurities might have an impact on the eutectic temperature. Moreover, the liquid in the micropocket is assumed to behave ideally and, therefore, Henry's coefficient is used to describe the partitioning between air and the micropocket. In reality, there may be some deviation from ideality as the concentration of solutes in the micropocket is likely to be too large to be considered as an ideal dilute solution. The non-ideality should be accounted for in terms of activity coefficient, $\gamma$. At equilibrium, the relationship between a solute $B$ and the solvent can be expressed as follow (Sander, 1999):

$$K_B = \frac{\gamma_B x_B}{P_B} \tag{21}$$

where $P_B$ is the vapour pressure of $B$, $\gamma_B$ is the activity coefficient of $B$ and $x_B$ is the mole fraction of $B$. The value of the activity coefficient approaches unity as the mole fraction of $B$ approaches zero ($\gamma_B \rightarrow 1$ as $x_B \rightarrow 0$) and, under such ideal-dilute condition, the equilibrium constant, $K_B$, is defined as Henry's law coefficient. Values of activity coefficient can be found experimentally. The available parameterisation of activity coefficient of $HNO_3(aq)$, $H^+$ and $NO_3^-$ is only accurate for concentration up to 28 m (Jacobson , 2005). When the molarity is higher than $\sim$4-5 m, depending on the temperature, the activity coefficient of $H^+$ and $NO_3^-$ increases as molarity increase. The concentration of the micropocket is estimated based on the parameterisation by Cho et al. (2002), which predicts a concentration a lot larger than the limit of activity coefficient parameterisation available at present. Hence, it is not possible to quantify the uncertainties caused by assuming the micropocket has ideal-solution behaviour. If the relationship between activity coefficient and molarity extend to molarity larger than 28 m, the activity coefficient will be larger than 1 and hence reduces the value of the equilibrium constant, $K_B$, compared to the Henry's Law coefficient. By means, the assumption of ideal-solution behaviour of micropocket is likely to overestimate the concentration of the micropocket. The activity coefficient of highly concentrated solution is needed to be found by further experimental studies.

## 6.5 Sensitivity Analysis

In order to assess the robustness of the findings presented here they were analysed as a function of model sensitivities to constraints, parameterisations and measurement uncertainties. Parameters were varied one at a time by the given range while keeping all others constraints and parameterisation the same (Table. 4, Column 1). The coefficient of variation, $C_v(RMSE)$, was calculated from each sensitivity test (Table. 4) and compared with the $C_v(RMSE)$ of the 'Control', which uses the observed values and parameterisation listed in Sect. 4 and Table. A1.

Both Model 1 and 2 are sensitive to the concentration of $HNO_3$ in the air and the concentration of $NO_3^-$ in snow. Reducing concentration of $HNO_3$ in the atmosphere by 20% or increasing the concentration of $NO_3^-$ in snow by 20% improves the performance of both models. This supports the suggestion that the atmospheric nitrate observed at Dome C only represents the upper limit of nitric acid and it is likely to lead to an overestimation of the concentration of nitrate in snow (Sect. 6) while at Halley, the skin layer snow might well be 'diluted' by snow sample from the deeper layer (Sect. 6).

Both models are sensitive to the value of SSA as a smaller SSA implies a smaller surface area per unit volume of snow, and hence, less surface sites available for adsorption per unit volume of snow. It has a more notable impact in Model 1 and in the winter, when the grain boundary processes play an important role for the overall snow nitrate concentration due to the cold temperature. A similar explanation applies the value of the maximum number of adsorption site, $N_{max}$. However, varying

the accommodation coefficient, $\alpha$, by $\pm$ 10% does not have a significant impact on the performance
of the models (Table 4).

Model 1 is very sensitive to the threshold temperature, $T_o$. At Dome C, the best match (lowest
$C_v$(RMSE)) between modelled and observation is with a threshold temperature 2 K larger than
the control $T_o$ = 238 K. However, increasing $T_o$ to 242 K worsens the model performance further
(Fig. 5A, Green line & Table 4). When a larger value of $T_o$ is used, a larger in-snow temperature is
required to assume the interface is 'Air-DI'. Nitrate concentrations at the grain boundary, $U(R_{\text{eff}})$,
have a much larger value when the interface between air and grain boundary is defined as 'Air-
DI' (Eq. 13) than when it is defined as 'Air-Ice' (Eq. 7). At Dome C, a larger value of $T_o$ may
have reduced the overestimation in late November due to a larger fraction of time falling below the
threshold but compromised the good fit from mid December onward and yield a higher $C_v$(RMSE).
At Halley, despite the improvement in $C_v$(RMSE) when a higher temperature threshold was used,
the modelled [$NO_3^-$] is still an order of magnitude larger than the observation (Fig. 7 B).

Model 1 is not sensitive to the pH of the DI. Even though the effective Henry's law coefficient
increases by an order of magnitude when pH increases from 5 to 6.5 (Fig. A3), the $C_v$(RMSE)
remains the same. This behaviour can be explained by the combination of the kinetic approach
and slow diffusion rate of nitrate in ice that the rate of change in the grain boundary concentration
remains small even if the boundary concentration increases.

Model 2 is sensitive to the eutectic temperature, $T_e$, but not as much as for $T_o$ in Model 1. Increas-
ing $T_e$ in Model 2, only improves the performance at Dome C but not Halley. Higher $T_e$ implies that
a larger temperature is required for the co-existence of liquid micropockets. For Dome C, increasing
$T_e$ by 2-4 K reduces the overestimation in November without compromising the results from mid
December onwards, as the average temperature during that period was higher than $T_e = 234$K.

## 7 Conclusions

Two surface physical models were developed from existing sphysical parameterisations and labo-
ratory data to estimate the bulk concentration of $NO_3^-$ in the skin layer of snow constrained by
observed atmospheric nitrate concentrations, temperature and humidity.

Model 1 assumes that below a threshold temperature, $T_o$, the outermost shell of a snow grain is
pure ice, whereas above $T_o$ the outermost shell is a disordered interface (DI). The nitrate concen-
tration at the air-ice boundary is defined by non-equilibrium kinetic adsorption and co-condensation
whereas the nitrate concentration at the air-DI boundary is defined by non-equilibrium kinetics based
on Henry's Law. A non-equilibrium grain boundary is assumed as the partial pressure of $HNO_3$ is
low in Antarctica and a large temperature gradient is expected across the snowpack surface which
leads to redistribution of water molecule at the grain surface. The boundary of the grain is also
assumed to be interacting with the bulk so that the mass transport is driven by the concentration

difference between the outermost model shell and centre of the grain and constrained by solid-state
diffusion. The uncertainties of Model 1 are 1) the temperature threshold, $T_o$, that defines the emergence of the 'air-DI' interface; 2) the partitioning coefficient of $HNO_3$ into the DI; 3) the interaction between the grain boundary and the bulk ice; and 4) the thickness of the DI and its dependence on temperature and ion concentration. Assuming too large of a DI thickness results in an overestimate of the bulk concentration of nitrate. The modelled skin layer concentration of $NO_3^-$ from Model 1 agreed reasonably well with observations at Dome C but overestimated observations by an order of magnitude at the relatively warmer Halley site. The poor performance of Model 1 at the warmer site suggests that as the temperature increases the disordered interface is becoming more liquid-like and disconnected from the bulk ice.

Model 2 assumes that below melting temperature, $T_m$, the outermost model shell of a snow grain is pure ice and above eutectic temperature, $T_e$, liquid exists in grooves at grain boundaries and triple junctions as micropockets. The nitrate concentration at the air-ice boundary is defined by non-equilibrium kinetic adsorption and co-condensation. The boundary of the grain is also assumed to be interacting with the bulk and the mass transport between the surface and centre of the grain is driven by solid-state diffusion. The nitrate concentration of the liquid micropocket is defined by Henry's law. Equilibrium between air and liquid in micropockets is assumed because the liquid micropocket volume is small and $HNO_3$ is very soluble in water implying fast interfacial mass transport. The main uncertainties in Model 2 are three-fold, 1) dry and wet deposition of atmospheric nitrate are currently not included in the model, but lead to episodic increases of $NO_3^-$ in surface snow; 2) the liquid micropocket is likely not an ideal solution due to high ionic strength, which is likely leading to overestimation of solvation; and 3) third the eutectic temperature of natural snow is assumed to be that of a single major ion - water system but may be different because snow ionic composition is complex. However, Model 2 reproduced the skin layer concentration of $NO_3^-$ with good agreement at both Dome C and Halley without any tuning parameters.

Both Model 1 and 2 suggest that in the winter the interaction of nitrate between the air and skin layer snow can be described as a combination of non-equilibrium kinetic ice surface adsorption and co-condensation coupled with solid diffusion within the grain. Only Model 2 provides a reasonable estimate at both sites year-round, that suggests in the summer, the major interface between snow grain and surrounding air is still air-ice, but it is the equilibrium solvation into liquid micropockets that dominates the exchange of nitrate between air and snow. Despite the simplified parameterisation of processes in Model 2, it provided a new parameterisation to describe the interaction of nitrate between air and snow as 'air-ice' with a liquid formed by impurities present as micropockets as suggested by Domine et al. (2013) instead of an 'air-DI' interface assumed by most models developed previously. Moreover, the non-equilibrium boundary between air and snow grain allows the models to work at sites with high rate of accumulation that the snow layer might be buried by fresh snowfall before reaching equilibrium.

Additional modelling studies, e.g. including uptake of other chemical species and aerosols such as $H_2SO_4$ and nitrate aerosols, backed up by field observations from other locations with various meteorological conditions as well as laboratory studies on the eutectic point of a multi-ion - $H_2O$ system, uptake coefficient at a higher temperature, are needed to improve the performance of Model 2. Moreover, the models presented here are describing the exchange between air and the skin layer of snowpack as the uptake processes are much quicker than the photochemical loss, and therefore, can be modelled by 'physical-only' processes. Atmospheric nitrate can reach deeper than the skin layer via wind pumping and temperature gradient, however, the nitric acid concentration in snow interstitial air is expected to be small compared to the overlying atmosphere due to the high uptake of nitrate near the surface of the snowpack. A smaller concentration of $HNO_3$ in snow interstitial air implies a smaller uptake in deeper snow, and hence the photochemical loss cannot be assumed to be negligible in deeper snow. Therefore, a more complex multi-layer model including both physical and chemical processes is required to reproduce the nitrate concentration in deeper snow and to implement in regional and global atmospheric chemistry models.

*Acknowledgements.* HGC is funded by the Natural Environment Research Council through Doctoral Studentship NE/L501633/1. We are thankful to our colleagues (Anna Jones, Neil Brough and Xin Yang) for helpful discussion.

## 8 Notation

| Symbol | Description | units |
|---|---|---|
| $\alpha$ | Accommodation coefficient | dimensionless |
| $A_{ice}$ | Surface area of ice per unit volume of snowpack | $m^2\,m_{snowpack}^{-3}$ |
| $C_v(RMSE)$ | Coefficient of variation | N/A |
| DI | Disordered Interface | N/A |
| $D_v$ | Water vapour diffusivity | $m^2\,s^{-1}$ |
| $D_s'$ | Gas-phase diffusivity in snow | $m^2\,s^{-1}$ |
| $\Delta H_f^0$ | enthalpy of fusion | $J\,mol^{-1}$ |
| $[HNO_{3\,(ads)}]$ | Nitric acid concentration contributed by surface adsorption | $molecule\,m^{-3}$ |
| $[HNO_{3\,(cc)}]$ | Nitric acid concentration contributed by co-condensation | $molecule\,m^{-3}$ |
| $[HNO_{3\,(DI)}]$ | Nitric acid concentration in the DI | $molecule\,m^{-3}$ |
| $[HNO_{3\,(g)}]$ | Nitric acid concentration in gas-phase | $molecule\,m^{-3}$ |
| $[HNO_{3\,(ice)}]$ | Nitric acid concentration in solid ice | $molecule\,m^{-3}$ |
| $[HNO_{3\,(surf)}]$ | Nitric acid concentration on surface of grain | $molecule\,m^{-3}$ |
| $[Ion_{tot,bulk}]$ | Total ionic concentration in melted snow sample | $molecule\,m^{-3}$ |
| $k_{ads}$ | Adsorption coefficient on ice | $m^3\,molecule^{-1}\,s^{-1}$ |
| $k_{des}$ | Desorption coefficient on ice | $s^{-1}$ |
| $k_{H^{cc}}$ | Henry's Law coefficient | dimensionless |
| $k_H^{eff}$ | Effective Henry's Law coefficient | dimensionless |
| $k_{diff}$ | Diffusivity in ice | $m^2\,s^{-1}$ |
| $k_w$ | Thermal conductivity of snowpack | $W\,m^{-1}\,K^{-1}$ |
| $K_a$ | Acid dissociation constant | $molecule\,m^{-3}$ |
| $K_{eq}$ | Equilibrium constant for Langmuir adsorption | $m^3\,molecule^{-1}$ |
| $m_{\bar{H}_2O}$ | Molecular mass of water | $kg\,mol^{-1}$ |
| $N_{max}$ | Maximum number of adsorption sites | $molecule\,m^{-2}$ |
| $[NO_{3\,(bulk)}^-]$ | Bulk nitrate concentration | $molecule\,m^{-3}$ |
| $\phi_{H_2O}$ | Liquid water fraction | dimensionless |
| $\Phi_{bulk}^{aq}$ | Fraction of the total amount of solute in aqueous phase | dimensionless |
| $R_{eff}$ | Effective radius of snow grain derived from SSA data | m |
| $R$ | Ideal gas constant | $J\,mol^{-1}\,K^{-1}$ |
| $\rho_{ice}$ | Density of ice | $kg\,m^{-3}$ |
| $\rho_v$ | Water vapour density | $kg\,m^{-3}$ |
| $[S]$ | Number of available surface sites per unit volume of air | $molecule\,m_{air}^{-3}$ |
| SSA | Specific surface area | $m^2\,kg^{-1}$ |
| $T_e$ | Eutectic temperature | K |
| $T_f$ | Reference temperature | K |
| $T_o$ | Threshold temperature in Model 1 | K |
| $\theta$ | Fraction of surface sites being occupied | dimensionless |
| $\overline{v}$ | Mean molecular speed | $m\,s^{-1}$ |
| $V_{air}$ | Volume of air per unit volume of snowpack | $m_{air}^3\,m_{snowpack}^{-3}$ |
| $V_{grain}$ | Volume of a snow grain | $m^3$ |

**Table 1.** Characteristic times associated with gas-phase diffusion, mass transport and uptake of gas into ice grain

| Process | Expression | Order of magnitude, s |
|---|---|---|
| Interfacial mass transport to a liquid surface[i] | $\frac{4R_{\mathrm{eff}}}{3\overline{v}\alpha_{aq}}$ | $10^{-7}$ |
| Gas-phase diffusion to the surface of a spherical droplet[ii] | $\frac{R_{\mathrm{eff}}^2}{3\,D'_s}$ | $10^{-4}$ |
| Molecular diffusion between snowpack and the atmosphere[iii] | $\frac{z^2}{D'_s}$ | $10^{0}$ |
| Liquid-phase diffusion within a water droplet[iv] | $\frac{4\,R_{\mathrm{eff}}^2}{\pi^2\,k_{\mathrm{diff(aq)}}}$ | $10^{0}$ |
| Surface adsorption on ice[v] | $\frac{1}{k_{\mathrm{des}}}$ | $10^{3}$ |
| Solid-state diffusion within a snow grain[vi] | $\frac{4\,R_{\mathrm{eff}}^2}{\pi^2\,k_{\mathrm{diff}}}$ | $10^{6}$ |
| Photolysis at a snowpack surface[vii] | $\frac{1}{J}$ | $> 10^{7}$ |

[i] Sander (1999), with an effective radius, $R_{\mathrm{eff}}$ = 70 μm, and accommodation coefficient on liquid water, $\alpha_{aq} = 7.5 \times 10^{-5} \exp(2100/\mathrm{Temp})$ (Ammann et al., 2013). [ii] Sander (1999), with an effective molecular diffusivity, $D'_s = D_a/\tau_g$, where the tortuosity, $\tau_g = 2$ and molecular diffusivity in free air at 296 K, $D_a(296\mathrm{K}) = 87$ Torr cm$^2$ s$^{-1}$ (Tang et al., 2014). [iii] Waddington et al. (1996), with a snow layer thickness, $z$ = 4 mm. [iv] Finlayson-Pitts and Jr. (2000), with a diffusion coefficient in liquid water, $k_{\mathrm{diff(aq)}} = 1 \times 10^{-9}$ m$^2$ s$^{-1}$ (Yuan-Hui and Gregory, 1974) . [v] Crowley et al. (2010), with an equilibrium constant for Langmuir adsorption, $K_{eq} = 2 \times 10^{-16}$ m$^3$ molecule$^{-1}$ and adsorption coefficient, $k_{\mathrm{ads}} = 1.7 \times 10^{-19}$ m$^3$ molecule$^{-1}$ s$^{-1}$. [vi] Finlayson-Pitts and Jr. (2000), with a diffusion coefficient in ice, $k_{\mathrm{diff}} = 6 \times 10^{-16}$ m$^2$ s$^{-1}$ (Thibert et al., 1998). [vii] Finlayson-Pitts and Jr. (2000), with a surface $NO_3^-$ photolysis rate coefficient, $J$, = $10^{-7}$ s$^{-1}$ (Thomas et al., 2011).

**Table 2.** Summary of model performance at Dome C based on the coefficient of variation of RMSE, $C_{\mathrm{v}}(\mathrm{RMSE})$

| Model description | Short name | Whole year DOY 30 - 385 | Winter-Spring DOY 90 - 318 | Summer DOY 319 - 385 |
|---|---|---|---|---|
| Surface Adsorption & Solid Diffusion | Model1-BCice | - | 0.65 | - |
| Ice Solubility & Solid Diffusion | Bock-BC1 | - | 0.52 | - |
| Surface Adsorption-Co Condensation/DI Solvation & Solid Diffusion | | | | |
| No threshold (no Solvation) | Model 1-none | 1.07 | 0.65 | 0.88 |
| $T_o$= 238 K | Model 1-238K | 1.34 | 0.73 | 1.11 |
| Surface Adsorption-Co Condensation & Solid Diffusion + micropocket | Model 2 | 0.84 | 0.73 | 0.67 |

**Table 3.** Summary of model performance at Halley based on the coefficient of variation of RMSE, $C_v(RMSE)$

| Model description | Short name | Whole year DOY 87 - 406 | Winter DOY 90 - 257 | Spring -Early Autumn DOY 258 - 406 |
|---|---|---|---|---|
| Surface Adsorption & Solid Diffusion | Model1-BCice | - | 1.13 | - |
| Ice Solubility & Solid Diffusion | Bock-BC1 | - | 1.12 | - |
| Surface Adsorption-Co Condensation/DI Solvation & Solid Diffusion | | | | |
| No threshold (no Solvation) | Model 1-none | 1.06 | 1.06 | 0.95 |
| $T_o$= 238 K | Model 1-238K | 89.28 | 27.78 | 87.15 |
| Surface Adsorption-Co Condensation & Solid Diffusion + micropocket | Model 2 | 0.84 | 1.08 | 0.65 |

**Table 4.** Sensitivity test for Model 1 and 2 based on the coefficient of variation of RMSE, $C_v(\mathrm{RMSE})$, the metric was used to measure a goodness of fit. Note that column one is not fitted to the observation and the values are only varying to show the sensitivity of the models against inputs and parameterisation.

| Parameter | | Model 1 Dome C Whole year | Model 1 Dome C Winter-Spring | Model 1 Dome C Summer | Model 1 Halley Whole year | Model 1 Halley Winter | Model 1 Halley Spring-Summer | Model 2 Dome C Whole year | Model 2 Dome C Winter-Spring | Model 2 Dome C Summer | Model 2 Halley Whole year | Model 2 Halley Winter | Model 2 Halley Spring-Summer |
|---|---|---|---|---|---|---|---|---|---|---|---|---|---|
| Control | | 1.34 | 0.73 | 1.11 | 89.28 | 27.78 | 87.15 | 0.84 | 0.73 | 0.67 | 0.84 | 1.08 | 0.65 |
| $[HNO_3]$ | $-20\%$ | 0.98 | 0.60 | 0.81 | 71.19 | 22.12 | 69.5 | 0.80 | 0.62 | 0.64 | 0.77 | 1.10 | 0.56 |
| | $+20\%$ | 1.73 | 0.90 | 1.45 | 107.36 | 33.43 | 104.80 | 0.95 | 0.88 | 0.76 | 0.92 | 1.07 | 0.75 |
| SSA | $-10\%$ | 1.06 | 0.63 | 0.88 | 79.35 | 24.79 | 77.46 | 0.83 | 0.67 | 0.67 | 0.84 | 1.10 | 0.65 |
| | $+10\%$ | 1.63 | 0.84 | 1.36 | 99.22 | 30.75 | 96.86 | 0.84 | 0.78 | 0.67 | 0.83 | 1.07 | 0.65 |
| $\alpha$ | $-10\%$ | 1.34 | 0.73 | 1.11 | 79.35 | 24.78 | 77.46 | 0.83 | 0.73 | 0.67 | 0.83 | 1.08 | 0.65 |
| | $+10\%$ | 1.34 | 0.73 | 1.11 | 79.35 | 24.80 | 77.46 | 0.83 | 0.73 | 0.67 | 0.83 | 1.08 | 0.65 |
| $N_{max}$ | $-10\%$ | 1.32 | 0.67 | 1.10 | 89.27 | 27.77 | 87.15 | 0.83 | 0.69 | 0.67 | 0.84 | 1.09 | 0.65 |
| | $+10\%$ | 1.36 | 0.80 | 1.13 | 89.29 | 27.78 | 87.15 | 0.84 | 0.77 | 0.67 | 0.84 | 1.07 | 0.65 |
| $T_o$ (Model 1) or | -2 K | 3.53 | 0.91 | 3.00 | 90.45 | 42.54 | 87.31 | 0.95 | 0.92 | 0.75 | 0.85 | 1.12 | 0.65 |
| $T_e$ (Model 2) | +2 K | 0.50 | 0.64 | 0.36 | 67.49 | 25.33 | 65.62 | 0.73 | 0.65 | 0.58 | 0.86 | 1.07 | 0.65 |
| | +4 K | 0.61 | 0.65 | 0.47 | 50.76 | 23.86 | 49.00 | 0.72 | 0.65 | 0.57 | 0.88 | 1.06 | 0.67 |
| pH | -0.4 | 1.34 | 0.73 | 1.11 | 89.28 | 27.78 | 87.15 | - | - | - | - | - | - |
| | +0.4 | 1.34 | 0.73 | 1.11 | 89.28 | 27.78 | 87.15 | - | - | - | - | - | - |
| | +0.8 | 1.34 | 0.73 | 1.11 | 89.28 | 27.78 | 87.15 | - | - | - | - | - | - |
| $[NO_3^-]$ | $-20\%$ | 1.85 | 0.98 | 1.54 | 111.87 | 34.84 | 109.2 | 0.99 | 0.96 | 0.79 | 1.09 | 1.08 | 0.93 |
| | $+20\%$ | 1.04 | 0.61 | 0.86 | 74.22 | 23.07 | 72.45 | 0.80 | 0.64 | 0.64 | 0.74 | 1.10 | 0.51 |

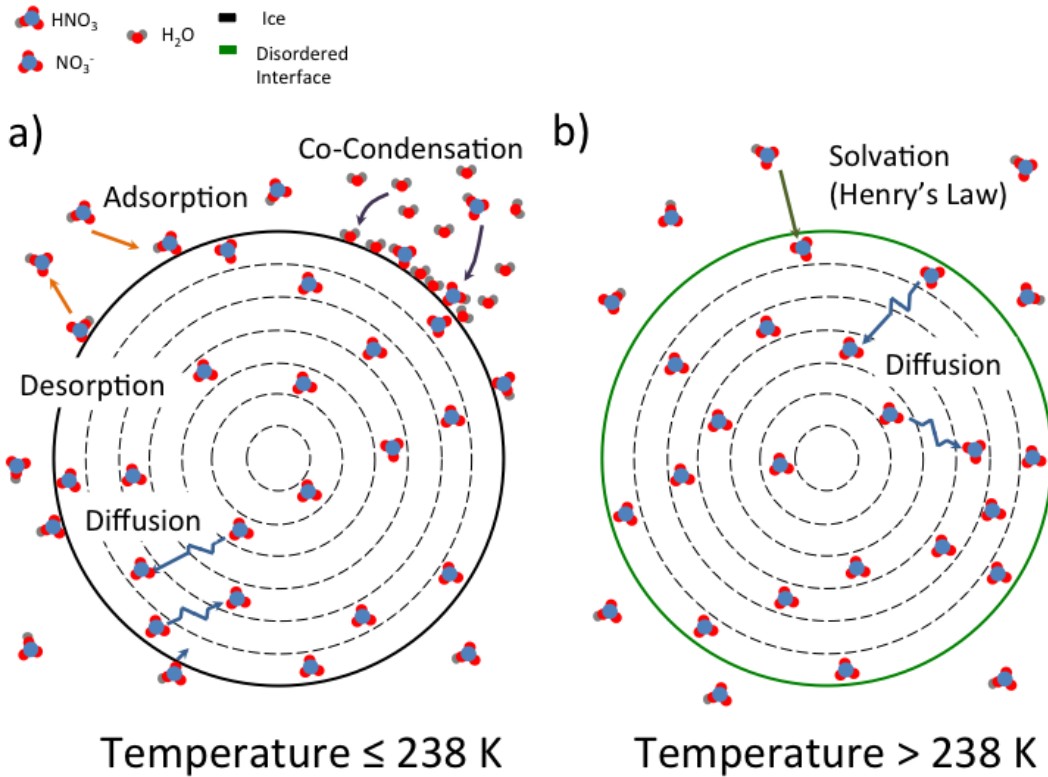

**Figure 1.** Schematic of Model 1. a) At T $\leq$ 238 K the concentration of $NO_3^-$ at the boundary of the snow grain is determined by Air-Ice processes, i.e. non-equilibrium adsorption and co-condensation. b) At T > 238 K the concentration of $NO_3^-$ at the boundary of the snow grain is determined by Air-DI processes, i.e. non-equilibrium solvation into DI.

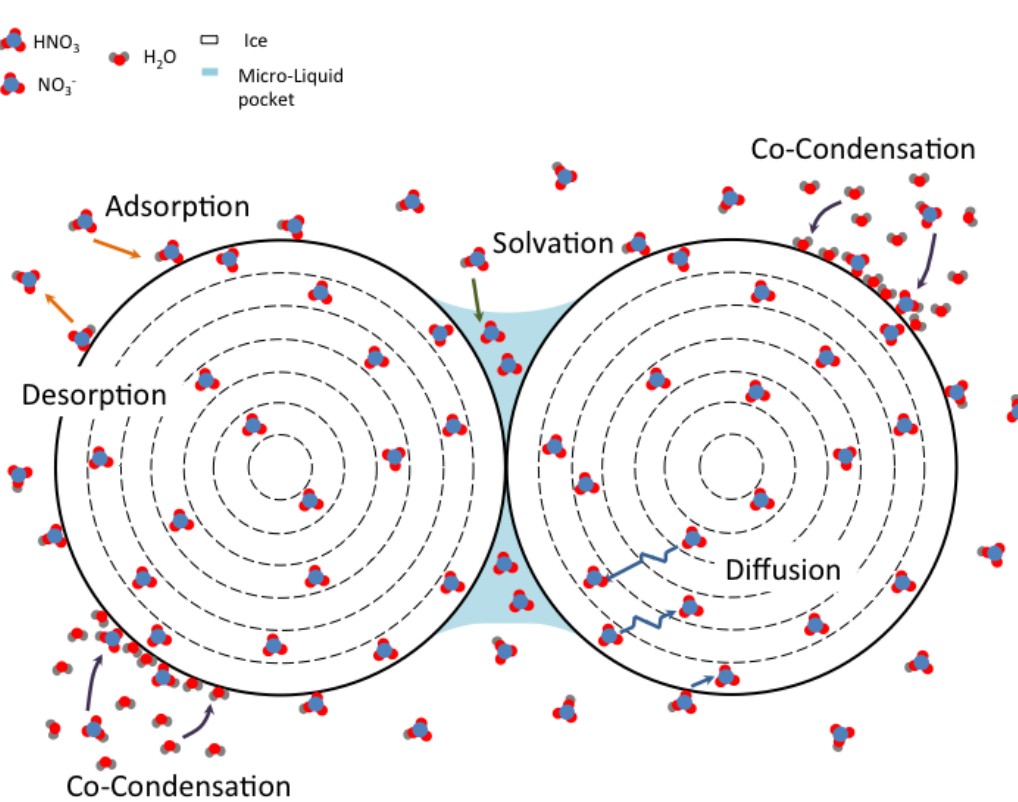

**Figure 2.** Schematic of Model 2. At T $< T_m$, the concentration of $NO_3^-$ at the boundary of the snow grain is determined by Air-Ice processes, i.e. non-equilibrium adsorption and co-condensation. At T $\geq T_e$, liquid is assumed to co-exist with ice and the liquid fraction is in the form of micropockets that are located at grain boundaries and triple junctions (Domine et al., 2013).

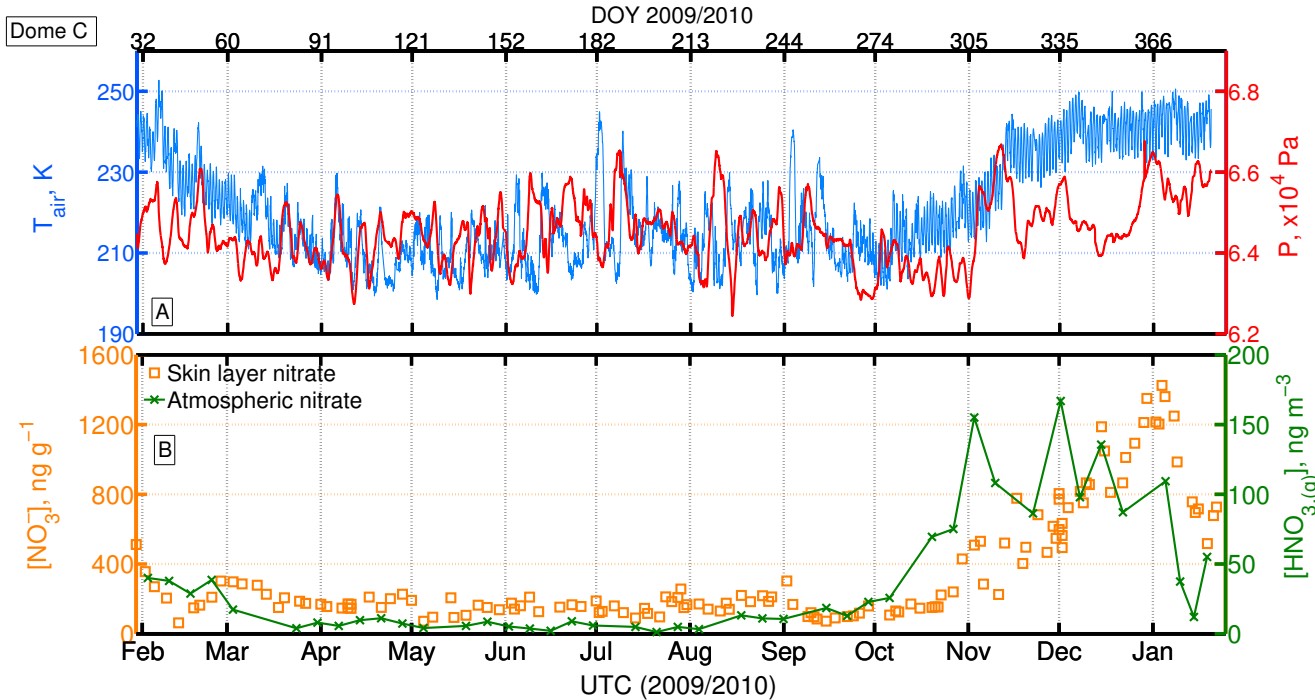

**Figure 3.** Atmospheric and snow observations from Dome C from Erbland et al. (2013)). (**A**) Air temperature (blue, left axis) and atmospheric pressure (red, right axis). (**B**) $NO_3^-$ in the snow skin layer (i.e. top $4 \pm 2$ mm, orange square, left axis) and atmospheric $NO_3^-$, i.e. sum of the atmospheric particulate $NO_3^-$ and $HNO_3$ (green, right axis).

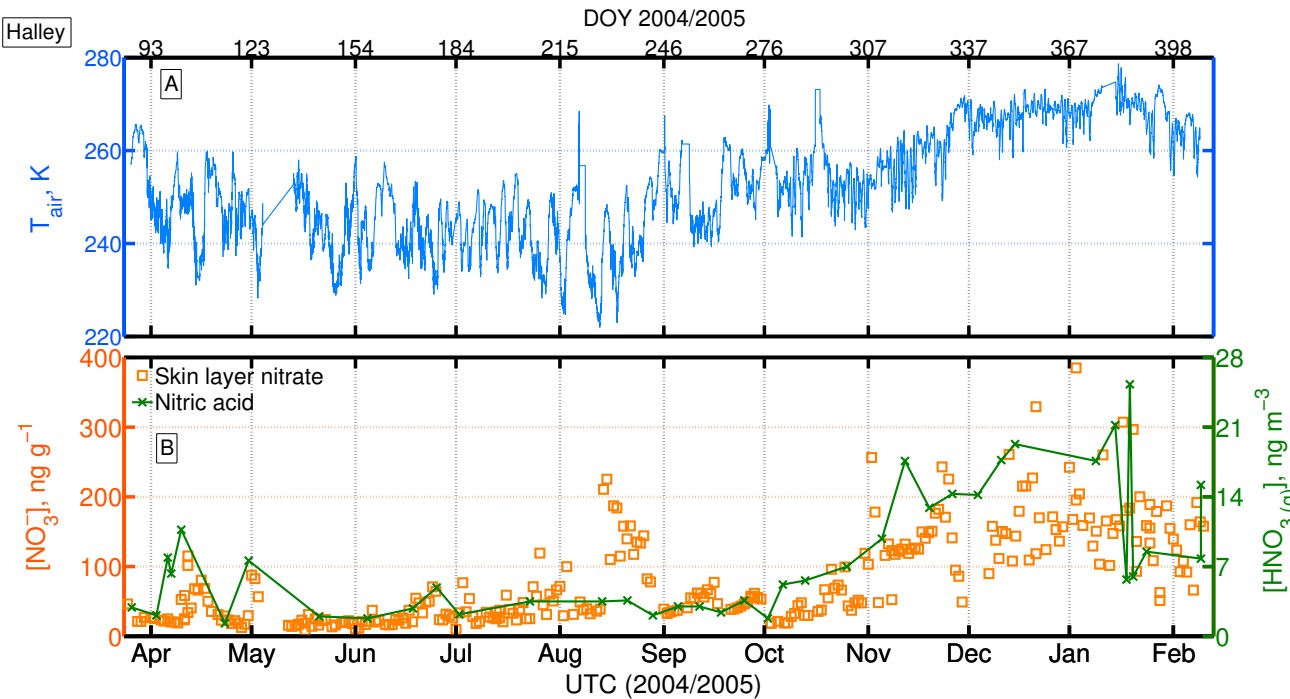

**Figure 4.** Atmospheric and snow observations at Halley between $27^{th}$ March 2004 and $9^{th}$ February 2005 from Jones et al. (2008). (**A**) Air temperature. (**B**) $NO_3^-$ in the surface snow (i.e. top $10 \pm 15$ mm, orange square, left axis) and gas-phase $HNO_3$ (green, right axis).

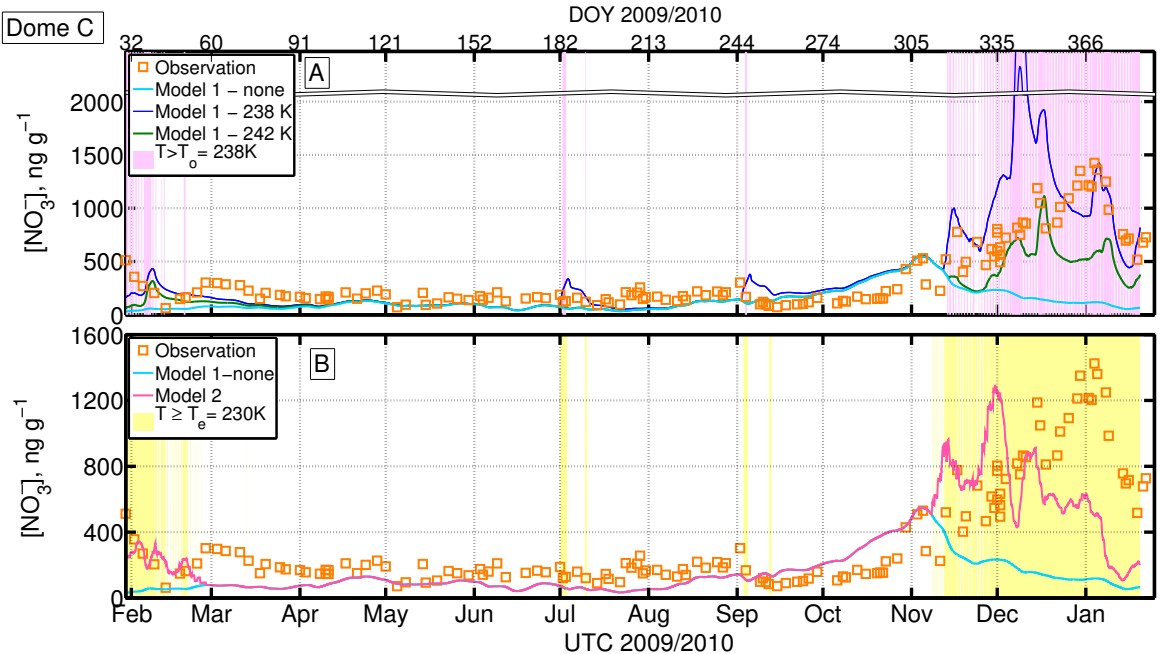

**Figure 5.** (**A**) Model 1 output of Dome C skin layer snow concentration of $NO_3^-$. At $T < T_o$ the interface between air and snow grain is assumed to be ice ('Air-Ice') and the $NO_3^-$ concentration is determined by a combination of non-equilibrium adsorption on ice and co-condensation coupled with solid-state diffusion. At $T > T_o$, the interface between air and snow grain is assumed to be a DI ('Air-DI'), i.e. the $NO_3^-$ concentration is determined by a combination of non-equilibrium solvation into the DI coupled with solid-state diffusion. Note that the y-axis is broken between 2000-3500 $ng\,g^{-1}$. Orange squares: observation; Light blue: Model 1 with $T_o > T_m$, i.e. only air-ice interaction; Dark blue: Model 1 with $T_o = 238$ K; Green: Model 1 with $T_o = 242$ K; Purple shaded area indicate times when $T > T_o = 238$ K; (**B**) Model 2 output of Dome C skin layer snow $NO_3^-$ concentration. The major interface between air and snow is assumed to be ice ('Air-Ice') at $T < T_m$ and the $NO_3^-$ concentration in ice is determined by a combination of non-equilibrium adsorption and co-condensation coupled with solid-state diffusion. Above $T > T_e = 230$ K, liquid co-exists with ice in the form of micropocket. The partition between air and micropocket is determined by Henry's law. Orange squares: observation; Light blue: Model 1 with $T_o > T_m$, i.e. air-ice only interaction; Pink: 'Model 2' - air-ice interaction plus micro-liquidpockets; Yellow shaded area indicates times when $T > T_e = 230$ K ($T_e$ for HNO$_3$-H$_2$O system).

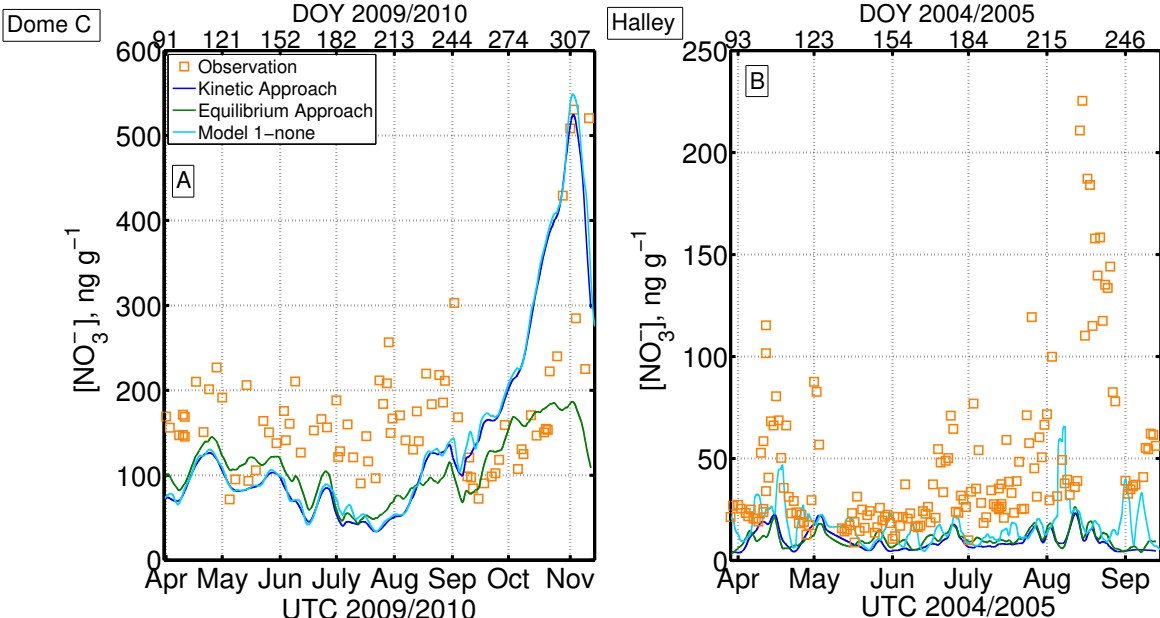

**Figure 6.** Comparison of the 'Kinetic' approach (this work, in dark blue) with the 'Equilibrium' approach (similar to Bock et al. (2016), in green), and the contribution from the co-condensation process (Results from Model 1- none, in light blue) in winter. The 'Kinetic' approach describes the air-snow interaction of nitrate as non-equilibrium kinetic surface adsorption coupled with solid diffusion inside the grain whereas the 'Equilibrium' approach describes the interaction as equilibrium solubility coupled with solid diffusion inside the grain. The 'Model 1-none' describes the interaction as co-condensation plus non-equilibrium kinetic surface adsorption coupled with solid diffusion within the grain. (**A**) Results at Dome C. (**B**) Results at Halley.

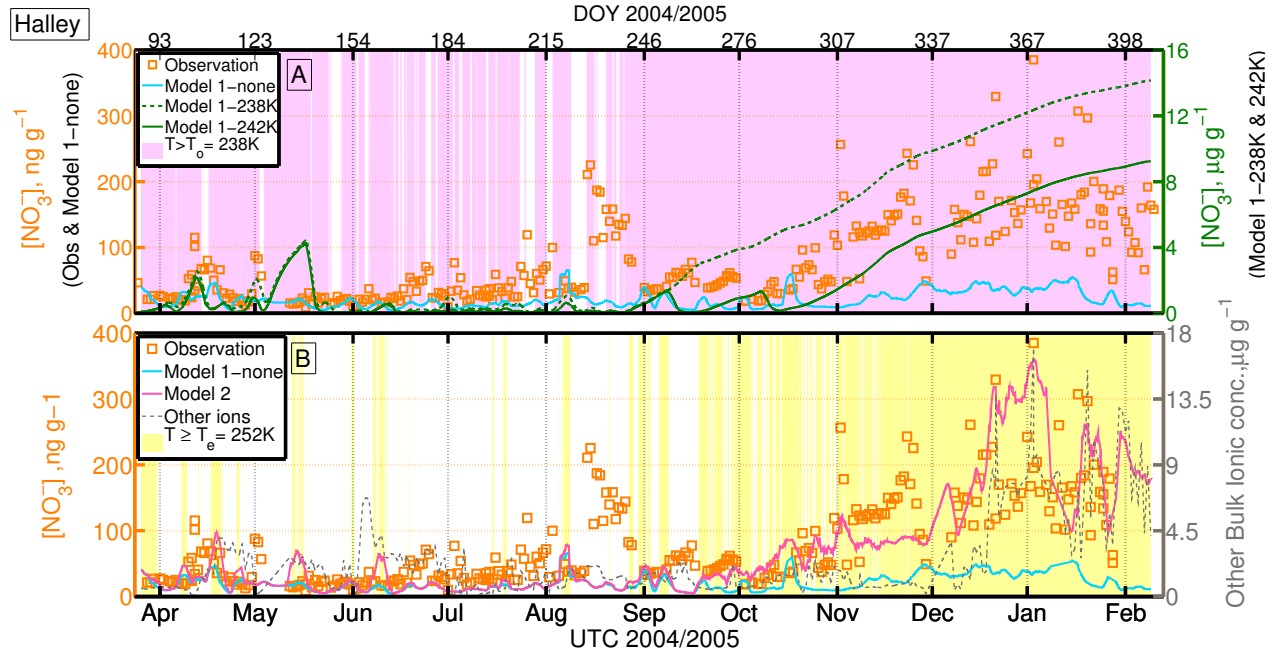

**Figure 7.** (**A**) Model 1 output of Halley skin layer snow concentration of $NO_3^-$. At $T < T_o$ the interface between air and snow grain is assumed to be ice ('Air-Ice') and the $NO_3^-$ concentration is determined by a combination of non-equilibrium adsorption on ice and co-condensation coupled with solid-state diffusion. At $T > T_o$, the interface between air and snow grain is assumed to be a DI ('Air-DI'), i.e. the $NO_3^-$ concentration is determined by a combination of non-equilibrium solvation into the DI coupled with solid-state diffusion. Orange squares: observation; Light blue: Model 1 with $T_o > T_m$, i.e. only air-ice interaction; Dark blue: Model 1 with $T_o = 238$ K; Green: Model 1 with $T_o = 242$ K; Purple shaded area indicate times when $T > T_o = 238$ K; (**B**) Model 2 output of Dome C skin layer snow $NO_3^-$ concentration. The major interface between air and snow is assumed to be ice ('Air-Ice') at $T < T_m$ and the $NO_3^-$ concentration in ice is determined by a combination of non-equilibrium adsorption and co-condensation coupled with solid-state diffusion. Above $T > T_e =252$ K, liquid co-exists with ice in the form of micropocket. The partition between air and micropocket is determined by Henry's law. Orange squares: observation; Light blue: Model 1 with $T_o > T_m$, i.e. air-ice only interaction; Pink: 'Model 2' - air-ice interaction plus micro-liquidpockets; Grey (Right axis) - measured bulk concentration of other ions, where other ions refers to the sum of $[Na^+]$ and $[Cl^-]$; Yellow shaded area indicates times when $T > T_e = 252$ K ($T_e$ for $NaCl-H_2O$ system)

## Appendix A:  Parameterisation

**Table A1.** Parameterisation for $HNO_3$

| Symbol | Parameter | Value/Parameterisation | units | Reference |
|---|---|---|---|---|
| $\alpha_0$ | Accommodation coefficient at reference temperature | $3 \times 10^{-3}$ [i] | Dimensionless | Hudson et al. (2002) |
| $k_{\text{diff}}$ | Diffusion coefficient of nitrate in ice | $1.37 \times 10^{-2610/T}$ | $\text{cm}^2\,\text{s}^{-1}$ | Thibert et al. (1998) |
| $k_w$ | Thermal conductivity of snowpack | $k_w = k_{ice}\left(\dfrac{\rho}{\rho_{ice}}\right)^{2-0.5\frac{\rho}{\rho_{ice}}}$ | $\text{Wm}^{-1}\text{K}^{-1}$ | Hutterli et al. (2003) therein |
| $k_{ice}$ | Thermal conductivity of ice | $k_{ice} = 9.828\exp(-0.00577T)$ | $\text{Wm}^{-1}\text{K}^{-1}$ | Hutterli et al. (2003) therein |
| $\Delta_{\text{sol}}H$ | Enthalpy of solution at standard temperature | $-72.3$ | $\text{kJ mol}^{-1}$ | Brimblecombe and Clegg (1988) |
| $\Delta_{\text{obs}}H$ | Enthalpy of uptake | $-44$ | $\text{kJ mol}^{-1}$ | Thomas et al. (2011) |
| $k_{\text{H}}^0$ | Henry constant at 298 K | $1.7 \times 10^5$ [ii] | $\text{M atm}^{-1}$ | Brimblecombe and Clegg (1988) |
| $N_{max}$ | Maximum adsorption site | $2.7 \times 10^{18}$ | $\text{molecules m}^{-2}$ | Crowley et al. (2010) |
| $\overline{v}$ | Mean molecular speed | $\sqrt{\dfrac{8RT}{M_m \pi}}$ [iii] | $\text{m s}^{-1}$ | Sander (1999) |
| $X_{\text{HNO}_3}^0$ | Molar fraction of $HNO_3$ in ice | $X_{\text{HNO}_3}^0 = 2.37 \times 10^{-12}\exp\left(\frac{3532.2}{T}\right)P_{\text{HNO}_3}^{1/2.3}$ | $\text{mol mol}^{-1}$ | Thibert et al. (1998) |
| $K_{eq}$ | Langmuir adsorption equilibrium constant | $-8.2 \times 10^{-18}\text{T} + 2.01 \times 10^{-15}$ | $\text{m}^3\text{molecule}^{-1}$ | Burkholder and Wine (2015) |
| $D_v$ | Water vapour diffusivity | $D_v = 2.11 \times 10^{-5}\left(\frac{T}{T_o}\right)^{1.94}\frac{P_o}{P}$ | $\text{m}^2\,\text{s}^{-1}$ | Pruppacher and Klett (1997) |

[i] Temperature dependent accommodation coefficient, $\alpha = \dfrac{\exp\{\ln(\frac{\alpha_0}{1-\alpha_0})[-\frac{\Delta_{\text{obs}}H}{R}\left(\frac{1}{T}-\frac{1}{T_f}\right)]\}}{1-\exp\{\ln(\frac{\alpha_0}{1-\alpha_0})[-\frac{\Delta_{\text{obs}}H}{R}\left(\frac{1}{T}-\frac{1}{T_f}\right)]\}}$, (Thomas et al., 2011), where $R$ is the molar gas constant, $T$ is the temperature, $T_f$ is the reference temperature (220 K) and $\alpha_0$ is the from Hudson et al. (2002) at 220 K

[ii] Temperature dependent dimensionless Henry's Law coefficient, $k_{\text{H}}^{\text{cc}} = k_{\text{H}}^0 \times RT \times \exp\left(\frac{-\Delta_{\text{sol}}H}{R}\left(\frac{1}{T}-\frac{1}{T^{\ominus}}\right)\right)$, where $T^{\ominus}$ is the standard temperature (298 K).

[iii] $M_m$ is the molar mass of the gas.

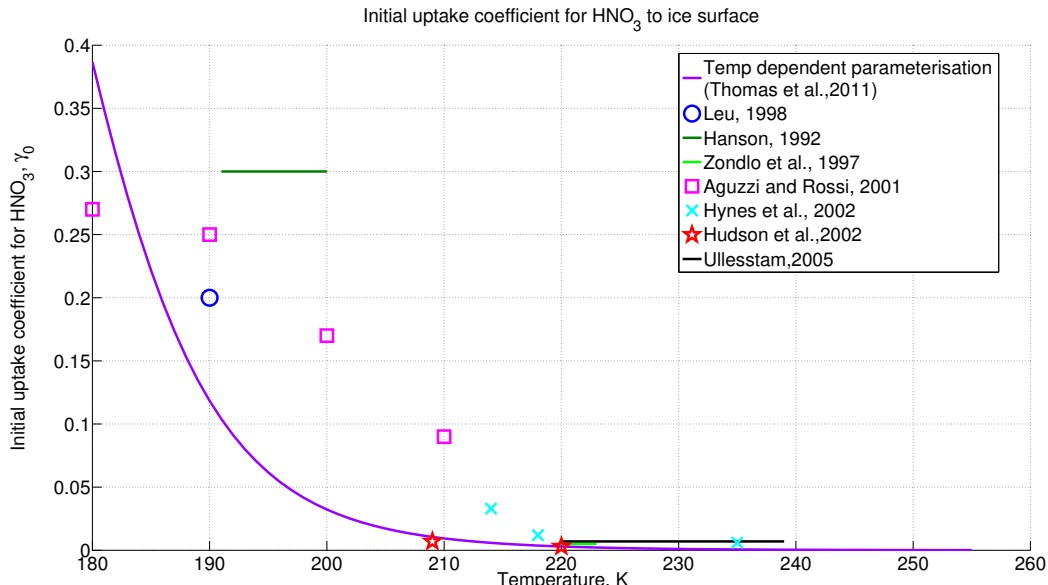

**Figure A1.** Initial uptake coefficient for $HNO_3$ as a function of temperature obtained from different studies. In this study the parameterisation of $\alpha(T)$ with $\alpha_0$ after Hudson et al. (2002) is used (Table A1, solid purple line) and is chosen to give the best representation of the dependency on temperature.

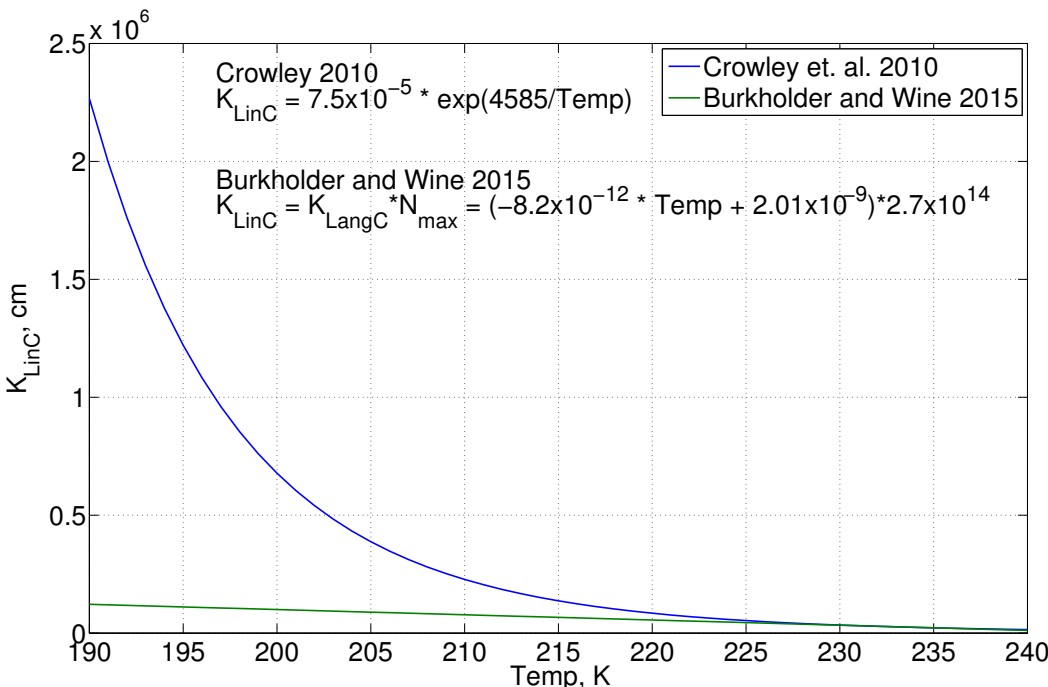

**Figure A2.** Langmuir adsorption equilibrium constant, $K_{\mathrm{LinC}} = K_{eq} \times N_{\max}$. The preferred temperature range for both parameterisation is 214-240 K and within this range the two parameterisations provide a comparable value. The Crowley et al. (2010) parameterisation deviate from the Burkholder and Wine (2015) parameterisation as temperature drop below 214 K due to the exponential temperature term. Here, the parameterisation from Burkholder and Wine (2015) was chosen based on the extreme cold temperature found in our validation sites (minimum winter temperature at Dome C is $\sim$ 199 K, Erbland et al., 2013).

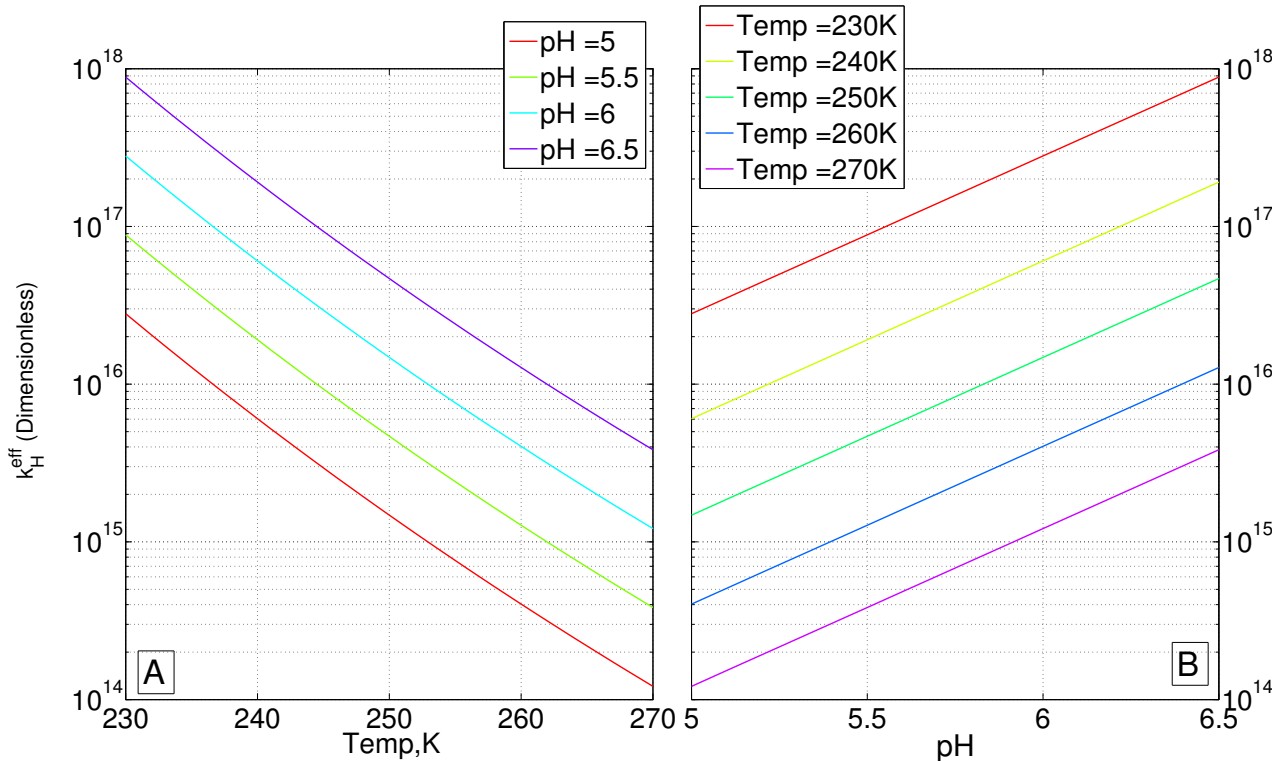

**Figure A3.** The dependence of the effective Henry's Law coefficient, $k_{H\text{eff}}$, of $HNO_3$ on (A) temperature and (B) pH

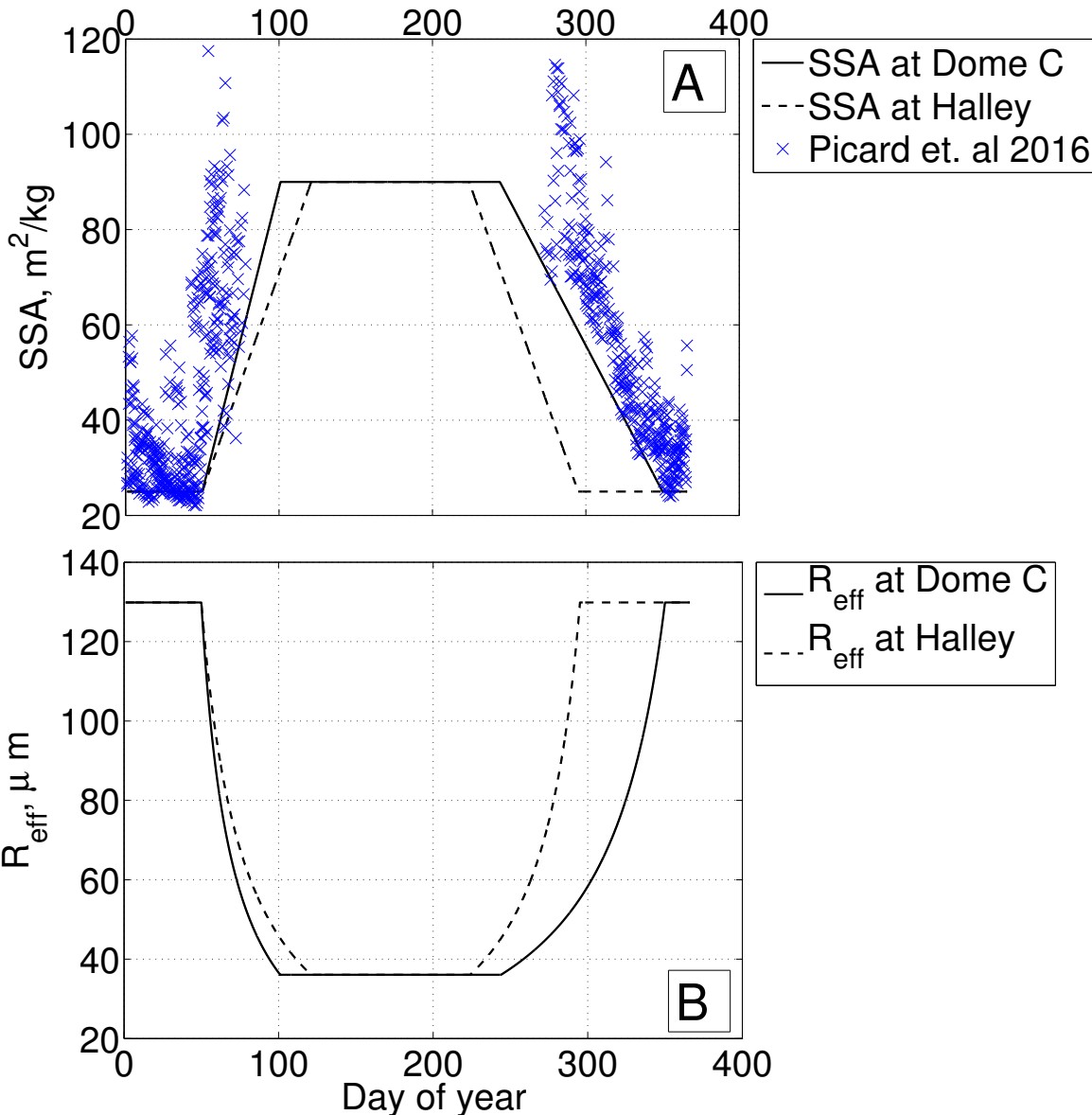

**Figure A4.** (**A**) Year-round estimates of the specific surface area (SSA) of snow at Dome C (−) and Halley (−−) were interpolated from observations at Dome C during 2012-2015 by Picard et al. (2016) (×). The SSA estimates for Halley take into account the shorter cold period compare to Dome C, which tends to have larger SSA. (**B**) Year-round estimates of effective grain radius ($R_{eff}$) at Dome C (−) and Halley (−−) derived from Eq. 6.

## Appendix B:  Derivation for non-equilibrium kinetics

The processes involved in the equilibrium of the gas-phase and the surface of a droplet (Fig. A5): 1) Gas-phase diffusion from far away (> μm) from the droplet to the surface of the droplet, which is likely to be driven by turbulence and molecular diffusion; 2) Interfacial mass transport; and 3) Condensed-phase diffusion and chemical reactions;

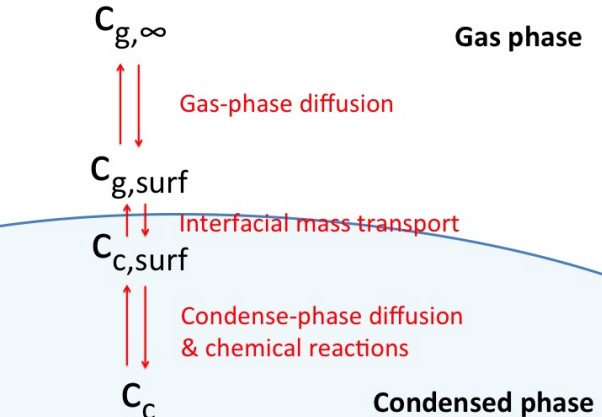

**Figure A5.** Processes involve in the equilibrium between gas-phase and condensed-phase, where $c_{g,\infty}$ is the gas-phase concentration in the snow interstitial air far away from the droplet, $c_{g,\mathrm{surf}}$ is the gas-phase concentration at the surface (outside the droplet), $c_{c,\mathrm{surf}}$ is the condensed-phase concentration at the surface (inside the droplet) and $c_c$ is the average condensed-phase concentration.

Transport of gas-phase species from the snow interstitial air to the surface of the droplet can be described using Fick's law as diffusion flux, $J_g$:

$$J_g = -D_g \frac{dc_g}{dx} \tag{B1}$$

where $D_g$ is the gas-phase diffusivity, and $\frac{dc}{dx}$ is the concentration gradient at the droplet surface that $\frac{dc_g}{dx} = \frac{c_{g,\infty} - c_{g,\mathrm{surf}}}{R_{\mathrm{eff}}}$ with $R_{\mathrm{eff}}$ as the radius of the droplet. The concentration change in the condensed-phase can be expressed as

$$\frac{dc_c}{dt} = \frac{A\,J_g}{V} = -\frac{A}{V}\frac{D_g}{R_{\mathrm{eff}}}(c_{g,\infty} - c_{g,\mathrm{surf}}) \tag{B2}$$

where $A$ is the surface area of the droplet and $V$ is the volume of the droplet. The first-order rate coefficient for the gas-phase diffusion process can be defined as $k_{dg} = \frac{A}{V}\frac{D_g}{R_{\mathrm{eff}}}$ (Sander, 1999). For an example, a liquid droplet with a radius $R_{\mathrm{eff}}$ the gas-phase diffusion rate coefficient $k_{dg} = \frac{3D_g}{R_{\mathrm{eff}}^2}$.

The interfacial mass transport from gas-phase to condensed-phase can be expressed in terms of accommodation coefficient, $\alpha$. The flux through the phase boundary into the droplet, $J_b^{in}$, is defined as:

$$J_b^{in} = \frac{\alpha \bar{v}}{4} c_{g,\mathrm{surf}} \tag{B3}$$

where the subscript $b$ stands for 'boundary' and $\bar{v}$ is the mean molecular velocity. The opposite flux, $J_b^{out}$, through the phase boundary out of the droplet can be expressed in the similar form as Eq. B3 that $J_b^{out} = \frac{\alpha_a \bar{v}_c}{4} c_{a,\mathrm{surf}}$, where $\bar{v}_c$ is the mean molecular velocity in condensed-phase and $\alpha_c$ is the condensed-phase accommodation coefficient. The net flux through the grain boundary, $J_b$, is the difference between the in and out flux.

$$J_b = J_b^{in} - J_b^{out} = \frac{\alpha\bar{v}}{4}\left(\frac{c_{c,\mathrm{surf}}}{K} - c_{g,\mathrm{surf}}\right) \tag{B4}$$

where $K$ is the equilibrium constant, of which $K = c_{c,\mathrm{surf}}^{eq}/c_{g,\mathrm{surf}}^{eq}$. For example, for a gas-aqueous interface, the ratio of aqueous-phase concentration to gas-phase concentration at equilibrium can be described as $c_{a,\mathrm{surf}}^{eq}/c_{g,\mathrm{surf}}^{eq} = k_H^{cc}$, where $c_{a,\mathrm{surf}}$ is the aqueous-phase concentration at the surface and $k_H^{cc}$ is the Henry's constant. The concentration change in the condensed phase due to interfacial mass transport can be expressed as:

$$\frac{dc_c}{dt} = -\frac{A\,J_b}{V} = \frac{A}{V}\frac{\alpha\bar{v}}{4}\left(c_{g,\mathrm{surf}} - \frac{c_{c,\mathrm{surf}}}{K}\right) \tag{B5}$$

The first-order rate coefficient for the interfacial mass transport, $k_b$, to a droplet with a radius $R_{\mathrm{eff}}$ can then be defined as $k_b = \frac{3\alpha\bar{v}}{4}R_{\mathrm{eff}}$. By assuming the fluxes of gas-phase diffusion, $J_g$, is equal to the interfacial mass transport, $J_b$, the rate of change of concentration in the condensed phase can be expressed as

$$\frac{dc_c}{dt} = \frac{A}{V}\left(\frac{R_{\mathrm{eff}}}{D_g} + \frac{4}{\bar{v}\alpha}\right)^{-1}\left[c_{g,\infty} - \frac{c_{c,\mathrm{surf}}}{K}\right] \tag{B6}$$

the term ' $\frac{A}{V}\left(\frac{R_{\mathrm{eff}}}{D_g} + \frac{4}{\bar{v}\alpha}\right)^{-1}$ ' is often referred as the mass transfer coefficient, $k_{mt}$, for a chemical species transfer from air to liquid/solid. The mass transfer coefficient for chemical into a spherical droplet with radius $R_{\mathrm{eff}}$ is $k_{mt} = (\frac{r^2}{3D_g} + \frac{4R_{\mathrm{eff}}}{3\bar{v}\alpha})^{-1}$ and if the surface of the droplet is described as DI then the concentration at the grain surface, $c_{c,\mathrm{surf}} = [\mathrm{HNO_{3,DI}}]$.

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
