# Peer review of "Modeling the Physical Multi-Phase Interactions of HNO3 Between Snow and Air on the Antarctic Plateau (Dome C) and coast (Halley)"

_Atmospheric Chemistry and Physics, 2016_

## Referee Comment (RC1) · Anonymous Referee #1 · 13 Jan 2017

In this work, Chan et al examined the physical exchange of HNO3 between the surface snow and the air above in the Antarctic Plateau, using a newly developed model framework. Snowpack is a complex multiphase system and to this day the air-snow interactions remain poorly understood (Domine et al 2013; etc). A better understanding of the physical and chemical processes in snow is in crucial need in order to evaluate the broader impacts of snowpack on the overlying atmosphere. Overall, I find the topic is of interest to the community and this study merits publication in Atmospheric Chemistry and Physics, once the following points have been addressed.

My main concern about this work is the lack of a clear description of the mass transfer between the atmospheric nitrate above snowpack and the nitrate concentration in the

surface skin layer of snow.

(1) Atmospheric nitrate (gaseous HNO3 + particulate nitrate) is assumed to be dominated by gaseous HNO3 (which is supported by previous studies). In this work, the physical exchange of gaseous HNO3 in the snow interstitial air (SIA) and the snow grains is described explicitly by different models. However, the mass exchange of HNO3 between the SIA and air above snow (where the atmospheric nitrate is measured) is missing. Mass exchange between the SIA and air above snow is largely controlled by processes such as turbulent transport and wind pumping. How these processes would affect the bulk nitrate in the skin layer of snow needs to be clearly addressed.

(2) Model 2 incorporates the micro-liquid pocket. This topic is of great interest since the brine formed by impurities may not cover the entire grain surface due to limited wettability at cold temperatures. However, instantaneous air/micro-liquid pocket equilibrium is assumed. This seems to be oversimplified. For highly soluble species such as HNO3 in liquid water (effective Henry's law constant > 10ˆ14 M atmˆ-1, Fig 1), interfacial transport or even gas diffusion (in this case, gas diffusion in the SIA) may well become the rate limiting steps. The timescale of the SIA/micropocket equilibrium needs to be examined before assuming equilibrium.

(3) From the model point of view, Model 2 does not really specify or depend on the location of liquid water, i.e. whether the liquid water is covering the whole/part of the grain surface as a thin layer, or is located in grooves at grain boundaries and tripe junctions. It appears mathematically that, in Eq(4) + Eq(17), only the liquid water content matters while the location of liquid water does not.

(4) The authors claim that the physical exchange models are based on "first principles" (what exactly are first principles btw) and hence without requiring any tuning parameters. This seems not true: some parameters involved in the models are still somewhat adjustable and/or lack direct observational support, such as max number of adsorption
* * *
Interactive
comment

sites, threshold temperature T0, microscopic H2O density gradient, eutectic temperature, etc.

(5) Comparison between models and measurements needs to be discussed in the context of their respective uncertainty ranges. What is the measurement uncertainty of skin layer nitrate concentration? What is the model uncertainty propagated from the inputs and parameters?

(6) The quality of English could use some polish.

In addition, the authors claim that the photochemistry of snow nitrate can be ignored due to slow photolysis in this region. Well, "what goes up must come down" and vice versa. What processes are then responsible for the loss of snow nitrate? And what is driving the seasonal variations of snow nitrate in this region? Snow nitrate can't cannnot always accumulate. This is perhaps not the main focus of this work, but the fact that only snow nitrate sources are included in the model may be quite confusing.

Specific comments:

Page 3, Line 61: the characteristic times of surface adsorption and solid-state diffusion for HNO3... please provide more details (either literature or point to later sections).

Page 3, Line 83: define skin layer. What is the thickness of this skin layer in the model and why this value is chosen? Or is it simply the layer in which the bulk ion concentrations are measured? Since the model is limited to the skin layer, it seems that there is no exchange between the skin layer and the deeper snow. However, previous studies(e.g. Traversi et al 2014) indicated that temperature gradients and wind pumping exist in the snowpack, therefore nitrate could be mobilized by physical processes reaching much deeper than the "skin layer" in this model (a few mm?).

Page 5, Line 141: the solid-state diffusivity is introduced here, and hence characteristic time can be calculated. Please compare to other processes, e.g. surface accommodation and gas-phase diffusion.

[Figure]

Page 6, Line 178: what is the size of snow grain?

Page 7, Eq 7: both adsorption and co-condensation contribute to surface HNO3. Is co-condensed HNO3 available for desorption? Judging from Eq 6 it seems the answer is yes, yet in Eq 7 it seems co-condensed HNO3 is not included. Also, will the co-condensed HNO3 molecules undergo solid diffusion?

Page 8, Line 248: what is the thickness of this DI covering the entire grain sur-face? Also, Eq 13 describes d[HNO3(DI)]/dt, and there should be another equation for d[HNO3(g)]/dt accordingly. Please provide this. Finally, I may be wrong but shouldn't mass transfer (Eq 13 and d[HNO3(g)]/dt) depend on liquid water content of some sort?

Page 8, Eq 8: this equation describes co-condensation. How about H2O sublimation? Does HNO3 undergo co-sublimation (or whatever the term should be) as well?

Page 10, Line 294: again, for highly soluble species in liquid, interfacial transport or gas diffusion may be limiting (Schwartz, 1986). Please calculate the equilibrium timescale and discuss in the context of other mass transfer processes.

Page 13, Line 399: define winter (and other seasons too). The Northerners would appreciate this.

Page 14, Line 432: "However, Model 1... overestimated concentration by a factor of 1.5-5 in December". Which model 1? With 238 K or 242 K?

Page 15, Line 476: "the combination of larger temperatures and a larger diurnal tem-perature range" this sentence is confusing.

Page 16, Line 493: "it is possible that the snow NO3- concentration measured from Halley might be 'diluted' from deeper snow layer..." then can you extend your model to cover deeper layers, or simply increase the skin layer thickness? Also, as shown in Fig 11, Model 2 underestimated nitrate for the majority of the time (Line 458-459). If measured snow NO3- was indeed diluted, would this mean the model underestimates even more?

Page 16, Line 497: what do you mean by "fixed by sea salt, ammonium or terrestrial dust"?

Page 16, Line 502: "the increase in sea salt concentration decreases the ratio of concentration of gaseous HNO3 to total atmospheric nitrate". Please provide evidence.

Page 16, Line 503: "A possible explanation for the overestimation of NO3- concentration in both Model 1 and 2 in November at Dome C" this is not a complete sentence.

Page 17, Line 546: "In the summer, other processes are replaced..." this sentence is ill-formed. What are you trying to say?

Page 19, Line 605: there is no purple on Fig 7.

Page 19, Line 628: Again this is only true if gas diffusion and interfacial transport are not limiting. Also, Model 1 output is quite sensitive to T0. How sensitive is Model 2 to the eutectic temperature?

Fig 1: Please include units for the effective Henry's law constant. Also I feel this belongs in the Supplementary Information. The temperature and pH dependencies of effective Henry's law constant, although are important, do not deserve the spot of the very first figure of this particular paper.

Fig 4, Fig 6-11: dates on the bottom axis are difficult to read, i.e. it is hard to identify "early Feb" or "early May", ... Please set date tick labels to the first day of each month. If not enough space, rotate 90 degrees.

Fig 5: figure legend very unclear. What exactly are the scatter points? And what are "Head 1 1213", "Head 2, 1213", ...?

Fig 7 & Fig 8: I think these two figures can be combined. Easier to tell the difference between Model 1 and Model 2. Same for Fig 10 & Fig 11.

Table A1: temperature dependent Henry's law constant: standard temperature in 258K?

[Figure]

References

Domine, F., Bock, J., Voisin, D., and Donaldson, D. J.: Can We Model Snow Photochemistry? Problems with the Current Approaches, The Journal of Physical Chemistry A, 117, 4733-4749, 10.1021/jp3123314, 2013.

Schwartz, S. E.: Mass-transport considerations pertinent to aqueous phase reactions of gases in liquid-water clouds, in: Chemistry of multiphase atmospheric systems, Springer, 415-471, 1986.

Traversi, R., Udisti, R., Frosini, D., Becagli, S., Ciardini, V., Funke, B., Lanconelli, C., Petkov, B., Scarchilli, C., Severi, M., and Vitale, V.: Insights on nitrate sources at Dome C (East Antarctic Plateau) from multi-year aerosol and snow records, 2014, 10.3402/tellusb.v66.22550, 2014.

———————————————————

---

## Referee Comment (RC2) · Anonymous Referee #2 · 20 Feb 2017

[Summary]

In this study, Chan et al. develop a set of process-oriented models in which the uptake and release of HNO3 from the surface "skin-layer" snowpack is simulated on the basis of "first principles" without employing additional tuning parameters from laboratory data. The reversible physical processes of surface adsorption/desorption, co-condensation with water vapor, solvation into the disordered interface (DI), solid-state diffusion into the core of ice grains and solvation into the micro-pockets of brine are formulated in the model, some of which are then turned on and off to identify the key process(es) involved in the incorporation of HNO3 into the surface snow in the Antarctic. By comparing their model results with observations of nitrate concentrations in the surface snow layer from

[Figure]

Dome C (high plateau) and Halley (coast) in the Antarctica, the authors conclude that the incorporation of HNO3 into the liquid-like DI is not a viable process. Although this appears to be one of the important results of the study, I have some difficulty in and/or objections to the rationale by the authors regarding the formulation of the HNO3 concentrations in the outermost layer of the snow grain, as detailed below. The authors also conclude that the incorporation of HNO3 at relatively warm temperatures (and yet below the freezing point) above the eutectic point(s) of the mixture(s) of ice and ionic impurities is accounted for largely by the micro-pockets of brine. This part of finding is quite convincing. I think the paper will merit publication in ACP once the authors have addressed the following concerns.

[Major comments]

1. Apparently, there is a loose interchange of what the grain-surface HNO3 concentration (HNO3 (surf)) represents while formulating the different processes involved/hypothesized in its determination. In Eq. (6) in Section 3.1.1, the authors simply take the sum of two terms, namely, the concentration due to surface adsorption (HNO3(ads)) and that due to co-condensation (HNO3(cc)). Although the unit of (HNO3(ads)) is carefully matched to allow this summation, I am not so sure if it is really legitimate to assume that all the surface-adsorbed HNO3 is automatically transferred into the bulk volume of the outermost solid-ice layer of the snow gain. It seems that the authors' claim for employing the first principles is partially broken here. Is it not more appropriate to assume that what happens on the surface stays on the surface and that [HNO3(ads)] is left out from Eq. (6)? I see the same problem in Eq. (12) in Section 3.1.2 where the authors assume that all the HNO3 dissolved in the liquid-like disordered interface (HNO3(DI)) is automatically transferred to the outermost solid-ice layer (HNO3(surf)). In my opinion, all these assumptions of automatic "phase" transfer (between the surface and the solid ice and between the liquid-like DI and the solid ice) should be adapted somehow to the one in compliance with the limitation of HNO3 solubility to the solid ice (Thibert et al., 1998). The authors run an alternative model by

calling it the "equilibrium approach", which I think should be adopted as a base case except that kinetic aspects should be formulated into this version of the model.

2. The authors do not provide sufficient details about their model formulation of the disordered interface (DI) on the surface of the ice grain. How thick is the DI? Does the thickness of the DI change with temperature? Does it make sense to assume the fixed (constant) pH especially when the chemical composition of the DI is controlled predominantly by $HNO_3(gas) = H^+(DI) + NO_3^-(DI)$ at Dome C? These are the critical points that should be discussed in detail before rejecting the hypothesis of the $HNO_3$ incorporation into the DI.

3. It is not clear enough whether the kinetic limitation to the growth and decay of the snow grain $HNO_3$ concentrations is caused mainly by mass transfer between the gas phase and the grain surface or by solid diffusion into the entire volume of the snow gain. This question should be discussed in some detail especially when contrasting the behavior of $HNO_3$ between the "kinetic" and "equilibrium" approaches such as in Section 6.1. Also, the authors may want to refer to the work by Bock et al. (2016) on the matter of timescales due to various kinetic processes.

4. I am puzzled by the description of the rate of snow grain growth and shrinkage in Section 3.1.1. Eq. (9) implies that the change of the snow grain volume is calculated by the molecular diffusion of water vapor through its microscopic concentration gradient around the snow grain. But then the authors admit that this approach does not work owing to the input data limitation and instead "the macroscopic (few mm) water vapour gradient across the skin layer was used to estimate the condensation and sublimation processes". Is the same equation still used for calculating dV/dt? In Sections 4.1 and 4.2, the authors state that meteorological input data have been obtained at 1.6 m and 1 m above the snow surface at Dome C and Halley, respectively. Is it then assumed that the water vapor concentrations are assumed to be constant with height between a few mm and 1-1.6 m above the snow surface? Please clarify. Also, is it possible to validate the authors' macroscopic approach of calculating the water vapor flux by

field observations if any? This seems to be important as background information for discussing the role of co-condensation in Section 6.2. By the way, I think $R_{\text{eff}}$ in Eq. (9) should be squared to be consistent in the physical dimension between LHS and RHS of the equation. Is it simply a typographic error?

5. The authors adopt the formulation of the $\alpha_0$ (Hudson et al., 2002), $N_{max}$ (Crowley et al., 2010) and $K_{eq}$ (Burkholder et al., 2015) from different sources. In fact, all of these could have been adopted from Crowley et al. (2010). It seems appropriate to discuss why the authors pick their experimental values/formulae from the different sources and how much difference their choice would generate in the model behavior.

6. The quality of English needs to be improved significantly. There are so many grammatical and spelling errors, only a tiny part of which I can comment below as technical suggestions. This problem is really glaring but may be largely corrected by a copy-editor once the manuscript is accepted for publication. Nonetheless, there seems to be a room for improvement that should be addressed by the authors before that stage. I strongly recommend careful and diligent proofreading by the team of the authors (especially if the editor asks another round of review).

[Minor comments]

1. I think that "T – Tf" should be reversed to "Tf – T" in Eq. (4) to let $\phi_{H2O}(T)$ be the positive values. And I think that this inherits from what I believe is a typographic error in Cho et al. (2002) cited for Eq. (4). Am I wrong? Please double check.

2. The variable "z" refers to the distance from the snow grain surface in Eq. (9), whereas it refers to the depth in the snowpack in Eq. (11). Please adjust the notation to avoid confusion between the two.

3. On Line 92, it is stated that "thickness of the DI" is a tuning parameter in Toyota et al. (2014). In fact, they calculate the thickness of the DI on the basis of the Cho et al. formula, which is used by the present authors for calculating the volume of the micropockets of brine. The difference from the present study is that Toyota et al. assume the brine covers the entire surface of the snow grain just like the DI.

4. On Line 255, it is stated that "Dg is the gas-phase diffusivity". It should be stated clearer that Dg is the gas-phase diffusivity of HNO3. It would also be nice to list how Dg is calculated in Table A1.

5. Lines 615-616: It appears to me in Figure 1 that the changes in pH of the order of 1 have a similar level of impact on the effective Henry's law coefficient to the changes in temperature of the order of 10 K. I don't quite understand what the authors try to point out here.

6. Lines 558-562, "…, which are 1-2 orders of magnitude higher than the averaged modelled temperature gradient (listed in Sect. 3.1.1)": It seems that this is not discussed/listed at all in Section 3.1.1. Please expand the discussion by referring to what the realistic range of the vertical temperature gradient should be.

7. Lines 625-631 and Figure 11: Please be more specific and detailed about what make up the "other ions".

8. Table A1: Sander (2015) is a compilation of Henry's law coefficients, but here it is cited for the temperature dependence of alpha. Please double check if it is the correct reference. Also, the "enthalpy of activation" is much too vague as terminology for $\Delta_{obs}H$. Please expand.

9. Table A1, values of $\Delta_{sol}H$ and $\Delta_{obs}H$: I think they should have been $-72.3$ and $-44$, respectively (the minus sign is missing). Please double check.

10. Table A1, footnote i: I suppose that the authors meant to formulate the temperature dependence of alpha somehow consistently with $d\ln[\alpha/(1-\alpha)]/d(1/T) = -\Delta_{obs}H/R$ (e.g., Jayne et al., 1991). But I cannot reconcile with the authors' formulation in this footnote. Am I wrong here? Please double check if it is formulated properly.

[Technical suggestions]

Line 216: d HNO3 / dt -> d [HNO3(ads)] / dt

Line 217: Substituting $k_{ads}$ -> Substituting $k_{des}$

Line 321: organic -> inorganic (?)

Line 407: tough -> trough (?)

Line 414, Eq. (19): $MM_{H2O}$ -> $M_{H2O}$

Line 603: ... varying $T_0$ by 4 K up to 242 K and pH by $\pm 0.4$ up and down between 5.2-6.4

Figure 1: Add the unit of temperature for the figure legends in (b): "T = 230 K", etc.

Figure 10: Is it not possible to use the same scaling in Y-axis for all the data shown here, for example, by using logarithmic scaling?

Table A1: Accommodation coefficient at standard temperature -> Accommodation coefficient at reference temperature (220 K)

Table A1, footnote ii: 258 K -> 298 K (?)

[Reference cited in my comments but not listed in the manuscript]

Jayne, J. T., Duan, S. X., Davidovits, P., Worsnop, D. R., Zahniser, M. S., and Kolb, C. E.: Uptake of gas-phase alcohol and organic acid molecules by water surfaces, J. Phys. Chem., 95, 6329-6336, 1991.

---

## Editor Comment (EC1) · T. Bartels-Rausch (Editor) · 19 Dec 2017

With this editorial comment, I'd like to summarize the discussion between me and the authors that continued via email after the referee comments have been received.

The work by Chan et al. describes field measurements and snow-chemistry modelling of HNO3 in Antarctic surface snow and the overlaying air. The performance of 2 model approaches to capture measurement at 2 sites that are distinct in ambient temperature –among other factors- are compared. One model introduces a liquid brine fraction to parameterize snow-chemistry, while the other model built on the quasi-liquid layer.

[Figure]

Interactive
comment

I find the comparison of these two concepts – liquid and liquid-like – ability to capture trends in a valuable data set most interesting for the general audience of ACP. As with any snow chemistry model, also the approaches presented here include parameters some of which are strictly not physically based or have a large uncertainty. This is not per se a disadvantage. The importance is to present enough details and to acknowledge that the models are not built on first principals alone, which is now the case throughout the manuscript. Further similarities to previous models are now acknowledged. Along with this came a change in tone of the manuscript and achievements in previous work are now acknowledged. While I remain very skeptical about de-facto parameterizing the QLL as a layer with a thickness in the $\mu$m range, I leave it to the reader to judge.

To ensure transparency, please find a copy of the email exchange with the authors here:

Question: Thanks for the revised version. I'm still a little confused about one aspect that a referee has previously raised as major concern. It is related to the volume of the DI. Actually, also to the volume of the [HNO3 (surf ) ] in general. I still have difficulties to understand 1) how you can get a mass balance if the DI has a volume of 0 and 2) how can define U(REff) without a volume of the DI? Answer: All surface or boundary concentrations actually refer to the volume of the outermost concentric model shell of the snow grain. In particular, at T>T0 the outermost model shell of the snow grain becomes essentially a DI, with Henry's Law describing air-DI exchange and defining its NO3- concentration. The outermost model shell thickness varies with grain size, which in turn is constrained by the observed seasonal cycle of SSA. The thickness ranges between 0.5 in winter and 1.5 $\mu$m in summer. To clarify we have updated the model description (P7,Âăl97-226) as well as the assumptions made regarding the DI in Model 1

Changes to text: "3 Modelling Approach The aim of this paper is to focus on the physical exchange mechanisms of HNO3 between air and snow to predict the concentration

of nitrate in the skin layer of the snowpack, as a first step towards a full snowpack model. The two models are constrained by the observed atmospheric concentration of HNO3, air temperature, skin layer temperature, atmospheric pressure and humidity. The loss or gain in the atmospheric HNO3 due to the mass exchange between air and snow are included implicitly by constraining the models with the observed atmospheric concentration of HNO3. The following assumptions were made in both Model 1 & 2: 1) the concentration of HNO3 in snow interstitial air is the same as in the overlying atmosphere justified by a short characteristic time scale for gas-phase diffusion of âĹij 1 s (Table 1); 2) the physical properties of the skin layer are homogeneous and include density and specific surface area (SSA); and 3) the snow grain is assumed to be a radially symmetrical sphere with an effective radius, $R_{eff}$, which is estimated from the SSA as follows:

Eq.6

where ice is the density of ice. Snow metamorphism and resulting changes in snow grain size are not modeled explicitly, but are approximated instead by prescribing temporal changes in SSA. Here an annual cycle of SSA is included based on observations at Dome C (Picard et al., 2016), ranging from 25 m2kg−1 in summer to 90 m2kg−1 in winter (details in Sect. 4.3 and Fig. A4a), and yielding a $R_{eff}$ of âĹij130 $\mu$m in the summer, which gradually reduces to âĹij 30 $\mu$m in the winter (Fig. A4b). Modeled co-condensation (Eq. 9 & 10) does not change model snow grain size, since the involved ice volumes are relatively small compared to the volume of the snow grain. The model set up implies also that the snow grain size remains constant during each model time step of $\Delta t$ = 10 min. For the calculation of solid-state diffusion the snow grain is divided into N concentric shells of equal thickness. To optimise model performance and computational cost, the number of concentric shells is fixed to N = 85, yielding a model shell thickness $\Delta r$ of âĹij 1.5 $\mu$m in summer and âĹij 0.5 $\mu$m in winter due to seasonal change in grain size. $\Delta r$ remains at all times smaller than the minimum typical length-scale, <x>, a molecule diffuses over a finite time, $\Delta t$, and described by the root-mean

square displacement , <x> = $\sqrt{6\Delta t k_{diff}}$. Minimum typical length-scales occur in winter when air temperatures are lowest, and for a modeling time step, $\Delta t$ = 10 min, they range between 1.5 $\mu$m at Dome C and 5.5 $\mu$m at Halley."

Sect 3.1, line 241-259:

'The physical properties of the DI are still poorly known, and currently there are no physical parameterizations available to estimate DI thickness, partitioning coefficients or diffusivities. Hence, for the DI in Model 1 the following four assumptions are made: 1) the partitioning between air and the DI follows Henry's law, similar to previous models (e.g.Thomas et al., 2011 & Toyota et al., 2014); 2) the model geometry described above implies that the DI, i.e. the outermost model shell of the snow grain, follows the seasonal cycle of snow grain specific surface area and has a thickness of 1.5 $\mu$m in summer decreasing to 0.5 $\mu$m in winter. A seasonal cycle is qualitatively consistent with laboratory measurements, which show that DI thickness increases with temperature (Bartels-Rausch et al., 2014). But the absolute model values are larger than previous lab measurements on pure ice, which range from the thickness of a monolayer of water (0.3 nm) to 100 nm, depending on the measurement technique (e.g. Bartels-Rausch et al., 2014), or values adopted in previous model studies (range 10-30 nm) (e.g. Thomas et al., 2011, Toyota et al., 2014, Murray et al., 2015). However, DI thickness is also sensitive to the type and concentration of impurities, and generally increases with ion concentration (e.g. Dash et al., 2006; Bartels-Rausch et al., 2014); 3) the DI is interacting with the bulk ice, i.e. solvated nitrate ions diffuse into the interior of snow grain and the mass transport is determined by the solid-state diffusion coefficient of ice, $k_{diff}$, and the concentration gradient across the snow grain; and 4) the solid-state concentration of nitrate in the bulk is limited by the thermodynamic equilibrium solubility of ice (e.g. by Thibert et al., 1998 as shown in Eq. 19), except the outermost model shell of the snow grain.'

Question: (The nitrate concentration at the centre is set to U (0) = 0 and at the grain boundary U (Reff ) = [HNO3 (surf ) ], which is defined by surface adsorption and cocondensation at temperatures below To (Eq. 7) or by solvation into the infinitesimal DI at temperature above To (Eq. 13).) Wouldn't then the concentration gradient and thus the diffusivity not also depend on the choice of the layer thickness in the diffusion parameterisation?

Answer: At T>To the concentration in the outermost model shell is determined by Henry's law and is independent of model shell thickness and grain size. AT T<To the concentration in the outermost model shell depends on grain size, because it depends on number of surface sites (Eq.3).Ăă But the concentration gradients and associated mass flux depend on the current snow grain size, which is not freely chosen but constrained by observed SSA. However, the absolute number of molecules in the outermost model shell depends on the choice of shellÂăthickness, and influences the NO3- bulk concentration. This model uncertainty is clarified and discussed in a short paragraph in Sect 6.3 line 680-694:

Changes to text: 'The onset and thickness of the DI not only depend on temperature, but also the speciation and concentration of impurities present within the snow grain (McNeill et al., 2012; Dash et al., 2006). Different impurities have different impacts on the hydrogen bonding network at the ice surface and hence have a different impact on the thickness of the DI, leading in general to a thickening compared to pure ice (Bartels-Rausch et al., 2014). However no accepted model parameterisation is available. In this model imposing a seasonal cycle of SSA and therefore grain size causes the thickness of the outermost model shell to vary between 1.5 $\mu$m in summer and 0.5 $\mu$m in winter (Sect. 3.1), relatively large values and potentially contributing to the positive bias in Model 1. This is explained as follows: the bulk concentration of NO3- is calculated as the sum of number of molecules in each model shell divided by the total volume of the snow grain (Eq. 17). At TÂă > ToÂă the outermost model shell is equivalent to a DI and its concentration is determined by Henry's Law (Eq. 13), which is independent of grain size and thus model shell thickness $\Delta$r. However, the absolute number of molecules in each model shell including the DI, increases with $\Delta$r yielding a larger bulk

concentration in summer. Choosing a thinner outermost model shell may reduce the Model 1 bias at Halley.'

---

## Author Response (AR1)

We thank the reviewers for their reviews and recommendation to publish. We have considered every point and corrected the paper to include their points. The referees comments are in blue, responds from the authors are in black and revised text are in red.

**Referee 1**

(1) Atmospheric nitrate (gaseous $HNO_3$ + particulate nitrate) is assumed to be dominated by gaseous $HNO_3$ (which is supported by previous studies). In this work, the physical exchange of gaseous $HNO_3$ in the snow interstitial air (SIA) and the snow grains is described explicitly by different models. However, the mass exchange of $HNO_3$ between the SIA and air above snow (where the atmospheric nitrate is measured) is missing. Mass exchange between the SIA and air above snow is largely controlled by processes such as turbulent transport and wind pumping. How these processes would affect the bulk nitrate in the skin layer of snow needs to be clearly addressed.

It is assumed that the boundary layer was well mixed such that the surface $HNO_3$ concentration is same as the observation made at approximately 1 m above the surface. A table of characteristic times of different processes has now been added to the manuscript. The focus of this paper is to describe the interaction between the skin layer of the snowpack (top 4 mm) and the overlying atmosphere. The characteristic time of molecular diffusion for vertical mass transport between the SIA at 4 mm and the air above is only of the order of a second, therefore, is assumed to be in equilibrium.

The characteristic time of various processes are listed in Table. 1

**Table 1.** Characteristic times associated with gas-phase diffusion, mass transport and uptake of gas into ice grain

| Process | Expression | Order of magnitude, s |
|---|---|---|
| Interfacial mass transport to a liquid surface[i] | $\frac{3\bar{v}\alpha_{aq}}{4R_{\mathrm{eff}}}$ | $10^{-7}$ |
| Gas-phase diffusion to the surface of a spherical droplet[ii] | $\frac{3\,D'_a}{R^2_{\mathrm{eff}}}$ | $10^{-4}$ |
| Molecular diffusion between snowpack and the atmosphere[iii] | $\frac{z^2}{D'_s}$ | $10^0$ |
| Liquid-phase diffusion within a water droplet[iv] | $\frac{4\,R^2_{\mathrm{eff}}}{\pi^2\,k_{\mathrm{diff(aq)}}}$ | $10^0$ |
| Surface adsorption on ice[v] | $\frac{1}{k_{\mathrm{des}}}$ | $10^3$ |
| Solid-state diffusion within a snow grain[vi] | $\frac{4\,R^2_{\mathrm{eff}}}{\pi^2\,k_{\mathrm{diff}}}$ | $10^6$ |
| Photolysis at a snowpack surface[vii] | $\frac{1}{J}$ | $10^7$ |

[i] Sander (1999), with an effective radius, $R_{\mathrm{eff}}$ = 70 μm, and accommodation coefficient on liquid water, $\alpha_{aq} = 7.5 \times 10^{-5}\exp(2100/\mathrm{Temp})$ (Ammann et al., 2013). [ii] Sander (1999), with an effective molecular diffusivity, $D'_s = D_a/\tau_g$, where the tortuosity, $\tau_g = 2$ and molecular diffusivity in free air at 296 K, $D_a(296\mathrm{K}) = 87\ \mathrm{Torr\,cm^2\,s^{-1}}$ (Tang et al., 2014). [iii] Waddington et al. (1996), with a snow layer thickness, $z$ = 4 mm. [iv] Finlayson-Pitts and Jr. (2000), with a diffusion coefficient in liquid water, $k_{\mathrm{diff(aq)}} = 1 \times 10^{-9}\ \mathrm{m^2\,s^{-1}}$ (Yuan-Hui and Gregory, 1974) . [v] Crowley et al. (2010), with an equilibrium constant for Langmuir adsorption, $K_{eq} = 2 \times 10^{-16}\ \mathrm{m^3\,molecule^{-1}}$ and adsorption coefficient, $k_{\mathrm{ads}} = 1.7 \times 10^{-19}\ \mathrm{m^3\,molecule^{-1}\,s^{-1}}$. [vi] Finlayson-Pitts and Jr. (2000), with a diffusion coefficient in ice, $k_{\mathrm{diff}} = 6 \times 10^{-16}\ \mathrm{m^2\,s^{-1}}$ (Thibert et al., 1998). [vii] Finlayson-Pitts and Jr. (2000), with a surface $NO_3^-$ photolysis rate, $J$, = $10^7\ \mathrm{s^{-1}}$ (Thomas et al., 2011).

Such information and assumptions are now included in in Sect. 4.1 (Page 12, line 359-363)

"The atmospheric boundary layer is assumed to be well mixed so that the atmospheric nitrate at the snowpack surface would be the same at 1 m. The characteristic transport time of $HNO_3$ from the snowpack surface to the skin layer (4 mm) is on the order of $10^0$ s, which is much shorter compared to the temporal resolution of the model of 10 min (Table 1), and therefore, the $HNO_3$ concentration of the skin layer was assumed to be the same as above the snow."

and Sect. 4.2 (Page 12, line 390-391)
"Again, the atmospheric boundary layer is assumed to be well mixed that the nitric acid concentration at the snowpack surface would be the same as at 7-8 m"

(2) Model 2 incorporates the micro-liquid pocket. This topic is of great interest since the brine formed by impurities may not cover the entire grain surface due to limited wettability at cold temperatures. However, instantaneous air/micro-liquid pocket equilibrium is assumed. This seems to be oversimplified. For highly soluble species such as $HNO_3$ in liquid water (effective Henry's law constant > 10^14 M atm^-1, Fig 1), interfacial transport or even gas diffusion (in this case, gas diffusion in the SIA) may well become the rate limiting steps. The timescale of the SIA/micropocket equilibrium needs to be examined before assuming equilibrium.

A table of characteristic times (Table 1) of different processes has been added to the manuscript. The characteristic times a) of interfacial mass transport across a liquid surface of a droplet with a 70 μm radius, b) gas-phase diffusion toward a droplet with 70 μm radius, and c) vertical mass transport to SIA at 4 mm depth are all significantly smaller than the characteristic time of surface adsorption, solid-state diffusion.

The following lines been change in Sect. 3.2, Line 314-320
"An instantaneous equilibrium is assumed because 1) the volume of the liquid solution is small ($10^{-7}$ –$10^{-6}$ % of the total volume of the ice grain, discussed below) 2) HNO3 is highly soluble in water; 3) the characteristic time of the interfacial mass transport across a liquid surface of a droplet with 70 μm is only $\sim 10^{-7}$ s (Table 1); and 4) the diffusion rate is faster in liquid (At 0∘C, $NO^-3$ diffusion of $NO^-3$ is $9.78 \times 10^{-10}$ $m^2$ $s^{-1}$ in liquid, Yuan-Hui and Gregory, 1974 ) than in ice (At 0∘ C $NO_3^-$ diffusion rate is $3.8 \times 10^{-14}$ $m^2$ $s^{-1}$ in ice). The characteristic time of liquid-phase diffusion within a 70μm diameter water droplet is $\sim 10^0$ s (Table 1)."

(3) From the model point of view, Model 2 does not really specify or depend on the location of liquid water, i.e. whether the liquid water is covering the whole/part of the grain surface as a thin layer, or is located in grooves at grain boundaries and tripe junctions. It appears mathematically that, in Eq(4) + Eq(17), only the liquid water content matters while the location of liquid water does not.

The reviewer is correct, in fact we don't know the location from the current data set; the liquid water is treated as micro-liquid pockets that can be found at an unspecified location in grooves at grain boundaries or triple junctions as stated in the Introduction (Line 101-102). The assumption implies the grain surface area being covered by liquid water is negligible and therefore mostly ice.

For clarification the following text has been added, in Sect. 3.2 (Page 10, line 302-303)

"Liquid water is assumed to be located in grooves at grain boundaries or triple junctions between grains and in the form of micropockets. This assumption implies the grain surface area being covered by liquid water is negligible. "

(4) The authors claim that the physical exchange models are based on "first principles" (what exactly are first principles btw) and hence without requiring any tuning parameters. This seems not true: some parameters involved in the models are still somewhat adjustable and/or lack direct observational support, such as max number of adsorption sites, threshold temperature T0, microscopic H2O density gradient, eutectic temperature, etc.

'First principles' are based on physical laws and relationship. The "tuning parameters" are referred to scaling factors that use to fit the model to observations. However, some of the physical parameter used in the current work have ill defined values which merited a study of the model sensitivity against some of the parameterisations and inputs were analyzed. The results of model sensitivity are now listed in Table 4.

**Table 4.** Sensitivity Test for Model 1 and 2 based on the coefficient of variation of RMSE, $C_v$(RMSE), the metric used as goodness of fit. Note that column one is not fitted to the observation and the values are only varying to show the sensitivity of the models against inputs and parameterisation.

| Parameter | | Model 1 | | | | | | Model 2 | | | | | |
| --- | --- | --- | --- | --- | --- | --- | --- | --- | --- | --- | --- | --- | --- |
| | | Dome C | | | Halley | | | Dome C | | | Halley | | |
| | | Whole year | Winter-Spring | Summer | Whole year | Winter | Spring-Summer | Whole year | Winter-Spring | Summer | Whole year | Winter | Spring-Summer |
| Control | | 1.34 | 0.73 | 1.11 | 89.28 | 27.78 | 87.15 | 0.84 | 0.73 | 0.67 | 0.84 | 1.08 | 0.65 |
| [HNO$_3$] | −20% | 0.98 | 0.60 | 0.81 | 71.19 | 22.12 | 69.5 | 0.80 | 0.62 | 0.64 | 0.77 | 1.10 | 0.56 |
| | +20% | 1.73 | 0.90 | 1.45 | 107.36 | 33.43 | 104.80 | 0.95 | 0.88 | 0.76 | 0.92 | 1.07 | 0.75 |
| SSA | −10% | 1.06 | 0.63 | 0.88 | 79.35 | 24.79 | 77.46 | 0.83 | 0.67 | 0.67 | 0.84 | 1.10 | 0.65 |
| | +10% | 1.63 | 0.84 | 1.36 | 99.22 | 30.75 | 96.86 | 0.84 | 0.78 | 0.67 | 0.83 | 1.07 | 0.65 |
| $\alpha$ | −10% | 1.34 | 0.73 | 1.11 | 79.35 | 24.78 | 77.46 | 0.83 | 0.73 | 0.67 | 0.83 | 1.08 | 0.65 |
| | +10% | 1.34 | 0.73 | 1.11 | 79.35 | 24.80 | 77.46 | 0.83 | 0.73 | 0.67 | 0.83 | 1.08 | 0.65 |
| $N_{max}$ | −10% | 1.32 | 0.67 | 1.10 | 89.27 | 27.77 | 87.15 | 0.83 | 0.69 | 0.67 | 0.84 | 1.09 | 0.65 |
| | +10% | 1.36 | 0.80 | 1.13 | 89.29 | 27.78 | 87.15 | 0.84 | 0.77 | 0.67 | 0.84 | 1.07 | 0.65 |
| $T_o$ (Model 1) or | -2 K | 3.53 | 0.91 | 3.00 | 90.45 | 42.54 | 87.31 | 0.95 | 0.92 | 0.75 | 0.85 | 1.12 | 0.65 |
| $T_e$ (Model 2) | +2 K | 0.50 | 0.64 | 0.36 | 67.49 | 25.33 | 65.62 | 0.73 | 0.65 | 0.58 | 0.86 | 1.07 | 0.65 |
| | +4 K | 0.61 | 0.65 | 0.47 | 50.76 | 23.86 | 49.00 | 0.72 | 0.65 | 0.57 | 0.88 | 1.06 | 0.67 |
| pH | -0.4 | 1.34 | 0.73 | 1.11 | 89.28 | 27.78 | 87.15 | - | - | - | - | - | - |
| | +0.4 | 1.34 | 0.73 | 1.11 | 89.28 | 27.78 | 87.15 | - | - | - | - | - | - |
| | +0.8 | 1.34 | 0.73 | 1.11 | 89.28 | 27.78 | 87.15 | - | - | - | - | - | - |
| [NO$_3{}^-$] | −20% | 1.85 | 0.98 | 1.54 | 111.87 | 34.84 | 109.2 | 0.99 | 0.96 | 0.79 | 1.09 | 1.08 | 0.93 |
| | +20% | 1.04 | 0.61 | 0.86 | 74.22 | 23.07 | 72.45 | 0.80 | 0.64 | 0.64 | 0.74 | 1.10 | 0.51 |

(5) Comparison between models and measurements needs to be discussed in the context of their respective uncertainty ranges. What is the measurement uncertainty of skin layer nitrate concentration? What is the model uncertainty propagated from the inputs and parameters?

Results of the sensitivity tests on atmospheric nitrate concentration, accommodation coefficient, maximum number of adsorption sites, threshold temperature or eutectic temperature and skin layer snow nitrate concentration are listed in Table 4 (See the comment above).

(6) The quality of English could use some polish.

In addition, the authors claim that the photochemistry of snow nitrate can be ignored due to slow photolysis in this region. Well, "what goes up must come down" and vice versa. What processes are then responsible for the loss of snow nitrate? And what is driving the seasonal variations of snow nitrate in this region? Snow nitrate can't cannnot always accumulate. This is perhaps not the main focus of this work, but the fact that only snow nitrate sources are included in the model may be quite confusing.

See the comment above

The observed atmospheric concentration $HNO_3$ is used as a model constrain, which implicitly included change in atmospheric $HNO_3$ concentration due to air-snow exchange.

In this particular region of the snowpack the loss of nitrate by photolysis is slow compared to the physical uptake of nitrate by adsorption and co-condensation.

The following text has been added to Sect. 3, Line 187-193

"The loss or gain in the atmospheric $HNO_3$ due to the mass exchange between air and snow are included implicitly by constraining the models with the observed $HNO_3$ concentration. The aim of this paper is to focus on the exchange mechanisms between air-snow, and by limiting the working layer to the skin layer, the following assumptions can be made, 1) homogenous physical properties across the skin layer, such as snow density and SSA. 2) the $HNO_3$ concentration in SIA is in equilibrium with the overlying atmosphere due to a short characteristic time (Table 1)."

Specific comments:

Page 3, Line 61: the characteristic times of surface adsorption and solid-state diffusion for $HNO_3$... please provide more details (either literature or point to later sections).

Details are now listed in Table 1 (See above comment)

Page 3, Line 83: define skin layer. What is the thickness of this skin layer in the model and why this value is chosen? Or is it simply the layer in which the bulk ion concentrations are measured?

Information regarding to the skin layer been added to Page 2, Line 52-54

"Here in this paper, the skin layer is defined as the top 4 mm of the snowpack, which is the depth of which the surface snow nitrate samples were collected at Dome C (Sect. 4.1)."

Since the model is limited to the skin layer, it seems that there is no exchange between the skin layer and the deeper snow. However, previous studies(e.g. Traversi et al 2014) indicated that temperature gradients and wind pumping exist in the snowpack, therefore nitrate could be mobilized by physical processes reaching much deeper than the "skin layer" in this model (a few mm?).

The aim of this paper is to demonstrate the interaction between skin layer nitrate and atmospheric HNO$_3$ with a simple physical model without a scaling factor. Atmospheric nitrate can reach deeper than the skin layer via wind pumping and temperature gradient, however, to reproduce nitrate concentration in deeper snow requires a complicated multi-layer model. Developing a multi-layer model is an extremely large undertaking and is being performed at the time of writing. The conclusion highlighted the referee's point and further work will address this.

Within the Conclusion, Line 704-716
"Despite the simplified parameterisation of processes in Model 2, such as the impurities content in snow, liquid pockets located in different locations were treated as one and had the same chemical properties as bulk liquid, it is a promising step towards parameterising the interactions between air and snow. The models presented here are describing the exchange between air and the skin layer that the uptake processes are much quicker than the photochemical loss, and therefore, can be modelled by physical only processes. Atmospheric nitrate can reach deeper than the skin layer via wind pumping and temperature gradient, however, the nitric acid concentration in SIA is expected to be small compared to the overlying atmosphere due to the high uptake of nitrate near the surface of the snowpack. A lower HNO3 concentration in SIA implies a smaller uptake in deeper snow, and hence the photochemical loss cannot be assumed to be negligible in deeper snow. Therefore, a more complex multi-layer model including both physical and chemical processes is required to reproduce the nitrate concentration in deeper snow and being implement in regional and global atmospheric chemistry model.."

Page 5, Line 141: the solid-state diffusivity is introduced here, and hence characteristic time can be calculated. Please compare to other processes, e.g. surface accommodation and gas-phase diffusion
The characteristic times of other processes are now listed in Table 1 (See the comment above)

Page 6, Line 178: what is the size of snow grain?
The sentence is now written as (Line 194-195)
"For simplicity, the snow grain is assumed to be a radially symmetrical sphere with a radius, Reff , which is estimated from the specific surface area (SSA) with the follow equation:
(... Eq. 6) "
Eq. 14 is now Eq. 6 and moved to Sect. 3.

An extra sub-plot of the effective grain radius has been added to the Appendix, Fig A3

[Figure]

**Figure A3.** (A) Year-round estimates of the specific surface area (SSA) of snow at Dome C (−) and Halley (−−) were interpolated from observations at Dome C during 2012-2015 by Picard et al. (2016) (×). The SSA estimates for Halley take into account the shorter cold period compare to Dome C, which tends to have larger SSA. (B) Year-round estimates of effective grain radius ($R_{eff}$) at Dome C (−) and Halley (−−) derived from Eq. 6.

Page 7, Eq 7: both adsorption and co-condensation contribute to surface $HNO_3$. Is co-condensed $HNO_3$ available for desorption? Judging from Eq 6 it seems the answer is yes, yet in Eq 7 it seems co-condensed $HNO_3$ is not included. Also, will the cocondensed $HNO_3$ molecules undergo solid diffusion?

Yes, the grain surface $HNO_3$ concentration has contributions from the sum of adsorption, desorption and co-condensation or co-sublimation. Condensation or sublimation depends on the sign of the water vapour gradient and hence the sign of the rate of volume change (Eq. 10)
The grain surface concentration of $HNO_3$ is then treated as the boundary concentration for solid grain diffusion driven by concentration gradient of the grain surface and the centre of grain.

For clarification the following text has been added, in Page 7, Line 215-217

"where [$HNO_3$(ads)] is the concentration contributed from the sum of surface adsorption and desorption (Eq. 8), and [$HNO_3$(cc)] is the concentration contributed from the co-condensation or co-sublimation (Eq. 9)."

and Page 8, Line 238-240
"The temperature gradient and relative humidity gradient between the surface of the snowpack and the skin layer create a gradient in water vapour pressure, which drives condensation or sublimation of snow, depending on the sign of the gradient."

Page 8, Line 248: what is the thickness of this DI covering the entire grain surface?
The DI is treated as the boundary of the snow grain, of which the concentration of DI is used as the boundary condition for the diffusion into the snow grain. Therefore, no thickness is assigned to the DI.
For clarification, the following lines (Page 9, line 275-277) are now included in the manuscript.
"Note that in this model the DI is treated as the boundary between the air and bulk ice. The concentration of the DI is used as the outermost boundary condition for solid-state diffusion within the grain, therefore, the DI has no thickness."

Also, Eq 13 describes d[$HNO_3$(DI)]/dt, and there should be another equation for d[$HNO_3$(g)]/dt accordingly. Please provide this. Finally, I may be wrong but shouldn't mass transfer (Eq 13 and d[$HNO_3$(g)]/dt) depend on liquid water content of some sort?
Both models presented here are constrained by the observed gas phase $HNO_3$ concentration with time, therefore, the loss of $HNO_3$(g) due to mass transfer is included implicitly.

Page 8, Eq 8: this equation describes co-condensation. How about H2O sublimation? Does $HNO_3$ undergo co-sublimation (or whatever the term should be) as well?
Both co-condensation or co-sublimation occur depending on the sign in Eq. 10.

Page 10, Line 294: again, for highly soluble species in liquid, interfacial transport or gas diffusion may be limiting (Schwartz, 1986). Please calculate the equilibrium timescale and discuss in the context of other mass transfer processes.
Details are now listed in Table 1.

Page 13, Line 399: define winter (and other seasons too). The Northerners would appreciate this.
Has been added to Page 11, Line 336-340. It reads
"… in summer (mid November till end of January) and down to –80∘C in the winter (April to mid September). The diurnal temperature variation is ~10 K in summer, spring (mid September till mid November) and autumn (February to March)."

Page 14, Line 432: "However, Model 1… overestimated concentration by a factor of 1.5-5 in December". Which model 1? With 238 K or 242 K?

Now (Line 448-449) written as "However, Model 1 (with T0 = 238 K) did not capture the peak in early February and overestimated concentration by a factor of 1.5-5 in December."

Page 15, Line 476: "the combination of larger temperatures and a larger diurnal temperature range" this sentence is confusing.
The sentence (Line 491-192) has been corrected and now reads
"… the combination of warmer temperatures and a larger range of diurnal temperature causes …"

Page 16, Line 493: "it is possible that the snow NO3- concentration measured from Halley might be 'diluted' from deeper snow layer..." then can you extend your model to cover deeper layers, or simply increase the skin layer thickness? Also, as shown in Fig 11, Model 2 underestimated nitrate for the majority of the time (Line 458-459). If measured snow NO3- was indeed diluted, would this mean the model underestimates even more?
The models presented here would lose their physical meaning by increasing the thickness, of which is assumed to be homogenous as well as in equilibrium with the atmosphere above. A multi-layer model is required to cover deeper layers. Moreover, Model 2 underestimates the concentration of nitrate at Halley mainly in the winter period where new snowfall events were accounted for the large surface snow nitrate.

Page 16, Line 497: what do you mean by "fixed by sea salt, ammonium or terrestrial dust"?
The sentence (Line 511-512) was rewritten as " Thirdly, atmospheric nitrate can be in a more stable forms of $NO_3^-$ , i.e. associated with $Na^+$,$Ca^{2+}$ or $Mg^{2+}$ (Beine et al., 2003)"

Page 16, Line 502: "the increase in sea salt concentration decreases the ratio of concentration of gaseous $HNO_3$ to total atmospheric nitrate". Please provide evidence.
A reference, Dasgupta et al., 2007, has been added.

Page 16, Line 503: "A possible explanation for the overestimation of NO3- concentration in both Model 1 and 2 in November at Dome C" this is not a complete sentence.
The sentence has been removed.

Page 17, Line 546: "In the summer, other processes are replaced..." this sentence is ill-formed. What are you trying to say?
The sentence (Line 562-564) is rewritten as "In the summer, the dominant process in Model 1 is solvation in DI (See Sect. 6.3) while in Model 2 the dominant process is partitioning in the micropockets (See Sect. 6.4), hence the contribution from co-condensation to the skin nitrate concentration is insignificant."

Page 19, Line 605: there is no purple on Fig 7.
Corrected. The sensitivity analysis is now moved to Sect. 6.5. The results from Model 1 at Halley are now in Fig. 8A. The purple line (on the right axis) is the results when T0 = 242 K and the text has been adjusted to demonstrate it.

Page 19, Line 628: Again this is only true if gas diffusion and interfacial transport are not limiting. Also, Model 1 output is quite sensitive to T0. How sensitive is Model 2 to the eutectic temperature?

A set of sensitivity tests have been run against inputs such as nitric acid concentration, SSA, accommodation coefficient ($\alpha$), maximum number of adsorption site (Nmax), and either the threshold temperature in Model 1(T0) or the eutectic temperature in Model 2 (Te). The coefficient of variation of RMSE (Cv(RMSE)) is used as a metric of the goodness of fit and is listed in Table 4.

Fig 1: Please include units for the effective Henry's law constant. Also I feel this belongs in the Supplementary Information. The temperature and pH dependencies of effective Henry's law constant, although are important, do not deserve the spot of the very first figure of this particular paper.

The plot of the temperature and pH dependencies of effective Henry's law constant (Figure 1) is now moved to the Appendix

Fig 4, Fig 6-11: dates on the bottom axis are difficult to read, i.e. it is hard to identify "early Feb" or "early May", ... Please set date tick labels to the first day of each month. If not enough space, rotate 90 degrees.

Figures are now has the first day of each month on the bottom axis and day of year (DOY) on the top of the graph to make it easier to read.

Fig 5: figure legend very unclear. What exactly are the scatter points? And what are "Head 1 1213", "Head 2, 1213", ...?

The figure (now Fig. 4) had been re-plotted and legend been clarified.

Fig 7 & Fig 8: I think these two figures can be combined. Easier to tell the difference between Model 1 and Model 2. Same for Fig 10 & Fig 11.

Fig 7 and Fig 8 are now combined as Fig 6, and, Fig 10and Fig 11 are now combined as Fig 8

Table A1: temperature dependent Henry's law constant: standard temperature in 258K?

The standard temperature for the calculation of temperature-dependent Henry's law is now corrected to 298K

Referee 2
[Major comments]

1. Apparently, there is a loose interchange of what the grain-surface $HNO_3$ concentration ($HNO_3$ (surf)) represents while formulating the different processes involved/hypothesized in its determination. In Eq. (6) in Section 3.1.1, the authors simply take the sum of two terms, namely, the concentration due to surface adsorption ($HNO_3(ads)$) and that due to co-condensation ($HNO_3(cc)$). Although the unit of ($HNO_3(ads)$) is carefully matched to allow this summation, I am not so sure if it is really legitimate to assume that all the surface-adsorbed $HNO_3$ is automatically transferred into the bulk volume of the outermost solid-ice layer of the snow gain. It seems that the authors' claim for employing the first principles is partially broken here. Is it not more appropriate to assume that what happens on the surface stays on the surface and that [$HNO_3(ads)$] is left out from Eq. (6)? I see the same problem in Eq. (12) in Section 3.1.2 where the authors assume that all the $HNO_3$ dissolved in the liquid-like disordered interface ($HNO_3(DI)$) is automatically transferred to the outermost solid-ice layer ($HNO_3(surf)$). In my opinion, all these assumptions of automatic "phase" transfer (between the surface and the solid ice and between the liquid-like DI and the solid ice) should be adapted somehow to the one in compliance with the limitation of $HNO_3$ solubility to the solid ice (Thibert et al., 1998). The authors run an alternative model by calling it the "equilibrium approach", which I think should be adopted as a base case except that kinetic aspects should be formulated into this version of the model.

In the models, the solid-state diffusion into the grain is driven by the concentration gradient between the grain boundary and the centre of the grain and regulated by the solid-state diffusion coefficient (Thibert et al., 1998). Abbatt, (1997), Huthwelker et al., (2004) and Cox et al., (2005) had observed a diffusion-like behaviour from flow tube study for trace gases uptake onto ice. The structure of the model presented in this paper is based on the suggestion from these references. References regarding the concurrence of surface adsorption and solid-state diffusion are now included in Sect. 2.2, Line 149- 151.

'A diffusion-like behaviour has been observed from flow tube studies for trace gas uptake onto ice (e.g. Abbatt, 1997; Huthwelker et al., 2004; Cox et al., 2005) and suggested the solid-state diffusion of nitrate molecules can occur con-currently with surface adsorption, such that ...'

The reasons for adopting a kinetic approach instead of an equilibrium approach are listed in Sect. 3.1.1 and Sect. 6.1. The ice solubility parameterisation by Thibert et al., (1998) was obtained after exposing the ice with gaseous $HNO_3$ for a period of 1-4 weeks, however, no information and no conclusion on the time taken to reach equilibrium was presented.

2. The authors do not provide sufficient details about their model formulation of the disordered interface (DI) on the surface of the ice grain. How thick is the DI? Does the thickness of the DI change with temperature? Does it make sense to assume the fixed (constant) pH especially when the chemical composition of the DI is controlled predominantly by $HNO_3(gas) = H^+(DI) + NO_3^-(DI)$ at Dome C? These are the critical points that should be discussed in detail before rejecting the hypothesis of the $HNO_3$ incorporation into the DI.

The DI is treated as the boundary layer of the snow grain. The concentration of DI is used as the boundary condition for the solid-state diffusion of nitrate into the snow grain. Therefore, no thickness is assigned to the DI.

For clarification, the following lines (Page 9, line 275-277) are now included in the manuscript.

"Note that in this model the DI is treated as the boundary between the air and bulk ice. The concentration of the DI is used as the outermost boundary condition for solid- state diffusion within the grain, therefore, the DI has no arbitrary thickness."

The sensitivity of Model 1 to the value of pH in the range of pH found in natural surface snow (5-6.5, Udisti et. al, 2004) is shown in Table 4. Changing the pH within this range does not have an impact on the model performance.

3. It is not clear enough whether the kinetic limitation to the growth and decay of the snow grain $HNO_3$ concentrations is caused mainly by mass transfer between the gas phase and the grain surface or by solid diffusion into the entire volume of the snow gain. This question should be discussed in some detail especially when contrasting the behavior of $HNO_3$ between the "kinetic" and "equilibrium" approaches such as in Section 6.1. Also, the authors may want to refer to the work by Bock et al. (2016) on the matter of timescales due to various kinetic processes.

A table of the characteristic times of various physical processes are listed in Table 1. At low partial pressures of $HNO_3$, the characteristic time for surface adsorption to reach equilibrium is of the order of $10^3$ s.

4. I am puzzled by the description of the rate of snow grain growth and shrinkage in Section 3.1.1. Eq. (9) implies that the change of the snow grain volume is calculated by the molecular diffusion of water vapor through its microscopic concentration gradient around the snow grain. But then the authors admit that this approach does not work owing to the input data limitation and instead "the macroscopic (few mm) water vapour gradient across the skin layer was used to estimate the condensation and sublimation processes". Is the same equation still used for calculating dV/dt?

For clarification now on Page 8, line 253-256 now read:

"For simplicity the macroscopic (few mm) water vapour gradient across the skin 240 layer was used to estimate the rate of volume change of snow grain due to condensation or sublimation, i.e. $(d\rho\upsilon /dx )_{x=r}$ in Eq. 10 is replaced by $(d\rho\upsilon /dz)_{z=4mm}$."

In Sections 4.1 and 4.2, the authors state that meteorological input data have been obtained at 1.6 m and 1 m above the snow surface at Dome C and Halley, respectively. Is it then assumed that the water vapor concentrations are assumed to be constant with height between a few mm and 1-1.6 m above the snow surface? Please clarify.

Information regarding the relative humidity used for calculation of water vapour gradient has been clarified in Sect 4.1, line 372-373

"Based on the assumption of a well mixed boundary layer, the RH above the snowpack surface was assumed to be the same as what measured at 1.6 m"

Also, is it possible to validate the authors' macroscopic approach of calculating the water vapor flux by field observations if any? This seems to be important as background information for discussing the role of co-condensation in Section 6.2. Reference to an observed temperature gradient across the top 2 cm of the snowpack at Dome C has been added to support the statement in Sect 3.1.1, Line 227-231

"Field observations (Frey et al., 2013) and the results from a heat transfer model (Hutterli et al., 2003) at Dome C in summer show absolute temperature gradients of 71 K m$^{-1}$ across the tope 2 cm and 130 K m$^{-1}$ across the top 4 mm of the snowpack, respectively. "

By the way, I think $R_{eff}$ in Eq. (9) should be squared to be consistent in the physical dimension between LHS and RHS of the equation. Is it simply a typographic error?

Yes, the error in Eq 9. is now fixed, thank you.

5. The authors adopt the formulation of the α0 (Hudson et al., 2002), $N_{max}$ (Crowley et al., 2010) and $K_{eq}$ (Burkholder et al., 2015) from different sources. In fact, all of these could have been adopted from Crowley et al. (2010). It seems appropriate to discuss why the authors pick their experimental values/formulae from the different sources and how much difference their choice would generate in the model behavior.

Information and reasons for the choice of parameterisation are now listed in the Appendix.

[Figure]

**Figure A2.** Initial uptake coefficient for HNO₃ as a function of temperature obtained from different studies. The parameterisation used within this study is formulated in Table A1 and is chosen to give the best representation of the dependency on temperature.

[Figure]

**Figure A3.** Langmuir adsorption equilibrium constant, $K_{\mathrm{LinC}} = K_{eq} \times N_{\mathrm{max}}$. The preferred temperature range for both parameterisation is 214-240 K and within this range the two parameterisations provide a comparable value. The Crowley et al. (2010) parameterisation deviate from the Burkholder and Wine (2015) parameterisation as temperature drop below 214 K due to the exponential temperature term. Here, the parameterisation from Burkholder and Wine (2015) was chosen based on the extreme cold temperature found in our validation sites.

6. The quality of English needs to be improved significantly. There are so many grammatical and spelling errors, only a tiny part of which I can comment below as technical suggestions. This problem is really glaring but may be largely corrected by a copyeditor once the manuscript is accepted for publication. Nonetheless, there seems to be a room for improvement that should be addressed by the authors before that stage. I strongly recommend careful and diligent proofreading by the team of the authors (especially if the editor asks another round of review).

[Minor comments]
1. I think that "T – Tf" should be reversed to "Tf – T" in Eq. (4) to let φH2O(T) be the positive values. And I think that this inherits from what I believe is a typographic error in Cho et al. (2002) cited for Eq. (4). Am I wrong? Please double check.
It is inherited from a typographic error in Cho et al (2002) and it is corrected now to "Tf - T".

2. The variable "z" refers to the distance from the snow grain surface in Eq. (9), whereas it refers to the depth in the snowpack in Eq. (11). Please adjust the notation to avoid confusion between the two.
The variable "z" is replaced by variable "x" to avoid confusion for representing the microscopic distance.

3. On Line 92, it is stated that "thickness of the DI" is a tuning parameter in Toyota et al. (2014). In fact, they calculate the thickness of the DI on the basis of the Cho et al. formula, which is used by the present authors for calculating the volume of the micro pockets of brine. The difference from the present study is

that Toyota et al. assume the brine covers the entire surface of the snow grain just like the DI.
Yes, sorry for incorrect information. The statement has been removed.

4. On Line 255, it is stated that "Dg is the gas-phase diffusivity". It should be stated clearer that Dg is the gas-phase diffusivity of $HNO_3$. It would also be nice to list how $D_g$ is calculated in Table A1.
The calculation of $D_g$ is now listed in the footnote of Table 1.

5. Lines 615-616: It appears to me in Figure 1 that the changes in pH of the order of 1 have a similar level of impact on the effective Henry's law coefficient to the changes in temperature of the order of 10 K. I don't quite understand what the authors try to point out here.
The sentence has been removed.
The sensitivity of Model 1 to pH is now on Page 21, line 677 – 682

6. Lines 558-562, ". . ., which are 1-2 orders of magnitude higher than the averaged modelled temperature gradient (listed in Sect. 3.1.1)": It seems that this is not discussed/listed at all in Section 3.1.1. Please expand the discussion by referring to what the realistic range of the vertical temperature gradient should be.
Reference to Frey et. al (2013) is included (Page 7, line 215 and Page 18, line 589).
Frey et. al (2013)  measured a temperature gradient of 71 $Km^{-1}$ across the top 2 cm of snowpack in Dome C.

7. Lines 625-631 and Figure 11: Please be more specific and detailed about what make up the "other ions".
Fig. 11 is now in Fig. 8B and the caption has been changed to
 "…other ions, where other ions refers to the sum of [Na+] and [Cl−]"

8. Table A1: Sander (2015) is a compilation of Henry's law coefficients, but here it is cited for the temperature dependence of alpha. Please double check if it is the correct reference. Also, the "enthalpy of activation" is much too vague as terminology for ΔobsH. Please expand.
ΔobsH is now referred to as the enthalpy of uptake. Reference to Thomas (2011) is used instead of Sander (2015).

9. Table A1, values of ΔsolH and ΔobsH: I think they should have been −72.3 and −44, respectively (the minus sign is missing). Please double check. 10.
Yes, has been corrected.
Table A1, footnote i: I suppose that the authors meant to formulate the temperature dependence of alpha somehow consistently with d $\ln[\alpha/(1 - \alpha)]/d(1/T) = -\Delta obsH/R$ (e.g., Jayne et al., 1991). But I cannot reconcile with the authors' formulation in this footnote. Am I wrong here? Please double check if it is formulated properly.
The listed formulation is the integrated form of the equation from Jayne et al., 1991

[Technical suggestions]
Line 216: d $HNO_3$ / dt -> d [$HNO_3$(ads)] / dt
Yes, has been corrected.

Line 217: Substituting kads -> Substituting kdes
Yes, has been corrected.

Line 321: organic -> inorganic (?)
Yes, has been corrected.

Line 407: tough -> trough (?)
The sentence has been removed.

Line 414, Eq. (19): $MM_{H2O}$ -> $M_{H2O}$
Yes, is now corrected.

Line 603: . . . varying T0 by 4 K up to 242 K and pH by ±0.4 up and down between 5.2-6.4
The sentence is now removed.

Figure 1: Add the unit of temperature for the figure legends in (b): "T = 230 K", etc.
The units have been added to the figure legend.

Figure 10: Is it not possible to use the same scaling in Y-axis for all the data shown here, for example, by using logarithmic scaling?

Table A1: Accommodation coefficient at standard temperature -> Accommodation coefficient at reference temperature (220 K)
Yes, has been corrected.

Table A1, footnote ii: 258 K -> 298 K (?)
Yes, has been corrected.

[revised manuscript text omitted]
_{3,\,(\mathrm{ads})} \underset{}{\overset{k_{\mathrm{diff}}}{\rightleftarrows}} HNO_{3,\,(\mathrm{ice})} \tag{R2}$$

where  $HNO_{3,\,(\mathrm{ice})}$ is the nitric acid incorporated into the ice matrix, occurs with **??**.

**2.3 Coexistence of Liquid Solution with Ice**

195  Liquid aqueous solution coexists with ice in the presence of soluble impurities, such as sea salt and acids. The liquid exist down to the eutectic temperature  defined by the composition and solubility of the impurities in the ice. **?** parameterised the  liquid water fraction,  $\phi_{\mathrm{H_2O}}(T)$, as a function of total ionic concentration of impurities, $\mathrm{Ion_{tot}}$, and temperature as follows:

200
$$\phi_{\mathrm{H_2O}}(T) = \frac{\overline{m}_{\mathrm{H_2O}} R T_f}{1000 \Delta H_f^0} \left( \frac{T}{T - T_f} \frac{T}{T_f - T} \right) \Phi_{\mathrm{bulk}}^{\mathrm{aq}} \left[ \mathrm{Ion_{tot\,(bulk)}} \right] \tag{4}$$

[revised manuscript text omitted]

$$[\text{HNO}_{3(\text{surf})}] = [\text{HNO}_{3(\text{ads})}] + [\text{HNO}_{3(\text{cc})}] \qquad \text{if} \quad T \leq 238\text{K} \qquad (7)$$

where $[\text{HNO}_{3(\text{ads})}]$ is the concentration contributed  by the sum of surface adsorption and desorption (Eq. **??**), and $[\text{HNO}_{3(\text{cc})}]$ is the concentration contributed  by co-condensation  or co-sublimation (Eq. **??**).

A non-equilibrium kinetic approach is taken instead of saturation or equilibrium adsorption for  two main reasons: Firstly, **?** have shown that for partial pressures of $\text{HNO}_3$ lower than $10^{-5}$ Pa the ice surface is not entirely covered and therefore undersaturated. The annual average atmospheric partial pressure of $\text{HNO}_3$ recorded at Dome C is $\sim 10^{-6}$ Pa (**?**) and is $\sim 10^{-7}$ Pa at Halley (**?**), hence, the ice surface is unlikely to be saturated with $\text{HNO}_3$. Secondly, natural snowpacks are constantly undergoing sublimation and condensation of $\text{H}_2\text{O}$, especially at the skin layer, due to temperature gradient over a range of timescales from a fraction of seconds to  days and seasons (**?**). **?** observed up to 60% of the total ice mass redistributed under a constant temperature gradient of 50 $\text{K m}^{-1}$ over a 12 hour period.  Field observations (**?**) and the results from a heat transfer model (**?**) at Dome C in summer show absolute temperature gradients of 71 $\text{K m}^{-1}$ across the top 2 cm and 130 $\text{K m}^{-1}$ across the top 4 mm of the snowpack , respectively. At Halley, the  modelled summer absolute temperature gradient  in the top cm of snow is about 41  $\text{K m}^{-1}$. Therefore, the  dynamic $\text{H}_2\text{O}$ exchange and redistribution at the snow grain

surface prevent the equilibrium of adsorption from being reached and require a kinetic approach. The net rate of adsorption can be described as $\frac{d\text{HNO}_3}{dt} = k_{\text{ads}}[\text{HNO}_{3(\text{g})}][\text{S}] - k_{\text{des}}[\text{HNO}_{3(\text{ads})}]$.  $\frac{d[\text{HNO}_{3(\text{ads})}]}{dt} = k_{\text{ads}}[\text{HNO}_{3(\text{g})}][\text{S}] - k_{\text{des}}[\text{HNO}_{3(\text{ads})}]$. Substituting $k_{\text{des}}$ with Eq.

(**??**), the net adsorption rate is expressed as

$$\frac{d[\text{HNO}_{3\,(\text{ads})}]}{dt} = k_{\text{ads}}\left([\text{HNO}_{3\,(\text{g})}]\,[\text{S}] - \frac{[\text{HNO}_{3\,(\text{ads})}]}{K_{\text{eq}}}\right) \tag{8}$$

The temperature gradient and relative humidity gradient between the surface of the snowpack and the skin layer create a gradient in water vapour pressure, which drives condensation or sublimation of ice, depending on the sign of the gradient.  Uptake of $\text{HNO}_3$ molecules to growing ice is known as co-condensation. The surface  concentration of $\text{NO}_3^-$ contributed by co-condensation or co-sublimation, $[\text{HNO}_{3\,(\text{cc})}]$, is given by

$$[\text{HNO}_{3(\text{cc})}] = X_{\text{HNO}_3}\,\frac{\rho_{ice}\,N_A}{\overline{m}_{\text{H}_2\text{O}}}\,\frac{\Delta t}{\text{V}_{\text{grain}}}\,\frac{d\text{V}}{dt} \tag{9}$$

where $X_{\text{HNO}_3}$ is the mole fraction of $\text{HNO}_3$ condensed along with water vapour ($X_{\text{HNO}_3} = \frac{P_{\text{HNO}_3}^{0.56}}{10^{3.2}}10^{-3.2}\,P_{\text{HNO}_3}^{0.56}$; **?**), $\rho_{ice}$ is the density of ice (in $\text{kg m}^{-3}$), $N_A$ is  Avogadro's constant ($6.022\times10^{23}\,\text{molecule mol}^{-1}$) and $\Delta t$ is the model time step. The rate of volume change of snow grain, $\frac{dV}{dt}$, is specified by the growth law by described (**?**)

$$\frac{d\text{V}}{dt} = \frac{4\,\pi\,\text{R}_{\text{eff}}}{\rho_{ice}}\,\frac{4\,\pi\,\text{R}_{\text{eff}}^2}{\rho_{ice}}\,D_v\left(\frac{d\rho_v}{dz}\,\frac{d\rho_v}{dx}\right)_{z=r\,x=r} \tag{10}$$

where  $D_v$ is the diffusivity of water vapour in air and $\frac{d\rho_v}{dz}\,\frac{d\rho_v}{dx}$ is the local water vapour density gradient, i.e. between air away from the snow grain and the air near the grain surface. However, to the author's knowledge there are no observations reported and the calculation of water vapour density at these microscopic scales is computational costly as it would require 3-D modelling of the metamorphism of the snow grain. For simplicity, the macroscopic (few $\text{mm}$) water vapour gradient across the skin layer was used to estimate the   rate of volume change of snow grain due to condensation or sublimation, i.e. $\left(\frac{d\rho_v}{dx}\right)_{x=r}$ in Eq. **??** is replaced by $\left(\frac{d\rho_v}{dz}\right)_{z=4\text{mm}}$. The water vapour density, $\rho_v$,  can be calculated as follows:

$$\rho_v = \frac{P_{sat}\,\text{RH}}{100\,R_v\,T} \tag{11}$$

[revised manuscript text omitted]

---

## Author Response (AR2)

We thank the reviewers for their reviews and recommendation to publish. We have considered every point and corrected the paper to include their points. The referees comments are in black, responds from the authors are in blue and revised text are in red.

Report #1
[Specific comments]

1.      The authors now explicitly argue that "the solid-state solution of nitrate molecules can occur concurrently with surface adsorption" (L150-151). If I understand correctly, the present model formulation dictates that a thermodynamic limitation does not exist anymore once HNO3 is retained on/in the outermost layer of their hypothesized spherical ice grain. As such, HNO3 trapped on the ice surface via Langmuir adsorption is assumed to be transferred instantaneously across the depth of the order of several micrometers to the outermost layer of bulk ice and subject to solid-state diffusion further inside based on the diffusivity having been determined experimentally. I am inclined to disagree to the claim that this model formulation is entirely based upon the first principles.
To avoid confusion we now emphasize throughout the paper that the models are based on physical paramaterisations and laboratory data.

A clarification has been added to the text (See below in red) in Section 3.1.3 (line 312-317) to explain what boundary conditions are used.
"The concentration gradient between the grain boundary and its centre drives solid state diffusion of nitrate within the bulk ice. The $NO_3^-$ concentration profile within the snow grain can be found by solving the following partial differential equation
$$\frac{\partial U(r)}{\partial t} = k_{diff}(\frac{\partial^2 U(r)}{\partial r^2})$$
where U(r) is the concentration at distance r from the centre of the snow grain and $k_{diff}$ is the solid-state diffusion coefficient, which is assumed to be homogeneous across the snow grain. The nitrate concentration at the centre is set to U(0) = 0 and at the grain boundary $U(R_{eff})$ = [HNO3(surf)] is defined by surface adsorption and co-condensation at temperatures below To (Eq. 7) or by solvation into the infinitesimal DI at temperature above To (Eq. 13)."

2.      It appears to me that the treatment of the surface to bulk mass transfer of HNO3 is even more problematic in "Model 1" where the authors assume that HNO3 dissolved in the DI is subject to solid-state diffusion into the bulk volume of ice without thermodynamic limitation. This assumption effectively asserts that the equilibrium solubility of HNO3 in the entire volume of solid ice is identical to that in the liquid-like DI. Does any of the earlier studies advocating the role of DI assume such an equivalence between the DI and the bulk ice? So I still feel that the present study does not really offer the refutation of the HNO3 uptake onto the DI as a viable process.
The text has been clarified in Sect 3.1 (line 216-230) about the assumptions made regarding the DI in this paper.
'The physical properties of the DI, such as the layer thickness, partitioning coefficient, diffusivity etc., are still poorly understood. The laboratory

measurements of thickness of DI of pure ice range from a monolayer of water to around a few hundreds of nm (Bartels-Rausch et al., 2014) depending on the measuring techniques and temperature. Thur there is no parameterisation available to estimate the thickness of DI as a function of temperature and/or concentration within the bulk. Also, no measured 230 values are available for the air-DI partitioning and the diffusivity of the DI. Therefore, for the DI in Model 1 has the following is assumed: 1) the partitioning between air and the DI is governed by Henry's law; 2) the DI is interacting with the bulk ice, which the nitrate molecules solvated into the DI are allowed to diffused into the bulk ice, the rate of the transport is limited by the diffusivity of solid ice; 3) the DI has an infinitesimal thickness and the concentration in the DI is acting as the boundary condition of the solid-state diffusion into the snow grain (See Sect. 3.1.3). Note that besides adopting Henry's law coefficient as the partitioning coefficient of the DI, the other assumptions made here for the DI is different from the assumptions made by previous models (e.g. Thomas et al., 2011; Toyota et al., 2014) that often assume the DI has a certain arbitrary thickness.'

3.      In section 5.1.1, the authors refer to their model results based upon "Kinetic approach" and "Equilibrium approach" suggested earlier by Bock et al. (2016). These models are not exactly the same as the standard suite of models, "Model 1" and "Model 2", in the present study. Hence the authors are apparently nudged to the names of the models used in Bock et al. (2016). I was confused when I first read this paper and again when I read the revised manuscript, partly because the transfer of HNO3 into liquid micro-pockets is treated by equilibrium in "Model 2". I would rather like the authors to streamline the naming convention so that the "Kinetic" and "Equilibrium" models referred to here are called by numbers. For example, the "Kinetic" model can be a sub-category of "Model 1". It is of course still useful to refer to the fact that these models are based on the "Kinetic" and "Equilibrium" approaches by Bock et al. (2016). Also, I would have liked it if the authors had introduced and discussed Eq. (19) in section 3, even though it does not belong to their standard model formulation. Theses are good suggestions: the equation Eq. (19) after Bock et al. 2016 approach is now moved to Sect. 3 and their approach is referred as the 'Bock-BC1' and the kinetic approach taken in this study is now referred as the 'Model 1-BCice', where BC stand for "boundary condition".

[Minor comments and technical suggestions]

L70-71, "Murray et al., 2015; 3)": An apparent typo.
Yes, this has been corrected
L77-78: Please double check if this sentence makes sense.
Yes, this has been corrected, now written as "The bulk concentration of $H_2O_2$ is determined by solid-state diffusion of $H_2O_2$ while the bulk concentration of HCHO is determined by linear isotherm adsorption of HCHO on ice."

L86: dynamicS
Yes, this has been corrected

L177: "a 2" -> "two"
Yes, this has been corrected

L178: "one" -> "an"
Yes, this has been corrected

L241: "… growth law described by Flanner and Zander (2006):"
Yes, this has been corrected

L334: below THE freezing POINT
Yes, this has been corrected

L354: "particulate nitrate" -> "total particulate and gaseous nitrate" (?)
Yes, this has been corrected as "atmospheric nitrate"

L389: well-mixed SO that
Yes, this has been corrected

L425: Cv(RMSE) = 0.73 (Table 4)
Yes, this has been corrected

L508: "inf" -> "in"
Yes, this has been corrected

L512-513: "surface, layer leading to" -> "surface layer, leading to"
Yes, this has been corrected

L513: "find in a more stable" -> "found in the particulate"
Yes, this has been corrected

L522: information IS AVAILABLE on
Yes, this has been corrected

L525 & L550: "maybe" -> "may be"
Yes, this has been corrected

L581: requireD
Yes, this has been corrected

L599 & L670: "larger" -> "higher"
Yes, this has been corrected

L665: "sensitivity" -> "sensitive"
Yes, this has been corrected

Table 1: Mathematical expressions (second column) for "Interfacial mass transport to a liquid phase" and "Gas-phase diffusion to the surface of a spherical droplet" should be inverted.

Yes, this has been corrected

Report #2

The authors have performed an interesting modeling study of an important and valuable dataset. The technical work is a worthwhile contribution to the literature; some new approaches are introduced and reasonable agreement with field data is achieved. However, the results are not currently presented in a manner that is acceptable for publication. The authors disparage previous work in the text for requiring tuning parameters and unrealistic assumptions. This is a fair criticism of nearly all modeling studies in this field, since fundamental understanding lags significantly behind our needs for process modeling, and therefore assumptions are required.

The problem is that the authors present their modeling approaches as being superior to previous work without acknowledging up front the great uncertainty surrounding several of the "first principals" assumptions made in this study. To avoid confusion, we now emphasize the models developed here are based on physical paramaterisations and laboratory data. This paper has been edited throughout to reflect this.

It is not my intention to call out these assumptions as erroneous, but rather to insist that the authors acknowledge them and discuss the uncertainties that they introduce. For example, in model 1: The range of the DI onset temperature, To, is based on experimental data, but it should be acknowledged that this is a type of tuning parameter.

We agree in considering the uncertainties is important, thus, the uncertainties presented in reported for experimental observations are considered in a sensitivity analysis in Sect 6.5.

However, threshold temperature $T_0$ is a parameter with physical meaning. Therefore, it is not an arbitrary tuning parameter. It is varied within the range of available experimental data, which might be narrowed in the future. The model uncertainties due to the uncertainty in the $T_0$ is explored in a sensitivity test presented in Sect 6.5.

Another assumption must be made regarding the depth/volume of the DI, although I did not see this discussed in the text. Uptake of HNO3 to the DI is assumed (as it is in nearly all snowpack modeling studies, for lack of a better option) to follow Henry's Law for bulk aqueous solutions, where in reality there is no physical basis for this assumption.
The DI in this study is assumed to have an infinitesimal thickness. Other assumptions made regard to the DI in this study have now been clarified in Sect. 3.1 (line 216-230)
'The physical properties of the DI, such as the layer thickness, partitioning coefficient, diffusivity etc., are still poorly understood. The laboratory measurements of thickness of DI of pure ice range from a monolayer of water to around a few hundreds of nm (Bartels-Rausch et al., 2014) depending on the measuring techniques and temperature. Thur there is no parameterisation available to estimate the thickness of DI as a function of temperature and/or

concentration within the bulk. Also, no measured 230 values are available for the air-DI partitioning and the diffusivity of the DI. Therefore, for the DI in Model 1 has the following is assumed: 1) the partitioning between air and the DI is governed by Henry's law; 2) the DI is interacting with the bulk ice, which the nitrate molecules solvated into the DI are allowed to diffused into the bulk ice, the rate of the transport is limited by the diffusivity of solid ice; 3) the DI has an infinitesimal thickness and the concentration in the DI is acting as the boundary condition of the solid-state diffusion into the snow grain (See Sect. 3.1.3). Note that besides adopting Henry's law coefficient as the partitioning coefficient of the DI, the other assumptions made here for the DI is different from the assumptions made by previous models (e.g. Thomas et al., 2011; Toyota et al., 2014) that often assume the DI has a certain arbitrary thickness.'

For model 2, eq. 4, which was developed assuming that the liquid solution is ideal and does not take into account partitioning of nitrate between the gas, liquid, and solid phases, is applied without discussion.
In Model 2, the partitioning of nitrate between air and ice is included as shown in the first term of Eq. 17, which reference back to Sect 3.1.1 as the same air-ice processes were applied.
In Section 3.2, the following lines been edited (line 341-457)
"The term '$\Sigma[NO_3^-](r) V(r) / V_{grain}$' in Eq. 17 is representing the nitrate concentration in the ice-phase and is applied to all temperatures below the melting temperature, $T_m$. At $T<T_m$, HNO3 can be adsorbed/desorbed and co-condensed/co-sublimated from the ice surface as was the case in Model 1 when $T <T_o$ (Sect. 3.1.1). The adsorbed and co-condensed molecules on the ice surface then diffuse into or out of the bulk ice depending on the concentration gradient of nitrate anion as was the case in Model 1 (Sect. 3.1.3). The nitrate in the snow grain contributed by these processes is referred to
as the ice-phase nitrate."
and line 351-353
"The liquid in the micropocket is assumed to be ideal and the partitioning between air and liquid micropocket is described by Henry's Law (Eq. 5). This implies instantaneous equilibrium between air and liquid pocket, and is justified because;..."

In Section 6.4, the following line (722-734) been added
'Moreover, the liquid in the micropocket is assumed to behave ideally and, therefore, Henry's coefficient is used to describe the partitioning between air and the micropocket. In reality, there may be some deviation from ideality as the concentration of solutes in the micropocket is likely to be too large to be considered as an ideal dilute solution. The non-ideality should be accounted for in terms of activity coefficient, $\gamma$. At equilibrium, the relationship between a solute B and the solvent can be expressed as follow (Sander, 1999):
$$K_B = \frac{\gamma_B x_B}{P_B}$$
where $P_B$ is the vapour pressure of B, $\gamma_B$ is the activity coefficient of B and $x_B$ is the mole fraction of B. The value of the activity coefficient approaches unity as the mole fraction of B approaches zero ($\gamma_B \rightarrow 1$ as $x_B \rightarrow 0$) and, under such ideal-dilute condition, the equilibrium constant, $K_B$, is defined as Henry's

coefficient. Values of activity coefficient can be found experimentally. The available parameterisation of activity coefficient of $HNO_3(aq)$, $H^+$ and $NO_3^-$ is only accurate for concentration up to 28 m (Jacobson , 2005). When the molarity is higher than $\sim$4-5 m, depending on the temperature, the activity coefficient of H+ and $NO_3^-$ increases as molarity increase. The concentration of the micropocket is estimated based on the parameterisation by Cho et al. (2002), which predicts a concentration a lot larger than the limit of activity coefficient parameterisation available at present. Hence, it is not possible to quantify the uncertainties caused by assuming the micropocket has ideal-solution behaviour. If the relationship between activity coefficient and molarity extend to larger molarity than 28 m, the activity coefficient would be larger than 1 and hence reduces the value of the equilibrium constant, $K_B$, compared to the Henry's Law coefficient. By means, the assumption of ideal-solution behaviour of micropocket is likely to overestimate the concentration of the micropocket. The activity coefficient of highly concentrated solution is needed to be found by further experimental studies. '

The authors should also take care to acknowledge when aspects of their models are adopted from or are similar to previous modeling studies, in addition to highlighting their innovations.

Previous modeling studies had been acknowledge, for example

[revised manuscript text omitted]

$$\text{HNO}_{3,(\text{g})} + \text{S} \underset{k_{\text{des}}}{\overset{k_{\text{ads}}}{\rightleftharpoons}} \text{HNO}_{3,(\text{ads})} \tag{R1}$$

150 where HNO$_{3,(\text{g})}$ and HNO$_{3,(\text{ads})}$ are the gas-phase and surface adsorbed nitric acid and $S$ is the surface site for adsorption. The concentration of surface site, i.e. number of site available per unit volume of air, is defined as follow:

$$[\text{S}] = (1 - \theta) \, N_{max} \, \frac{\text{A}_{\text{ice}}}{\text{V}_{\text{air}}} \tag{1}$$

Here, $\theta$ is the fraction of  surface sites being occupied, $N_{max}$ is the maximum number of
155 surface sites with a unit of molecule m$_{\text{ice}}^{-2}$, A$_{\text{ice}}$ is the surface area of ice per unit volume of snowpack with a unit of m$_{\text{ice}}^2$ m$_{\text{snowpack}}^{-3}$, and V$_{\text{air}}$ is the volume of air per unit volume of snowpack with a unit of m$_{\text{air}}^3$ m$_{\text{snowpack}}^{-3}$. Note that $[S]$ has a units of molecule m$^{-3}$. The adsorption coefficient, $k_{\text{ads}}$ ,and desorption coefficient, $k_{\text{des}}$, in R1 are defined as

$$k_{\text{ads}} = \frac{\alpha \overline{v}}{4} \frac{1}{N_{max}} \tag{2}$$

160 $$k_{\text{des}} = \frac{k_{\text{ads}}}{K_{\text{eq}}} \tag{3}$$

Note that $k_{\text{ads}}$ has a unit of $\text{m}^3\,\text{molecule}^{-1}\,\text{s}^{-1}$ while the unit of $k_{\text{des}}$ is $\text{
[revised manuscript text omitted]

---

## Author Response (AR3)

Thanks for all the comments and recommendation to publish. We have considered every point and corrected the paper to include their points. The editor comments are in black, responds from the authors are in blue and revised text are in red.

One point is connected to the thickness of the DI and eq. 14 What means infinitesimal thickness mathematically precisely and how do you define a concentration in such a DI with a volume of or approaching 0? With this question, I also refer to the molecular budget, which I'm sure you have looked at, but which would be worth mentioning.
Appendix B has been added to descript the deviation of equation 14.

**Appendix B: Derivation for non-equilibrium kinetics**

850 The processes involved in the equilibrium of the gas-phase and the surface of a droplet (Fig. A5):
1) Gas-phase diffusion from far away (> μm) from the droplet to the surface of the droplet, which is likely to be driven by turbulence and molecular diffusion; 2) Interfacial mass transport; and 3) Condensed-phase diffusion and chemical reactions;

[Figure]

**Figure A5.** Processes involve in the equilibrium between gas-phase and condensed-phase, where $c_{g,\infty}$ is the gas-phase concentration in the SIA far away from the droplet, $c_{g,surf}$ is the gas-phase concentration at the surface (outside the droplet), $c_{c,surf}$ is the condensed-phase concentration at the surface (inside the droplet) and $c_c$ is the average condensed-phase concentration.

Transport of gas-phase species from the SIA to the surface of the droplet can be described using
855 Fick's law as diffusion flux, $J_g$:

$$J_g = -D_g \frac{dc_g}{dx} \tag{B1}$$

where $D_g$ is the gas-phase diffusivity, and $\frac{dc}{dx}$ is the concentration gradient at the droplet surface that $\frac{dc_g}{dx} = \frac{c_{g,\infty} - c_{g,surf}}{R_{eff}}$ with $R_{eff}$ as the radius of the droplet. The concentration change in the condense-phase can be expressed as

$$\quad \frac{dc_c}{dt} = \frac{A J_g}{V} = -\frac{A}{V} \frac{D_g}{R_{eff}} (c_{g,\infty} - c_{g,surf}) \tag{B2}$$

where $A$ is the surface area of the droplet and $V$ is the volume of the droplet. The first-order rate coefficient for the gas-phase diffusion process can be defined as $k_{dg} = \frac{A}{V} \frac{D_g}{R_{eff}}$ (Sander, 1999). For an example, a liquid droplet with a radius $R_{eff}$ the gas-phase diffusion rate coefficient $k_{dg} = \frac{3D_g}{R_{eff}^2}$.

The interfacial mass transport from gas-phase to condensed-phase can be expressed in terms of
865 accommodation coefficient, $\alpha$. The flux through the phase boundary into the droplet, $J_b^{tn}$, is defined as:

$$J_b^{tn} = \frac{\alpha \bar{v}}{4} c_{g,surf} \tag{B3}$$

where the subscript $b$ stands for 'boundary' and $\bar{v}$ is the mean molecular velocity. The opposite flux, $J_b^{out}$, through the phase boundary out of the droplet can be expressed in the similar form as Eq.

870 B3 that $J_b^{out} = \frac{\alpha_c \bar{v}_c}{4} c_{a,surf}$, where $\bar{v}_c$ is the mean molecular velocity in condensed-phase and $\alpha_c$ is the condensed-phase accommodation coefficient. The net flux through the grain boundary, $J_b$, is the difference between the in and out flux.

$$J_b = J_b^{in} - J_b^{out} = \frac{\alpha \bar{v}}{4}\left(\frac{c_{c,surf}}{K} - c_{g,surf}\right) \qquad (B4)$$

where $K$ is the equilibrium constant, of which $K = c_{c,surf}^{eq}/c_{g,surf}^{eq}$. For example, for a gas-aqueous
875 interface, the ratio of aqueous-phase concentration to gas-phase concentration at equilibrium can be described as $c_{a,surf}^{eq}/c_{g,surf}^{eq} = k_H^{cc}$, where $c_{a,surf}$ is the aqueous-phase concentration at the surface and $k_H^{cc}$ is the Henry's constant. The concentration change in the condensed phase due to interfacial mass transport can be expressed as:

$$\frac{dc_c}{dt} = -\frac{A J_b}{V} = \frac{A}{V}\frac{\alpha \bar{v}}{4}\left(c_{g,surf} - \frac{c_{c,surf}}{K}\right) \qquad (B5)$$

880 The first-order rate coefficient for the interfacial mass transport, $k_b$, to a droplet with a radius $R_{eff}$ can then be defined as $k_b = \frac{3\alpha \bar{v}}{4}R_{eff}$. By assuming the fluxes of gas-phase diffusion, $J_g$, is equal to the interfacial mass transport, $J_b$, the rate of change of concentration in the condensed phase can be expressed as

$$\frac{dc_c}{dt} = \frac{A}{V}\left(\frac{R_{eff}}{D_g} + \frac{4}{\bar{v}\alpha}\right)^{-1}\left[c_{g,\infty} - \frac{c_{c,surf}}{K}\right] \qquad (B6)$$

885 the term '$\frac{A}{V}\left(\frac{R_{eff}}{D_g} + \frac{4}{\bar{v}\alpha}\right)^{-1}$' is often referred as the mass transfer coefficient, $k_{mt}$, for a chemical species transfer from air to liquid/solid. The mass transfer coefficient for chemical into a spherical droplet with radius $R_{eff}$ is $k_{mt} = (\frac{r^2}{3D_g} + \frac{4R_{eff}}{3\bar{v}\alpha})^{-1}$ and if the surface of the droplet is described as DI then the concentration at the grain surface, $c_{c,surf} = [HNO_{3,DI}]$.

Did I understand correctly, that the flux into the DI matches the flux from the DI into the bulk and that the concentration in the DI based on Henry is established? Yes, that is correct.
If so, I do not understand the statement on page 25 (825): "In general, the grain boundary concentration of nitrate defined by solvation into the DI is much larger than when it is defined by the combination of surface adsorption and co-condensation on ice." Does this imply that the fluxes are balanced? The statement is refer to the boundary of the snow grain, the statement is now written as follow for clarity
"The concentration of the nitrate at the grain boundary, $U(R_{eff})$, have a much larger value when the interface between air and grain boundary is defined as 'Air-DI' (Eq. 13) than when it is defined as 'Air-Ice' (Eq. 7). "

The second aspect touches question 2 of report 1: You don not explicitly mention an upper limit of solubility in the bulk ice. That of the DI is given by Henry. That of the inner ice is given by the solid solution? If correct, I suggest to clearly state this (lines 216-230) to prevent the impression that the whole grain can become liquid-like as the DI holding such high concentrations of solutes.
It has been clarified in P.8 line 239
" 3) the DI has an infinitesimal thickness and the concentration of nitrate in the DI is acting as the boundary condition of the solid-state diffusion into the snow grain, which the solid-state concentration of nitrate in the bulk is limited by the solubility of ice. "

Some minor suggestions:

Page 3, line 90: Please define Co-condensation here.

Definition of co-condensation is added: "contributed by co- condensation, which is the simultaneous condensation of water vapour and trace gases at the air-ice interface, has an empirical relationship"

Page 4, line 124. I would not say that the idea of liquid co-existing with ice comes from the Domine paper. This is given by thermodynamics; Domine strongly argued for arrangement in pockets as you state later. I suggest to rather cite Cho or McNeill, ACP (2011)

The statement (P.4 line 113 ) is now cited to Cho et. al. (2002)

Page 7, line 210: I suggest to mention that the pH in the liquid content of the ice is not equal to the pH of molten snow.

The following sentence been added on P.7 line 197-199

" Note that the range of pH measured by Udisti et al. (2004) is the pH of the melted sample, which might be different from the pH of the ice co-existed liquid. However, the pH of the liquid water co-existing with the ice cannot be measured with the current techniques yet. "

Page 9, line 275: The adsorption as shown by Ullerstam might be under saturated, but it is still in equilibrium. So, I can't quite follow this argument.

The sentences have now been rearrange to explain why the ice surface is not saturated and not in equilibrium. P. 8, line 256 – 263:

"Ullerstam et al. (2005b) have shown that for partial pressures of $HNO_3$ lower than $10-5$ Pa the ice surface is not entirely covered with $HNO_3$, and therefore, undersaturated. The annual average atmospheric partial pressure of $HNO_3$ recorded at Dome C is $\sim 10-6$ Pa (Traversi et al., 2014) and is $\sim 10-7$ Pa at Halley (Jones et al., 2008), hence, the ice surface is unlikely to be saturated with $HNO_3$. A non-equilibrium kinetic approach is taken instead of an equilibrium adsorption as natural snowpacks are constantly undergoing sublimation and condensation of $H_2O$, especially at the skin layer, due to temperature gradient over a range of timescales from a fraction of seconds to days and seasons (Bartels-Rausch et al., 2014). "

Page 12, line 371:replace water with brine or solution.

"liquid water " has been replaced with "liquid solution"